# A Geometry-Aware Efficient Algorithm for Compositional Entropic Risk Minimization

Xiyuan Wei [1] [*]    Linli Zhou [1] [*]    Bokun Wang [2]    Chih-Jen Lin [3] [4]    Tianbao Yang [1]

## Abstract

This paper studies optimization for a family of problems termed **compositional entropic risk minimization**, in which each data's loss is formulated as a Log-Expectation-Exponential (Log-E-Exp) function. The Log-E-Exp formulation serves as an abstraction of the Log-Sum-Exponential (LogSumExp) function when the explicit summation inside the logarithm is taken over a gigantic number of items and is therefore expensive to evaluate. While entropic risk objectives of this form arise in many machine learning problems, existing optimization algorithms suffer from several fundamental limitations including non-convergence, numerical instability, and slow convergence rates. To address these limitations, we propose a geometry-aware stochastic algorithm, termed **SCENT**, for the dual formulation of entropic risk minimization cast as a min–min optimization problem. The key to our design is a **stochastic proximal mirror descent (SPMD)** update for the dual variable, equipped with a Bregman divergence induced by a negative exponential function that faithfully captures the geometry of the objective. Our main contributions are threefold: (i) we establish an $O(1/\sqrt{T})$ convergence rate of the proposed SCENT algorithm for convex problems; (ii) we theoretically characterize the advantages of SPMD over standard SGD update for optimizing the dual variable; and (iii) we demonstrate the empirical effectiveness of SCENT on extreme classification, partial AUC maximization, contrastive learning and distributionally robust optimization, where it consistently outperforms existing baselines. Code is available at github.com/Optimization-AI/SCENT.

[*]Equal contribution  [1]Texas A&M University [2]University of Texas, Austin [3]National Taiwan University [4]Mohamed bin Zayed University of Artificial Intelligence. Correspondence to: Tianbao Yang <tianbao-yang@tamu.edu>.

*Proceedings of the 43$^{rd}$ International Conference on Machine Learning*, Seoul, South Korea. PMLR 306, 2026. Copyright 2026 by the author(s).

## 1. Introduction

This paper considers the following optimization problem:

$$\min_{\mathbf{w}\in\mathcal{W}} F_{\text{CERM}}(\mathbf{w}) := \frac{1}{n}\sum_{i=1}^{n}\log\left(\mathbb{E}_{\zeta\sim\mathbb{P}_i}\exp(s_i(\mathbf{w};\zeta))\right), \quad (1)$$

where $\mathcal{W}\subset\mathbb{R}^d$, $\mathbb{P}_i$ denotes a distribution and $s_i(\mathbf{w};\zeta): \mathbb{R}^d \to \mathbb{R}$ denotes a random risk function associated with an anchor data $i$. Since in risk-averse decision making (Föllmer & Schied, 2010), the Log-E-Exp function $\log\left(\mathbb{E}_{\zeta\sim\mathbb{P}_i}\exp(s_i(\mathbf{w};\zeta))\right)$ is called the entropic risk, we term the above problem as Compositional Entropic Risk Minimization (**CERM**).

CERM abstracts important yet challenging machine learning problems in broad applications. We give two examples below. The well-known multi-class logistic regression aims to optimize the following cross-entropy loss for a set of training data $\{\mathbf{x}_i, y_i\}_{i=1}^{n}$,

$$\min_{\mathbf{w}\in\mathcal{W}} \frac{1}{n}\sum_{i=1}^{n}\log\left[\sum_{k=1}^{K}\exp(h(\mathbf{x}_i)^{\top}(\mathbf{w}_k - \mathbf{w}_{y_i}))\right], \quad (2)$$

where $h(\mathbf{x}_i)\in\mathbb{R}^d$ denotes the given feature vector of $\mathbf{x}_i$ and $y_i \in \{1,\dots,K\}$ denotes $\mathbf{x}_i$'s class label, and $\mathbf{w} = (\mathbf{w}_1,\dots,\mathbf{w}_K)$ denotes the weight vectors of the model. The log-sum-exp function naturally arises from the negative log-likelihood induced by the softmax function $\frac{\exp(h(\mathbf{x}_i)^{\top}\mathbf{w}_{y_i})}{\sum_{k=1}^{K}\exp(h(\mathbf{x}_i)^{\top}\mathbf{w}_k)}$ for each data. If we let $\mathbb{U}_{[K]}$ denote uniform distribution over $\{1,\dots,K\}$ and $s_i(\mathbf{w};\zeta) = h(\mathbf{x}_i)^{\top}\mathbf{w}_{\zeta} - h(\mathbf{x}_i)^{\top}\mathbf{w}_{y_i}$ for $\zeta\sim\mathbb{U}_{[K]}$, the multi-class logistic regression problem then becomes a special case of CERM. The expectation $\mathbb{E}_{\zeta\sim\mathbb{U}_{[K]}}$ captures the challenge that the number of classes $K$ is gigantic so that the summation inside the logarithmic function cannot be easily computed. This problem is known as the extreme classification (XC) problem (Bengio et al., 2019).

The second example arises in partial AUC maximization for imbalanced binary classification. Let $\mathcal{S}_+ = \{\mathbf{x}_i^+\}_{i=1}^{n_+}$ denote a set of $n_+$ positive examples and $\mathcal{S}_- = \{\mathbf{x}_i^-\}_{i=1}^{n_-}$ denote a set of $n_-$ negative examples. For imbalanced classification problem ($n_+ \ll n_-$), one-way partial AUC maximization aims to learn a model $\mathbf{w}$ to maximize the par-

tial area under the ROC curve, which has been formulated into the following optimization problem (Zhu et al., 2022):

$$\min_{\mathbf{w}\in\mathcal{W}} \frac{1}{n_+} \sum_{i=1}^{n_+} \tau \times \tag{3}$$

$$\log \left[ \frac{1}{n_-} \sum_{j=1}^{n_-} \exp \left( \frac{\ell(\mathbf{w}^\top (h(\mathbf{x}_j^-) - h(\mathbf{x}_i^+)))}{\tau} \right) \right],$$

where $\tau > 0$ is a hyperparameter, and $\ell(\cdot) \geq 0$ is a non-decreasing surrogate loss function. As a result, if we let $s_i(\mathbf{w}; \zeta) = \ell(\mathbf{w}^\top (h(\zeta) - h(\mathbf{x}_i^+)))/\tau$ with $\zeta$ being a random sample from $\mathcal{S}_-$, then the above problem becomes an instance of CERM. Other examples arise in contrastive losses for representation learning (Yuan et al., 2022; Wang & Isola, 2020), listwise cross-entropy loss for learning to rank (Xia et al., 2008), and KL-regularized distributionally robust optimization (Qi et al., 2021; Li et al., 2021).

The unique challenge of CERM is that both the inner expectation and the out summation (for a large $n$) are expensive to evaluate. While different techniques have been proposed to address this challenge, including mini-batch approximation, compositional optimization and optimizing a dual formulation, they suffer from several notable limitations (please refer to next section for details). The limitations include: (i) lack of convergence guarantee when biased gradient estimators are employed; (ii) numerical instability arising from the exponential function; and (iii) slow theoretical convergence for convex problems, often accompanied by coarse-grained analyses that overlook the impact of exponentially large constants in convergence bounds. This paper aims to design a better stochastic algorithm with an improved convergence analysis under convexity. Our algorithm is based on solving an equivalent min–min optimization problem derived from the dual formulation of the entropic risk (Ben-Tal & Teboulle, 1986):

$$\min_{\mathbf{w}\in\mathcal{W}, \boldsymbol{\nu}\in\mathbb{R}^n} F(\mathbf{w}, \boldsymbol{\nu}) := \frac{1}{n} \sum_{i=1}^{n} \mathbb{E}_{\zeta\sim\mathbb{P}_i}[e^{s_i(\mathbf{w};\zeta)-\nu_i} + \nu_i]. \tag{4}$$

Our contributions are summarized as follows:

- We design a novel geometry-aware stochastic algorithm that employs a stochastic proximal mirror descent (SPMD) method to update the dual variable, thereby mitigating the effect of an exponentially large smoothness parameter and stochastic variance. The proposed framework also establishes theoretical connections to, and provides insights into, existing methods based on mini-batch approximation and compositional optimization.

- We present a novel convergence analysis of the proposed method in the convex setting, yielding an improved convergence rate of $O(1/\sqrt{T})$. This addresses a long-standing challenge in the analysis of compo-

sitional optimization, where existing results typically exhibit worse complexities for convex compositional problems.

- We provide a rigorous comparison between convergence bounds obtained using SPMD updates and that using SGD updates for optimizing the dual variable, providing theoretical insights into the superiority of our method. Our analysis characterizes the intrinsic complexity of the problem through the second-order moment ratio of the random variable $e^{s_i(\cdot;\zeta)}$.

- We conduct extensive experiments on extreme classification with hundreds of thousands of class labels, partial AUC maximization, CLIP and distributionally robust optimization (DRO), demonstrating the effectiveness and robustness of our approach.

## 2. Related Works

While many ad hoc methods have been proposed for specific applications of CERM, we focus on reviewing studies that examine the design and analysis of optimization algorithms.

**Mini-batch Approximation.** The idea of this approach is to simply approximate the Log-E-Exp function by using a mini-batch of samples to approximate the inner expectation. Since this approach yields a gradient estimator that is biased, we refer to it as **biased SGD (BSGD)** following (Hu et al., 2020). This approach has been widely used for optimizing contrastive losses (Chen et al., 2020; Radford et al., 2021). Yuan et al. (2022) analyzed the convergence of this approach for optimizing a contrastive loss and showed that it has a large optimization error when the batch size is small. Levy et al. (2020) applied this idea to DRO problems. Their result also indicates that the large mini-batch approach for finding an $\epsilon$-optimal solution to the Log-E-Exp function requires a sample complexity of $O(1/\epsilon^3)$ with a large batch size of $O(1/\epsilon)$ for convex problems. We will show that BSGD can be recovered from our algorithmic framework by using a step size of infinity for the dual variable, which explains its limitation from another perspective.

**Solving the min-min formulation.** The equivalent minimization formulation of Log-E-Exp function in (4) has been known for decades, dating back to the 1980s, where it was introduced as a special case of the optimized certainty equivalent in mathematical economics (Ben-Tal & Teboulle, 1986). A straightforward approach is to apply SGD to the min-min problem (4), e.g., updating $\boldsymbol{\nu}$ first by a stochastic coordinate descent step and then updating $\mathbf{w}$ by a SGD step, which is referred to as **alternating SGD (ASGD)**.

Fagan & Iyengar (2018) have noted numerical instability issues when applying the SGD steps to the min–min formulation. To address these issues, they proposed an implicit

*Table 1.* Comparison of stochastic compositional optimization methods for optimizing convex compositional entropic risks (1). $f(\cdot) = \log(\cdot)$ denotes the log function, and $g(\mathbf{w}) = \mathbb{E}_\zeta e^{s(\mathbf{w};\zeta)}$. "Single-loop" means no inner loop is required. $\epsilon$ is the accuracy level of objective gap. The $n = 1$ results extend naturally to finite $n$ by scaling the mini-batch sizes and sample complexity by a factor of $n$.

| Method | $n$ | Mini-batch size | Iteration complexity | Sample complexity | Smoothness | Single-loop |
|---|---|---|---|---|---|---|
| BSGD (Hu et al., 2020) | $\infty$ | $\mathcal{O}(\epsilon^{-2})$ | $\mathcal{O}(\epsilon^{-2})$ | $\mathcal{O}(\epsilon^{-4})$ | ✗ | ✓ |
| BSGD (Hu et al., 2020) | $\infty$ | $\mathcal{O}(\epsilon^{-1})$ | $\mathcal{O}(\epsilon^{-2})$ | $\mathcal{O}(\epsilon^{-3})$ | $f$ | ✓ |
| BSGD (Levy et al., 2020) | 1 | $\mathcal{O}(\epsilon^{-1})$ | $\mathcal{O}(\epsilon^{-2})$ | $\mathcal{O}(\epsilon^{-3})$ | ✗ | ✓ |
| SCGD (Wang et al., 2017) | 1 | $\mathcal{O}(1)$ | $\mathcal{O}(\epsilon^{-4})$ | $\mathcal{O}(\epsilon^{-4})$ | $f$ | ✓ |
| SCGD (Wang et al., 2017) | 1 | $\mathcal{O}(1)$ | $\mathcal{O}(\epsilon^{-3.5})$ | $\mathcal{O}(\epsilon^{-3.5})$ | $f, g$ | ✓ |
| SOX (Wang & Yang, 2022) | finite | $\mathcal{O}(1)$ | $\mathcal{O}(n\epsilon^{-3})$ | $\mathcal{O}(n\epsilon^{-3})$ | $f, g$ | ✗ |
| MSVR (Jiang et al., 2022) | finite | $\mathcal{O}(1)$ | $\mathcal{O}(n\epsilon^{-2})$ | $\mathcal{O}(n\epsilon^{-2})$ | $f, g$ | ✗ |
| **SCENT (ours)** | finite | $\mathcal{O}(1)$ | $\mathcal{O}(n\epsilon^{-2})$ | $\mathcal{O}(n\epsilon^{-2})$ | ✗ | ✓ |

SGD method for XC that employs a joint proximal mapping of a stochastic estimator of the min-min objective to update both $\mathbf{w}$ and $\boldsymbol{\nu}$. There are three key differences between their approach and ours. First, their method is proposed specifically for XC with a linear model. Second, their method applies a joint proximal mapping over both the primal and dual variables, whereas our method employs a proximal mapping only for the dual variable. Third, they define the proximal mapping using the Euclidean distance. As a consequence, their method requires an additional solver to compute the proximal mapping, making it more difficult to implement in practice and incurring a higher per-iteration computational cost of $O\big(B^2(B + m)\log(1/\epsilon) + Bmd\big)$, where $\epsilon \ll 1$ is the accuracy for solving the proximal mapping, $B$ is the number of sampled data points, $m$ is the number of sampled classes, and $d$ is the dimensionality of $\mathbf{w}$. In contrast, our method has simple updates for both $\mathbf{w}$ and $\boldsymbol{\nu}$, whose cost dominated by $O(Bmd)$ for computing the logits. To reduce the computation overhead, they proposed another method named U-max, which shifts to the BSGD update whenever the updated dual variables cause a numerical issue.

**Compositional optimization techniques.** A useful technique for tackling the Log-E-Exp function is to cast it as an instance of compositional objective $f(g(\mathbf{w}))$, where $f(\cdot) = \log(\cdot)$ and $g(\mathbf{w}) = \mathbb{E}_\zeta[e^{s(\mathbf{w};\zeta)}]$ is the inner function. As a result, compositional optimization techniques can be employed such as stochastic compositional gradient descent (SCGD) (Wang et al., 2017). The key idea of SCGD is to approximate the inner function $g(\mathbf{w}) = \mathbb{E}_\zeta e^{s(\mathbf{w};\zeta)}$ by a moving-average estimator $u$ and compute a gradient estimator by $\nabla f(u)\nabla e^{s(\mathbf{w};\zeta')}$. (Qi et al., 2021; Li et al., 2021) were among the first works to leverage compositional optimization techniques for minimizing Log-E-Exp functions. Later, it was further analyzed in (Qi et al., 2023b;a) for optimizing the KL-regularized or constrained DRO problems. Wang & Yang (2022) extended this idea to solving a family of compositional optimization problems known as FCCO that covers CERM as a special case. Their algorithm, termed

SOX, maintains a moving-average estimator $u_i$ for each $i$ and updates them in a coordinate-wise manner. Later, this idea was applied to optimizing a variety of losses, including global contrastive losses (Yuan et al., 2022; Qiu et al., 2023; Wei et al., 2024), listwise cross-entropy loss (Qiu et al., 2022), and one-way partial AUC loss (Zhu et al., 2022).

While these methods are effective in practice, existing convergence analyses for convex problems suffer from (i) worse rates than $O(1/\sqrt{T})$ (Wang et al., 2017; Wang & Yang, 2022), (ii) requiring the convexity of the outer function $f$ to achieve an $O(1/\sqrt{T})$ rate (Wang & Yang, 2025; Zhang & Lan, 2020), and (iii) requiring a double-loop algorithm design (Wang & Yang, 2022; Jiang et al., 2022). Moreover, these works rely on coarse-grained analyses that assume Lipschitz continuity and smoothness of the exponential functions, thereby failing to capture the fundamental complexity of the problem. This work brings new insights into compositional optimization techniques for optimizing Log-E-Exp functions in our geometry-aware algorithmic framework. A comparison of these works and our method is shown in Table 1.

**Other methods.** Other techniques have been explored for tackling the complexity of the expensive normalization in the softmax function corresponding to the summation over $k$ in (2). For example, the noise contrastive estimation (NCE) technique addresses the expensive log-normalization by transforming the problem into a binary classification that contrasts the real data from data drawn from a noise distribution (Gutmann & Hyvärinen, 2010). However, the noise distributions could have a dramatic impact on the convergence speed (Liu et al., 2022; Jiang et al., 2023). Other approaches consider different sampling strategies to approximate the normalization term in softmax, e.g., incorporating hard negative mining strategies (Dahiya et al., 2023; Xiong et al., 2021; Yang et al., 2020), and active classes selection (Song et al., 2020). Recently, Lin et al. (2025) prove that any sampled estimators of softmax must

be biased. Wei et al. (2026) have considered a neural approximation method to learn the normalizers based on the min-min formulation for CLIP training. Instead of optimizing $\boldsymbol{\nu} \in \mathbb{R}^n$, they express each $\nu_i$ as the output of a neural network depending on the input's representation. A recent work (Gladin et al., 2025) has proposed a softplus approximation of LogSumExp, which yields a min-min formulation similar to (4) except that $e^{s(\mathbf{w};\zeta)-\nu}$ is approximated by $\log(1 + \rho e^{s(\mathbf{w};\zeta)-\nu})/\rho$, with $\rho > 0$ being a hyperparameter. This is equivalent to applying a truncation to the exponential function $e^{s(\mathbf{w};\zeta)-\nu}$, where $\rho$ controls the trade-off of the approximation accuracy and curvature of the function. Unlike these methods, our approach performs exact optimization and does not rely on approximation schemes.

## 3. A Geometry-aware Algorithm and its Convergence Analysis

Our algorithm is designed for solving the equivalent min-min optimization problem (4). We first present our algorithm for $n = 1$, where $F(\mathbf{w}, \nu) := \mathbb{E}_{\zeta \sim \mathbb{P}}[e^{s(\mathbf{w};\zeta)-\nu} + \nu]$, and then extend it to the case $n \gg 1$, as the fundamental challenge lies at handling log-E-Exp function.

The key novelty of our design is a **geometry-aware algorithm**. Let us first discuss the motivation. One challenge for solving the min-min optimization problem is that the objective function $F(\mathbf{w}, \nu)$ could have exponentially large smoothness constant in terms of $\nu$, which we will formally analyze in Section 4.3. Hence, a vanilla gradient method that uses the first-order approximation of $F$ will inevitably be impacted by the large smoothness parameter.

To mitigate the adverse effects of a large smoothness parameter with respect to $\nu$, we resort to the classical approach of proximal mapping, which has been widely used to handle a non-smooth function in composite objectives consisting of a smooth loss and a non-smooth regularizer (Lan, 2020). This approach enables optimization algorithms to retain the favorable convergence properties of smooth optimization and often leads to faster convergence despite the presence of non-smooth terms. Analogously, even when a function is smooth but characterized by a very large smoothness parameter, applying the proximal mapping technique can effectively alleviate the negative impact of this large smoothness constant.

However, there is an important distinction from classical proximal methods, which typically rely on full access to the function of interest for computing the proximal mapping. In our setting, we cannot directly apply the proximal mapping of $F(\mathbf{w}, \nu)$ as we only have access to a stochastic estimator:

$$\Phi(\mathbf{w}, \nu; \zeta) = e^{s(\mathbf{w};\zeta)-\nu} + \nu,$$

with $\zeta \sim \mathbb{P}$. As a result, it becomes necessary to explicitly

---

**Algorithm 1** The SCENT Algorithm for Solving CERM

1: Initialize $\mathbf{w}_1, \boldsymbol{\nu}_0$, step sizes $\eta_t$ and $\alpha_t$, $\varphi(\nu) = e^{-\nu}$.
2: **for** $t = 1 \ldots, T - 1$ **do**
3:     Sample $\mathcal{B}_t \subset \{1, \ldots, n\}$ with $|\mathcal{B}_t| = B$
4:     **for** each $i \in \mathcal{B}_t$ **do**
5:        Update $\nu_{i,t} = \arg\min_\nu e^{s_i(\mathbf{w}_t;\zeta_{i,t})-\nu} + \nu + \frac{1}{\alpha_t}D_\varphi(\nu, \nu_{i,t-1})$
6:     **end for**
7:     Compute the gradient estimator by $\mathbf{z}_t = \frac{1}{B}\sum_{i\in\mathcal{B}_t} e^{s_i(\mathbf{w}_t;\zeta'_{i,t})-\nu_{i,t}}\nabla s_i(\mathbf{w}_t;\zeta'_{i,t})$
8:     Update $\mathbf{w}_{t+1} = \Pi_{\mathcal{W}}[\mathbf{w}_t - \eta_t \mathbf{z}_t]$ (use momentum-based or Adam-based update in practice)
9: **end for**

---

account for the noise introduced by this stochastic approximation. To this end, we introduce a Bregman divergence $D_\varphi(\cdot, \cdot)$ and update $\nu_t$ according to the following scheme:

$$\nu_t = \arg\min_\nu \Phi(\mathbf{w}_t, \nu; \zeta_t) + \frac{D_\varphi(\nu, \nu_{t-1})}{\alpha_t}, \quad (5)$$

where $\zeta_t \sim \mathbb{P}$ is a random sample and $\alpha_t > 0$ is the step size. We refer to the update as **stochastic proximal mirror descent (SPMD)** update. To respect the geometry of the stochastic objective $\Phi(\mathbf{w}_t, \nu; \zeta_t)$, we construct a tailored Bregman divergence induced by $\varphi(\nu) = e^{-\nu}$, namely,

$$D_\varphi(\nu, \nu_{t-1}) = e^{-\nu} - e^{-\nu_{t-1}} + e^{-\nu_{t-1}}(\nu - \nu_{t-1}). \quad (6)$$

An additional advantage of this choice is that it admits a closed-form update for $\nu_t$, as formalized in the following lemma, whose proof is presented in Appendix B.1.

**Lemma 3.1.** *The update of $\nu_t$ defined in (5) with a Bregman divergence defined in (6) satisfies*

$$e^{\nu_t} = \frac{1}{1 + \alpha_t e^{\nu_{t-1}}}e^{\nu_{t-1}} + \frac{\alpha_t e^{\nu_{t-1}}}{1 + \alpha_t e^{\nu_{t-1}}}e^{s(\mathbf{w}_t;\zeta_t)}. \quad (7)$$

From (7), the update of $\nu_t$ can be reliably implemented by:

$$\nu_t = \nu_{t-1} + \log(1 + \alpha_t e^{s(\mathbf{w}_t;\zeta_t)}) - \log(1 + \alpha_t e^{\nu_{t-1}}).$$

Due to the presence of the logarithmic function, the numerical overflow can be effectively avoided in implementation.

With $\nu_t$, we update $\mathbf{w}_{t+1}$ by:

$$\begin{aligned}\mathbf{z}_t &= e^{s(\mathbf{w}_t;\zeta'_t)-\nu_t}\nabla s(\mathbf{w}_t; \zeta'_t), \\ \mathbf{w}_{t+1} &= \Pi_{\mathcal{W}}[\mathbf{w}_t - \eta_t \mathbf{z}_t],\end{aligned} \quad (8)$$

where $\zeta'_t \sim \mathbb{P}$ is a random sample independent from $\zeta_t$, and $\Pi_{\mathcal{W}}[\cdot]$ is the Euclidean projection onto $\mathcal{W}$.

Next, we extend this idea to the general case when $n \gg 1$ in (4). In this case, the problem poses an additional challenge: when $n$ is large, updating all components of $\boldsymbol{\nu}$ becomes prohibitive, as it would require processing the entire dataset. To tackle this challenge, we consider the stochastic

block coordinate update. Let

$$\Phi_i(\mathbf{w}, \nu_i; \zeta) = e^{s_i(\mathbf{w}; \zeta) - \nu_i} + \nu_i.$$

At iteration $t$, we randomly choose $B$ samples $\mathcal{B}_t \subset [n]$. We update $\nu_{i,t}$ similar to (5) if $i \in \mathcal{B}_t$, otherwise keep it intact:

$$\nu_{i,t} = \begin{cases} \arg\min_\nu \Phi_i(\mathbf{w}_t, \nu; \zeta_{i,t}) + \frac{D_\varphi(\nu, \nu_{i,t-1})}{\alpha_t} & i \in \mathcal{B}_t \\ \nu_{i,t-1} & i \notin \mathcal{B}_t \end{cases}$$
$$(9)$$

where $\zeta_{i,t} \sim \mathbb{P}_i$. Then we compute the gradient estimator with respect to $\mathbf{w}_t$ and update it by

$$\mathbf{z}_t = \frac{1}{|\mathcal{B}_t|} \sum_{i \in \mathcal{B}_t} e^{s_i(\mathbf{w}_t; \zeta'_{i,t}) - \nu_{i,t}} \nabla s_i(\mathbf{w}_t; \zeta'_{i,t}),$$
$$\mathbf{w}_{t+1} = \Pi_\mathcal{W}[\mathbf{w}_t - \eta_t \mathbf{z}_t],$$
$$(10)$$

where $\zeta'_{i,t} \sim \mathbb{P}_i$ are samples independent from $\zeta_{i,t}$. We present the full algorithm in Algorithm 1, which is referred to as SCENT (short for **S**tochastic optimization of **C**ompositional **ENT**ropic risk). We give two remarks about the use of the algorithm in practice. First, a momentum-based or Adam-based update for $\mathbf{w}$ can be incorporated to further enhance performance, depending on applications. Second, for practical simplicity, we can use the same random samples $\zeta'_{i,t} = \zeta_{i,t}$ in the update of $\nu_{i,t}$ and $\mathbf{w}_{t+1}$. For the purpose of theoretical analysis, we restrict our attention to the version in Algorithm 1.

In fact, the algorithmic framework in Algorithm 1 provides a unified perspective for understanding both BSGD and compositional optimization techniques. We present detailed derivation in Appendix A and summarize our findings here. First, BSGD can be recovered as a special case of our framework by setting $\alpha_t = \infty$. Due to this choice, BSGD lacks the mechanism to account for any noise in the stochastic estimator for updating $\nu_t$, which is the primary reason why it fails to guarantee convergence when the batch size for approximating the inner function is small. Second, compositional optimization algorithms such as SCGD for optimizing the Log-E-Exp function ($n = 1$) corresponds to a particular setting of $\alpha_t = \gamma'_t e^{-\nu_t}$ for some $\gamma'_t > 0$ in the framework of SCENT. This perspective allows us to establish an improved complexity of $O(1/\epsilon^2)$ of SCGD for optimizing the Log-E-Exp function. The SOX algorithm for solving CERM corresponds to the proposed framework with a coordinate-wise step size $\alpha_{i,t} = \gamma'_t e^{-\nu_{i,t-1}}$ for some $\gamma'_t > 0$ in the SPMD step for updating $\nu_{i,t}$. It turns out that this choice may slow down the convergence as observed in our experiments.

## 3.1. Convergence Analysis

We define the following notations:

$$F_i(\mathbf{w}, \nu_i) = \mathbb{E}_{\zeta \sim \mathbb{P}_i}[\Phi_i(\mathbf{w}, \nu_i; \zeta)],$$
$$D_\varphi(\boldsymbol{\nu}_*, \boldsymbol{\nu}) = \sum_{i=1}^n D_\varphi(\nu_{i,*}, \nu_i), \qquad (11)$$
$$(\mathbf{w}_*, \boldsymbol{\nu}_*) = \arg\min_{\mathbf{w}, \boldsymbol{\nu}} F(\mathbf{w}, \boldsymbol{\nu}).$$

And we let $\nabla_\mathbf{w} F(\mathbf{w}, \boldsymbol{\nu})$ and $\nabla_{\boldsymbol{\nu}} F(\mathbf{w}, \boldsymbol{\nu})$ denote the partial gradient in terms of $\mathbf{w}, \boldsymbol{\nu}$, respectively. Since $\boldsymbol{\nu}_t$ is updated using the stochastic block coordinate method that is dependent on random mini-batch $\mathcal{B}_t$, expectation of $\mathbf{z}_t$ in (10) is not the full gradient $\nabla_\mathbf{w} F(\mathbf{w}_t, \boldsymbol{\nu}_t)$, i.e., $\mathbb{E}_{\mathcal{B}_t, \zeta'_t}[\mathbf{z}_t] \neq \nabla_\mathbf{w} F(\mathbf{w}_t, \boldsymbol{\nu}_t)$. To ease analysis, we introduce a virtual sequence $\bar{\boldsymbol{\nu}}_t$ that updates all coordinates of $\boldsymbol{\nu}_{t-1}$:

$$\bar{\nu}_{i,t} = \arg\min_\nu \Phi_i(\mathbf{w}_t, \nu; \zeta_{i,t}) + \frac{D_\varphi(\nu, \nu_{i,t-1})}{\alpha_t}, \forall i$$

Note that $\bar{\boldsymbol{\nu}}_t$ is independent of $\mathcal{B}_t$: $\mathbb{E}_{\mathcal{B}_t, \zeta'_t}[\mathbf{z}_t] = \nabla_\mathbf{w} F(\mathbf{w}_t, \bar{\boldsymbol{\nu}}_t)$.

We first outline the high-level idea of the convergence analysis under the convexity of $s_i(\mathbf{w}; \zeta)$. First, we will prove the joint convexity of $F(\mathbf{w}, \boldsymbol{\nu})$ in terms of both $\mathbf{w}$ and $\boldsymbol{\nu}$. Then we will prove the convergence in terms of the joint objective gap $F(\hat{\mathbf{w}}_T, \hat{\boldsymbol{\nu}}_T) - F(\mathbf{w}_*, \boldsymbol{\nu}_*)$ for some $\hat{\mathbf{w}}_T, \hat{\boldsymbol{\nu}}_T$, which implies the convergence of the primal objective gap $F_{\text{CERM}}(\hat{\mathbf{w}}_T) - F_{\text{CERM}}(\mathbf{w}_*) \leq F(\hat{\mathbf{w}}_T, \hat{\boldsymbol{\nu}}_T) - F(\mathbf{w}_*, \boldsymbol{\nu}_*)$.

Since $\mathbf{w}_t, \bar{\boldsymbol{\nu}}_t$ are updated using different schemes, we need to analyze the update of $\mathbf{w}_t$ and $\bar{\boldsymbol{\nu}}_t$ separately, and then merge them to obtain the joint objective gap. To this end, we will first establish a bound for linearized regrets $\mathbb{E}[\nabla_\mathbf{w} F(\mathbf{w}_t, \bar{\boldsymbol{\nu}}_t)^\top(\mathbf{w}_t - \mathbf{w}_*)]$ and $\mathbb{E}[\nabla_{\boldsymbol{\nu}} F(\mathbf{w}_t, \bar{\boldsymbol{\nu}}_t)^\top(\bar{\boldsymbol{\nu}}_t - \boldsymbol{\nu}_*)]$ in terms of $\mathbf{w}_t$ and $\bar{\boldsymbol{\nu}}_t$, respectively. The analysis for the former is mostly straightforward following existing analysis of the projected SGD update. The challenge lies at bounding $\mathbb{E}[\nabla_{\boldsymbol{\nu}} F(\mathbf{w}_t, \bar{\boldsymbol{\nu}}_t)^\top(\bar{\boldsymbol{\nu}}_t - \boldsymbol{\nu}_*)]$ for the SPMD update, which is the major novelty of the analysis.

Next, we present the key results for bounding the two linearized regrets and a final convergence bound for SCENT, with all proofs deferred to Appendix C. To this end, we first define the variance terms due to the stochastic estimators used for updating $\mathbf{w}_{t+1}$ and $\boldsymbol{\nu}_t$:

$$\sigma_{i,t}^2 := \mathbb{E}_{\zeta'_{i,t} \sim \mathbb{P}_i}[\|e^{s_i(\mathbf{w}_t; \zeta'_{i,t}) - \nu_{i,t}} \nabla s_i(\mathbf{w}_t; \zeta'_{i,t})\|_2^2],$$

$$\delta_{i,t}^2 := \mathbb{E}_{\zeta_{i,t} \sim \mathbb{P}_i}[e^{-\nu_{i,t-1}}|e^{s_i(\mathbf{w}_t; \zeta_{i,t})} - \mathbb{E}_{\zeta_i \sim \mathbb{P}_i}[e^{s_i(\mathbf{w}_t; \zeta_i)}]|^2].$$

And let $\sigma_t^2, \delta_t^2$ be the average of $\sigma_{i,t}^2, \delta_{i,t}^2$, respectively:

$$\sigma_t^2 = \frac{1}{n} \sum_{i=1}^n \sigma_{i,t}^2, \quad \delta_t^2 = \frac{1}{n} \sum_{i=1}^n \delta_{i,t}^2.$$

We note that $\sigma_t^2$ is the variance of stochastic gradient for updating $\mathbf{w}_t$ and $\delta_t^2$ is the variance proxy of stochastic noise for updating $\nu_t$.

We impose the following assumption for the analysis.

**Assumption 3.2.** We assume that: (i) $s_i(\cdot; \zeta)$ is convex and differentiable, $\forall \zeta$; (ii) $s_i(\mathbf{w}; \zeta) \in [c_0, c_1], \forall \mathbf{w} \in \mathcal{W}, \zeta$; (iii) there exists $G$ such that $\mathbb{E}_\zeta[\|\nabla s_i(\mathbf{w}_t; \zeta)\|_2^2] \leq G^2, \forall t$.

We first show that under Assumption 3.2, the SPMD update guarantees that $\delta_t^2, \sigma_t^2$ are finite. The key is to show that $\nu_{i,t}$ is always bounded in $[c_0, c_1]$.

**Lemma 3.3.** *For the SPMD update (9), if $\boldsymbol{\nu}_0 \in [c_0, c_1]^n$ it is guaranteed that $\nu_{i,t} \in [c_0, c_1], \forall i \in [n], t \geq 1$. Moreover, $\delta_{i,t}$ and $\sigma_{i,t}$ are finite, $\forall i \in [n], t \geq 1$.*

This is one advantage of the SPMD update over the SGD update for $\nu_t$, since the latter either does not guarantee this boundedness or requires an explicit projection onto $[c_0, c_1]$.

The following lemma establishes the bound for the linearized primal regret $\mathbb{E}[\eta_t \nabla_\mathbf{w} F(\mathbf{w}_t, \bar{\boldsymbol{\nu}}_t)^\top (\mathbf{w}_t - \mathbf{w}_*)]$.

**Lemma 3.4.** *Under Assumption 3.2, we have*

$$\mathbb{E}[\eta_t \nabla_\mathbf{w} F(\mathbf{w}_t, \bar{\boldsymbol{\nu}}_t)^\top (\mathbf{w}_t - \mathbf{w}_*)]$$
$$\leq \mathbb{E}\left[\frac{1}{2}\|\mathbf{w}_t - \mathbf{w}_*\|_2^2 - \frac{1}{2}\|\mathbf{w}_{t+1} - \mathbf{w}_*\|_2^2\right] + \frac{\eta_t^2 \sigma_t^2}{2}.$$

The following lemma is our key result for bounding the linearized dual regret $\mathbb{E}[\nabla_\nu F(\mathbf{w}_t, \bar{\boldsymbol{\nu}}_t)^\top (\bar{\boldsymbol{\nu}}_t - \boldsymbol{\nu}_*)]$.

**Lemma 3.5.** *Under Assumption 3.2 (ii) and setting $\alpha_t \leq \min_i \rho e^{-\nu_{i,t-1}}$ for some constant $\rho > 0$, we have*

$$\mathbb{E}[\alpha_t \nabla_\nu F(\mathbf{w}_t, \bar{\boldsymbol{\nu}}_t)^\top (\bar{\boldsymbol{\nu}}_t - \boldsymbol{\nu}_*)]$$
$$= \mathbb{E}\left[\alpha_t \cdot \frac{1}{n} \sum_{i=1}^n \nabla_\nu F_i(\mathbf{w}_t, \bar{\nu}_{i,t})^\top (\bar{\nu}_{i,t} - \nu_{i,*})\right]$$
$$\leq \frac{1}{B} \cdot \mathbb{E}[D_\varphi(\boldsymbol{\nu}_*, \boldsymbol{\nu}_{t-1}) - D_\varphi(\boldsymbol{\nu}_*, \boldsymbol{\nu}_t)] + C\alpha_t^2 \delta_t^2.$$

*where $C = (1 + \rho)(1 + c_1 - c_0)$.*

We highlight the challenge in proving the above bound. Due to the SPMD update of $\bar{\boldsymbol{\nu}}$, it is easy to establish:

$$\alpha_t \nabla_\nu \Phi_i(\mathbf{w}_t, \bar{\nu}_{i,t}; \zeta_{i,t})(\bar{\nu}_{i,t} - \nu_{i,*})$$
$$\leq D_\varphi(\nu_{i,*}, \nu_{i,t-1}) - D_\varphi(\nu_{i,*}, \bar{\nu}_{i,t}) - D_\varphi(\bar{\nu}_{i,t}, \nu_{i,t-1}).$$

In order to bound $\mathbb{E}[\alpha_t \nabla_\nu F_i(\mathbf{w}_t, \nu_{i,t})(\bar{\nu}_{i,t} - \nu_{i,*})]$, we need to bound the difference

$$\mathbb{E}[\alpha_t(\nabla_\nu F_i(\mathbf{w}_t, \bar{\nu}_{i,t}) - \nabla_\nu \Phi_i(\mathbf{w}_t, \bar{\nu}_{i,t}; \zeta_{i,t}))(\bar{\nu}_{i,t} - \nu_{i,*})].$$

Although $\nabla_\nu \Phi_i(\mathbf{w}_t, \bar{\nu}_{i,t}; \zeta_{i,t})$ is an unbiased estimator of $\nabla_\nu F_i(\mathbf{w}_t, \bar{\nu}_{i,t})$, the above expectation is not zero since $\bar{\nu}_{i,t}$ depends on the random variable $\zeta_{i,t}$. To address this challenge, we develop a novel analysis to prove the above lemma. We also remark that the condition $\alpha_t \leq \min_i \rho e^{-\nu_{i,t-1}}$ is useful to mitigate the impact of the variance in $\Phi_i(\mathbf{w}_t, \bar{\nu}_{i,t}; \zeta_{i,t})$. Finally, we present the convergence bound of SCENT.

**Theorem 3.6.** *Under Assumption 3.2, let $\eta_t = \eta \alpha_t$ for some constant $\eta > 0$, and $\alpha_t = \frac{\alpha}{\sqrt{T}} < \rho \min_i e^{-v_{i,t-1}}$ for some*

*constant $\alpha, \rho > 0$. Then SCENT guarantees that*

$$\mathbb{E}[F_{\text{CERM}}(\bar{\mathbf{w}}_T) - F_{\text{CERM}}(\mathbf{w}_*)]$$
$$\leq \frac{1}{2\eta\alpha\sqrt{T}}\|\mathbf{w}_1 - \mathbf{w}_*\|_2^2 + \frac{D_\varphi(\boldsymbol{\nu}_*, \boldsymbol{\nu}_0)}{\alpha B \sqrt{T}} + \frac{\alpha V}{\sqrt{T}}, \quad (12)$$

*where $\bar{\mathbf{w}}_T = \frac{\sum_{t=1}^T \mathbf{w}_t}{T}$, $V = \frac{\eta \sum_{t=1}^T \sigma_t^2}{2T} + \frac{C \sum_{t=1}^T \delta_t^2}{T}$.*

**Remark:** Since $V$ is finite, the above theorem implies a convergence rate of $O(1/\sqrt{T})$. In Corollary B.8, we show the same order of convergence rate for SCGD for optimizing the Log-E-exp function ($n = 1$), which corresponds to SCENT with $\alpha_t = \gamma_t' e^{-\nu_{t-1}}$ for some $\gamma_t'$. In contrast, existing analysis of SCGD for convex compositional optimization yields a worse complexity of $O(1/T^{1/4})$ (Wang et al., 2017). A key to our improved complexity is to use a single time-scale step sizes for $\mathbf{w}, \boldsymbol{\nu}$, i.e., $\eta_t \propto \alpha_t$, while Wang et al. (2017) use two time-scale step sizes.

# 4. Analysis of the Convergence Bound

A caveat of the convergence bound in Theorem 3.6 is its dependence on the quantity $V$, which averages the variance terms $\delta_t^2$ and $\sigma_t^2$ over all iterations. Traditional convergence analysis of stochastic optimization usually assumes that the variance terms at each iteration are bounded. However, they become more intricate for the considered problem because of the joint update of $\mathbf{w}_t, \boldsymbol{\nu}_t$ and the involved exponential function. Although Lemma 3.3 guarantees that these variance terms are bounded, it naturally raises the question of whether they could grow exponentially as in worst-case analysis, or more importantly, whether the resulting convergence bound may involve exponentially large constants that cannot be controlled. A further fundamental question concerns how to rigorously quantify the advantages of the SPMD update over the standard SGD update for $\nu_t$.

We address these questions in this section. First, we establish upper bounds for $\delta_t^2$ and $\sigma_t^2$, demonstrating that these quantities remain well controlled as the algorithm converges. Second, we fix $\mathbf{w}$ and analyze the SPMD update for the dual optimization problem. In particular, we derive an upper bound of SPMD that is characterized by a key quantity that captures the intrinsic complexity of the problem.

## 4.1. Analysis of the Variance Terms

For simplicity of exposition, we focus on the case of $n = 1$ with $F(\mathbf{w}, \nu) = \mathbb{E}_\zeta[e^{s(\mathbf{w}; \zeta) - \nu} + \nu]$. We define:

$$z(\mathbf{w}; \zeta) = e^{s(\mathbf{w}; \zeta)}, \ \mu(\mathbf{w}) = \log \mathbb{E}_\zeta e^{s(\mathbf{w}; \zeta)},$$
$$m_t = \mathbb{E}_\zeta e^{s(\mathbf{w}_t; \zeta)}, \ \mu_t = \mu(\mathbf{w}_t) = \log m_t.$$

The proofs of the results in this section are presented in Appendix D. For the analysis in this section, we make two assumptions regarding $\mathbf{w}$ only.

**Assumption 4.1.** We assume that there exist constants $\kappa, \sigma'$ such that (i) $\mathbb{E}[z^2(\mathbf{w}; \zeta)] / (\mathbb{E}[z(\mathbf{w}; \zeta)])^2 \leq \kappa, \forall \mathbf{w}$; (ii) $\mathbb{E}[\|e^{s(\mathbf{w}_t; \zeta') - \mu_t} \nabla s(\mathbf{w}_t; \zeta')\|_2^2] \leq \sigma'^2, \forall t$;

**Remark:** These assumptions are necessary to quantify the variance terms. As shown in Appendix E, the dependence on $\kappa$ is unavoidable for a family of algorithms. The second assumption is the standard bounded stochastic gradient assumption of the objective $F_{\text{CERM}}(\mathbf{w})$.

**Lemma 4.2.** *Under Assumption 4.1, we have*

$$\sigma_t^2 \leq 4(\sigma')^2 \big(F(\mathbf{w}_t, \nu_t) - F(\mathbf{w}_*, \nu_*) + 1\big)^2,$$
$$\delta_t^2 \leq 2(\kappa - 1) m_t \Big(F(\mathbf{w}_t, \nu_{t-1}) - F(\mathbf{w}_*, \nu_*) + 1\Big).$$

**Remark:** The first result indicates that when $F(\mathbf{w}_t, \nu_t) - F(\mathbf{w}_*, \nu_*) \to 0$, the variance term $\sigma_t^2$ caused by the stochastic update of $\mathbf{w}_t$ will be dominated by $O((\sigma')^2)$. The second result shows that when $F(\mathbf{w}_t, \nu_{t-1}) - F(\mathbf{w}_*, \nu_*) \to 0$, the variance term $\delta_t^2$ caused by the stochastic update of $\nu_t$ will be dominated by $2(\kappa - 1) m_t$. Large $m_t$ can be mitigated by choosing small $\alpha_t$. Indeed, if $s(\mathbf{w}_t; \zeta) > 0$ causes exponentially large $m_t$, it can be mitigated by exponentially small $D_\varphi(\nu_*, \nu_0) = e^{-\nu_*}(1 - e^{\nu_* - \nu_0} + e^{\nu_* - \nu_0}(\nu_* - \nu_*))$ with $\nu_0 \gg \nu_*$ through the choice of $\alpha$ in the bound (12). We will make this more explicit in the analysis presented in next subsection.

## 4.2. Analysis of SPMD for fixed w

In this subsection, we further simplify the setting in order to quantify the fundamental complexity of optimizing the dual variable $\nu$ with fixed $\mathbf{w}$. To this end, we consider the following problem:

$$\min_\nu F(\nu) := \mathbb{E}_\zeta e^{s(\zeta) - \nu} + \nu, \qquad (13)$$

where we omit $\mathbf{w}$ in $s(\zeta)$. We define

$$z := e^{s(\zeta)}, \quad m := \mathbb{E}[z] > 0, \quad \kappa := \frac{\mathbb{E}[z]^2}{(\mathbb{E}[z])^2},$$

where $\kappa$, the second-order moment ratio, is key to quantify the fundamental complexity of the problem. Larger $\kappa$ indicates heavier tails or higher variability relative to the mean.

It is easy to derive that $\nu_* = \arg\min_\nu F(\nu) = \log m$. Nevertheless, we consider a black-box oracle model for the algorithm, where the underlying distribution of $z$ is unknown and for any query $\nu$ the oracle returns

$$\Phi(\nu; \zeta) = z e^{-\nu} + \nu, \quad g(\nu; \zeta) = \nabla \Phi(\nu; \zeta) = 1 - z e^{-\nu}.$$

In the theorem below, we present a convergence result of the SPMD method defined by:

$$\nu_t = \arg\min_\nu \Phi(\nu; \zeta_t) + \frac{D_\varphi(\nu, \nu_{t-1})}{\alpha_t}. \qquad (14)$$

**Theorem 4.3.** *Suppose $s(\zeta) \in [c_0, c_1]$. By setting $\alpha_t =$*

$\alpha = \sqrt{\frac{D_\varphi(\nu_*, \nu_0) m}{2 C T \text{Var}(z)}} \leq \min(\frac{m}{4 C \text{Var}(z)}, \rho e^{-\nu_{t-1}})$ *for some $\rho > 0$ and sufficiently large $T$, SPMD guarantees that*

$$\frac{1}{T} \sum_{t=1}^T \mathbb{E}[F(\nu_t) - F(\nu_*)] \leq \qquad (15)$$

$$4\sqrt{2} \sqrt{\frac{C(\kappa - 1)(1 - r_0 + r_0 \log r_0)}{T}} + \frac{F(\nu_0) - F(\nu_*)}{T}.$$

*where $C = (1 + \rho)(1 + c_1 - c_0)$, and $r_0 := e^{\nu_* - \nu_0}$.*

**Remark:** When $\nu_0 \gg \nu_*$, then $1 - r_0 + r_0 \log r_0 = O(1)$, the dominating term is $O(\sqrt{\frac{\kappa}{T}})$. This upper bound characterizes the intrinsic complexity of SPMD, which depends on the second-order moment ratio $\kappa$. If $s(\zeta) \sim \mathcal{N}(\mu, \sigma^2)$, then $\kappa = e^{\sigma^2}$, which does not depend on the exponential of the mean $\mu$ but rather $e^{\sigma^2}$. In Appendix E, we prove a lower bound showing that the dependence on $\kappa$ is unavoidable for a family of algorithms under the black-box oracle model.

## 4.3. Comparison with a Convergence Bound of the SGD Update

Below, we present a standard convergence bound of SGD for optimizing $F(\nu)$. In order to control the variance, we consider projected SGD. Let $\Pi_{[c_0, c_1]}$ denote projection onto $[c_0, c_1]$. The projected SGD update is

$$\nu_{t+1} = \Pi_{[c_0, c_1]}\big(\nu_t - \alpha' g(\nu_t, \zeta_t)\big), \qquad (16)$$

where $\{\zeta_t\}_{t \geq 0}$ are i.i.d. copies of $\zeta$ and $\alpha' > 0$ is the step size. We quantify the smoothness on the bounded domain of the objective, which introduces an exponential constant.

**Lemma 4.4.** *On $[c_0, c_1]$, the function $F(\nu) = m e^{-\nu} + \nu$ is L-smooth with*

$$L = \sup_{\nu \in [c_0, c_1]} F''(\nu) = \sup_{\nu \in [c_0, c_1]} m e^{-\nu} = m e^{-c_0} = e^{\nu_* - c_0}.$$

**Theorem 4.5.** *By choosing the optimal $\alpha' = \frac{|\nu_0 - \nu_*| e^{c_0}}{\sqrt{2 T \text{Var}(z)}} \leq \frac{1}{L} = \frac{e^{c_0}}{m}$, SGD has a convergence upper bound:*

$$\frac{1}{T} \sum_{t=1}^T \mathbb{E}[F(\nu_t) - F(\nu_*)] \leq \sqrt{2} |\nu_0 - \nu_*| e^{\nu_* - c_0} \sqrt{\frac{\kappa - 1}{T}}.$$

**Remark:** The ratio of the convergence bound of SPMD to that of SGD is $\frac{1}{|\nu_0 - \nu_*| e^{\nu_* - c_0}}$. Notably, this ratio becomes exponentially small in regimes where $\nu_* \gg c_0$, highlighting the superior efficiency of SPMD. If $s(\zeta) \sim \mathcal{N}(\mu, \sigma^2)$, then $\nu_* = \log m = \mu + \sigma^2/2$, and the ratio is proportional to $e^{-\sigma^2/2} e^{c_0 - \mu}$, which decreases exponentially as $\sigma$ increases.

We also note that the term $e^{\nu_* - c_0} \sqrt{\kappa - 1}$ in the upper bound arises from the variance of the stochastic gradient $g(\nu_t, \zeta_t)$. Even if one replaces the Bregman divergence with the Euclidean distance in the proximal update—recovering the stochastic proximal point method (Chadha et al., 2022)—

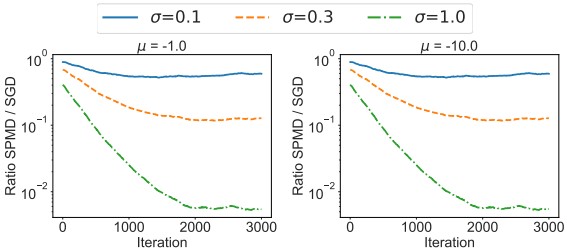

*Figure 1.* Ratio between the error of SPMD and that of SGD when trained on Gaussian noise with different means and variances.

this variance-dependent term persists in the upper bound. This justifies the use of the Bregman divergence in (14).

To justify the theoretical analysis, we compare SPMD and SGD in a controlled synthetic data setting where $s(\zeta) \sim \mathcal{N}(\mu, \sigma^2)$. We vary $\mu, \sigma$ and compare the convergence error of SPMD and SGD in Figure 1, where it clearly shows that the ratio between SPMD's convergence error to that of SGD decreases as $\sigma$ increases and is independent of $\mu$.

## 5. Experiments

In this section, we provide empirical justification of the effectiveness of our approach. Specifically, we compare our proposed method with multiple baselines on different tasks, including extreme classification (XC, Section 5.1) and partial AUC maximization (Section 5.2). We also conduct experiments on distributionally robust optimization and CLIP training, whose results are deferred to Appendix F due to space limit. For all experiments in this section, we run each method three times with different random seeds, and report the average performance with error bars. The explicit updates of SCENT for each task are presented in Appendix F.4.

### 5.1. Extreme Classification

**Datasets.** We consider the Glint360K dataset (An et al., 2021) and the TreeOfLife-10M dataset (Stevens et al., 2024): the former is a face dataset consisting of 17 million images from 360 thousand individuals (i.e., 360K classes), while the latter is a biology dataset of 10 million images from 160 thousand species. We use the low-dimensional features of the images to train the classifier. In particular, we leverage a ResNet-50 encoder (He et al., 2016) pretrained on Glint360K and a CLIP ViT-B/16 model (Dosovitskiy et al., 2021) pretrained on TreeOfLife-10M, released by the authors of the respective datasets, to process the images of the respective datasets into features. More details can be found in Appendix F.4.

**Baselines.** We compare our method with the following baselines: BSGD, ASGD for solving the same min-min formulation, SOX, the U-max method in Fagan & Iyengar (2018)

and ASGD for solving the softplus approximation (Gladin et al., 2025). For all the methods, we use a batch size of 128 and train the model for 50 epochs using the SGD optimizer for the model parameter. The details of hyperparameter tuning are presented in Appendix F.4. In Appendix F.1, we also include results using the momentum optimizer for the model parameter $\mathbf{w}$ with similar results as discussed below.

**Results.** We present the cross entropy loss value curves on the training data and validation data in Figure 2, from which we have the following observations. First, on all datasets, ASGD, U-max and ASGD (Softplus) perform similarly. Second, BSGD is better than ASGD on Glint360k data but is worse than ASGD on TreeOfLife-10M data. Last, SOX and SCENT are consistently better than all methods and SCENT performs better than SOX. This justifies our choice of the geometry-aware update of the dual variable.

### 5.2. Partial AUC Maximization

**Datasets.** We consider the binary classification task on imbalanced image datasets. Specifically, we use the CIFAR-10 and CIFAR-100 dataset (Krizhevsky, 2009) in our experiments. To make the datasets imbalanced, for both datasets, we take first half of classes as the negative class and last half of classes as the positive class. Then we construct an imbalanced version by randomly removing 80% samples from the positive class, which we use for training. The model we train is a ResNet18 (He et al., 2016). Similar to Zhu et al. (2022), we add a pretraining stage that optimizes the base model using the binary cross-entropy loss with the SGD optimizer, and then freeze the backbone and optimize the classifier layer by using different methods.

**Baselines.** We use the same baselines as previous subsection for comparison. For all the methods, we use a batch size of 64 and train the model for 60 epochs using the SGD optimizer. The details of hyperparameter tuning are presented in Appendix F.4. In Appendix F.1, we also include more results using the momentum optimizer for the model parameter $\mathbf{w}$ with similar conclusions as discussed below.

**Results.** We plot loss curves on the training data in Figure 3 for different $\tau$. Across different datasets and $\tau$ choices, we have the following observations. First, ASGD, U-max and ASGD (Softplus) do not perform well for this task, whose gap with BSGD are usually large. Second, SOX and SCENT enjoy the best results among all methods and SCENT is slightly better than SOX. This also justifies our choice of the geometry-aware update. From the results on XC and partial AUC maximization, we can conclude that SCENT yields the best performance.

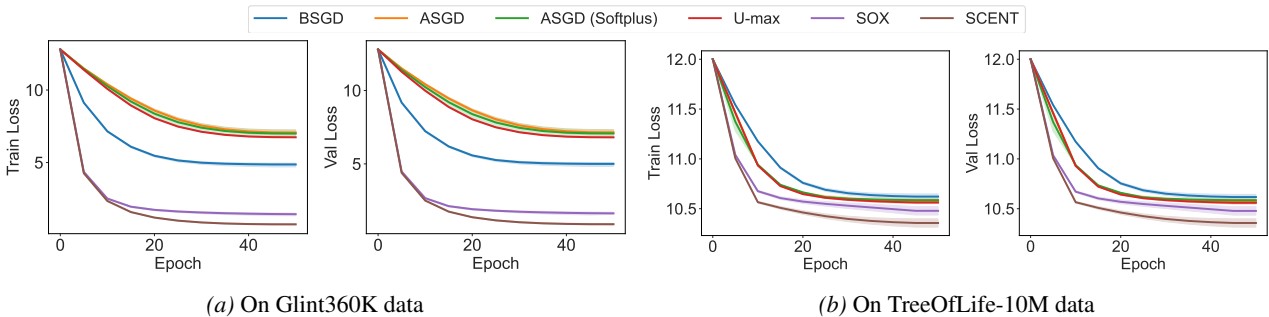

*(a)* On Glint360K data

*(b)* On TreeOfLife-10M data

*Figure 2.* (a): Cross-entropy loss curves of different methods on the training set (left) and validation set (right) of Glint360K. (b): Cross-entropy loss curves of different methods on the training set (left) and validation dataset (right) of TreeOfLife-10M.

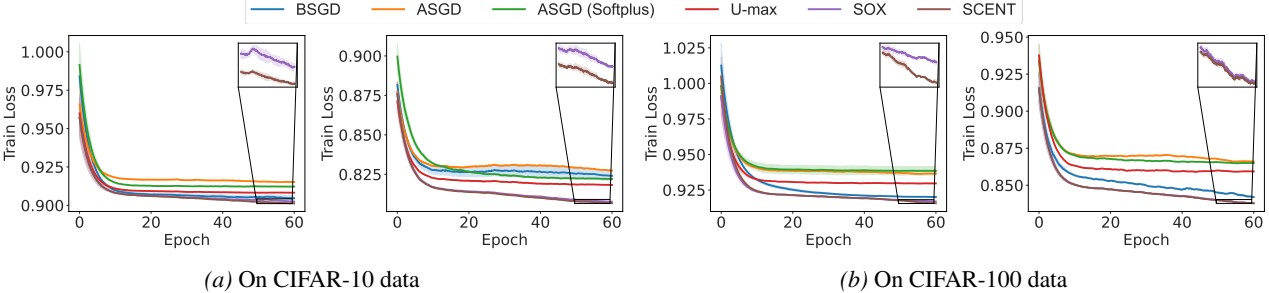

*(a)* On CIFAR-10 data

*(b)* On CIFAR-100 data

*Figure 3.* Training loss curves of different methods for partial AUC maximization. (a): on the dataset CIFAR-10 with $\tau = 0.05$ (left) and $\tau = 0.1$. (b): on the dataset CIFAR-100 with $\tau = 0.05$ (left) and $\tau = 0.1$.

## 6. Conclusion and Discussion

In this paper, we have studied the problem of efficiently optimizing the compositional entropic risk. Leveraging a min-min formulation of the risk, we proposed a novel geometry-aware stochastic proximal mirror descent (SPMD) update for the dual variable. Theoretically, we analyzed the convergence of the algorithm for convex problems, and we provide comparison between the SPMD update and SGD update. Empirically, we conducted extensive experiments on extreme classification, partial AUC maximization, contrastive learning and distributionally robust optimization to demonstrate the effectiveness of our algorithm.

In this work, we only consider theoretical analysis for the convex setting. It is worth exploring how to conduct the convergence analysis for non-convex setting to exhibit the potential advantages over existing methods, e.g., Wang & Yang (2022). Another limitation of this work is the inapplicability of SCENT to $n = \infty$ since it is not possible to maintain $\nu$ for all individual samples. One way to address this limitation is to follow Wei et al. (2026) by learning a parameterized neural network $\nu(\mathbf{w}', \mathbf{x}_i)$ to predict $\nu_i$.

## Acknowledgment

We thank the anonymous reviewers for their helpful comments. We are grateful to Egor Gladin for identifying a bug in our earlier implementation of ASGD (softplus) for partial AUC maximization. X. Wei, L. Zhou, and T. Yang were partially supported by NSF Award 2306572. C. Lin was supported in part by National Science and Technology Council of Taiwan grants NSTC-113-2222-E-002-005-MY3 and NSTC-114-2634-F-002-007. This work used GPU resources at TAMU ACES and NCSA Delta through allocation CIS230245 from the Advanced Cyberinfrastructure Coordination Ecosystem: Services & Support (ACCESS) program, which is supported by U.S. National Science Foundation grants #2138259, #2138286, #2138307, #2137603, and #2138296.

## Impact Statement

This paper presents work whose goal is to advance the field of Machine Learning. There are many potential societal consequences of our work, none which we feel must be specifically highlighted here.

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

# A. Details of BSGD/ASGD/SCGD and Connections with SCENT

In this section, we present details of existing methods for optimizing Log-E-Exp and CERM, and build the connection with the proposed algorithmic framework. For simplicity of exposition, we focus on the Log-E-Exp function, which corresponds to $n = 1$ in CERM:

$$\min_{\mathbf{w} \in \mathcal{W}} F_{\text{CERM}}(\mathbf{w}) := \log \left( \mathbb{E}_\zeta e^{s(\mathbf{w}; \zeta)} \right). \tag{17}$$

For the moment, we just take $\mathcal{W} = \mathbb{R}^d$. A naive idea one might consider is that, since the logarithm is a monotonic function, one could instead optimize $\mathbb{E}_\zeta e^{s(\mathbf{w}; \zeta)}$, to which standard stochastic optimization algorithms can be directly applied. This approach is ineffective, as it not only introduces numerical instability due to the exponential function, but also fails to extend to CERM settings with multiple components ($n > 1$).

The challenge of optimizing Log-E-Exp lies at computing the gradient:

$$\nabla F_{\text{CERM}}(\mathbf{w}) = \frac{1}{\mathbb{E}_\zeta[e^{s(\mathbf{w}; \zeta)}]} \mathbb{E}_\zeta[e^{s(\mathbf{w}; \zeta)} \nabla s(\mathbf{w}; \zeta)],$$

which is prohibitive due to expectations in both the numerator and the denominator. Next, we present several algorithms that have been considered in the literature.

## A.1. Biased SGD with Mini-batch Approximation.

A simple approach is to consider an approximation of Log-E-Exp using a mini-batch $\mathcal{C}$: $\log \left( \frac{1}{|\mathcal{C}|} \sum_{\zeta \in \mathcal{C}} e^{s(\mathbf{w}; \zeta)} \right)$. At the $t$-th iteration, $\mathbf{w}_t$ is updated by

$$\mathbf{w}_{t+1} = \mathbf{w}_t - \eta_t \sum_{\zeta \in \mathcal{C}_t} \frac{e^{s(\mathbf{w}_t; \zeta)}}{\sum_{\zeta' \in \mathcal{C}_t} e^{s(\mathbf{w}_t; \zeta')}} \nabla s(\mathbf{w}_t; \zeta). \tag{18}$$

**Limitation:** However, since the gradient estimator is a biased estimation of $\nabla F_{\text{CERM}}(\mathbf{w}_t)$, this method does not converge if the size of $\mathcal{C}_t$ is small, i.e., it requires a large batch size to ensure convergence of convex objective.

## A.2. Alternating SGD for Solving the Dual Reformulation.

One way to avoid the biased gradient estimation is to cast the Log-E-Exp problem into an equivalent minimization form:

$$\log \left( \mathbb{E}_\zeta e^{s(\mathbf{w}; \zeta)} \right) = \min_\nu \mathbb{E}_\zeta[e^{s(\mathbf{w}; \zeta) - \nu} + \nu - 1]. \tag{19}$$

Then, the original optimization problem (17) is transformed into a min-min optimization:

$$\min_{\mathbf{w}, \nu} F(\mathbf{w}, \nu) := \mathbb{E}_\zeta[e^{s(\mathbf{w}; \zeta) - \nu} + \nu],$$

where we ignore the constant $-1$ in the objective. A benefit of this reformulation is that unbiased stochastic gradient of $\mathbf{w}$ and $\nu$ can be easily computed so that standard SGD can be applied to update them. Below, we present a variant using alternating updates. Given $(\mathbf{w}_t, \nu_{t-1})$, we first update $\nu_t$ by a SGD step, and then update $\mathbf{w}_{t+1}$ given $\nu_t$ by another SGD step:

$$\nu_t = \nu_{t-1} - \alpha'_t[1 - e^{s(\mathbf{w}_t; \zeta_t) - \nu_{t-1}}],$$
$$\mathbf{w}_{t+1} = \mathbf{w}_t - \eta_t e^{s(\mathbf{w}_t; \zeta'_t) - \nu_t} \nabla s(\mathbf{w}_t; \zeta'_t),$$

where $\zeta_t, \zeta'_t$ are independent random variables.

**Limitation:** Although simple in design, this algorithm suffers from severe numerical instability issues and converges slowly in practice.

## A.3. Stochastic Compositional Gradient Descent (SCGD) for Compositional Optimization.

Another perspective is to view the original problem (17) as an instance of stochastic compositional optimization:

$$\min_{\mathbf{w} \in \mathcal{W}} f(g(\mathbf{w})),$$

where $f(\cdot) = \log(\cdot)$ and $g(\mathbf{w}) = \mathbb{E}_\zeta[e^{s(\mathbf{w};\zeta)}]$. Various studies have considered this problem and proposed different algorithms. We consider a basic algorithm called SCGD, which has the following update:

$$u_t = (1 - \gamma_t)u_{t-1} + \gamma_t e^{s(\mathbf{w}_t;\zeta_t)}$$

$$\mathbf{w}_{t+1} = \mathbf{w}_t - \eta_t \frac{e^{s(\mathbf{w}_t;\zeta_t')}}{u_t} \nabla s(\mathbf{w}_t; \zeta_t'), \tag{20}$$

where $\gamma_t \in (0, 1)$, $u_t$ is a moving-average estimator of the inner function $g(\mathbf{w}_t)$ and the update of $\mathbf{w}_{t+1}$ uses a stochastic gradient estimator $\nabla f(u_t)\nabla e^{s(\mathbf{w}_t;\zeta_t')}$.

**Limitation:** While SCGD and its variants have been successfully applied to the optimization of the Log-E-Exp function (Wang et al., 2017), the existing convergence rate of SCGD for convex problems is known to be worse than that of standard SGD. In particular, the result in Wang & Yang (2022) has a rate of $O(1/T^{1/4})$ for convex problems, which is slower than the typical rate of $O(1/\sqrt{T})$. The algorithm presented above can be extended to optimizing the CERM problem (1), though it suffers from the same issues (please see Corollary B.8).

### A.4. Understanding BSGD/SCGD in the Framework of SCENT

Indeed, we can show that BSGD and SCGD can be viewed as SCENT (Algorithm 1) with specific choices of the learning rate $\alpha_t$.

**BSGD corresponds to $\alpha_t = \infty$.** Let us first consider the SPMD update in (5) with a mini-batch of inner samples $\mathcal{C}_t$, i.e.,

$$\nu_t = \arg\min_\nu \frac{1}{|\mathcal{C}_t|} \sum_{\zeta' \in \mathcal{C}_t} \Phi(\mathbf{w}_t, \nu; \zeta') + \frac{1}{\alpha_t} D_\varphi(\nu, \nu_{t-1}).$$

Similar to (7), we can show that the solution to the above problem satisfies

$$e^{\nu_t} = \frac{1}{1 + \alpha_t e^{\nu_{t-1}}} e^{\nu_{t-1}} + \frac{\alpha_t e^{\nu_{t-1}}}{1 + \alpha_t e^{\nu_{t-1}}} \frac{1}{|\mathcal{C}_t|} \sum_{\zeta' \in \mathcal{C}_t} e^{s(\mathbf{w}_t;\zeta')}.$$

As a result, if $\alpha_t = \infty$, then $e^{\nu_t} = \frac{1}{|\mathcal{C}_t|} \sum_{\zeta' \in \mathcal{C}_t} e^{s(\mathbf{w}_t;\zeta')}$. Then the update of $\mathbf{w}_t$ in (8), if using the same sample $\zeta_t' \in \mathcal{C}_t$ and ignoring the projection, becomes:

$$\mathbf{w}_{t+1} = \mathbf{w}_t - \frac{\eta_t}{|\mathcal{C}_t|} \sum_{\zeta \in \mathcal{C}_t} e^{s(\mathbf{w}_t;\zeta) - \nu_t} \nabla s(\mathbf{w}_t; \zeta),$$

which is exactly the BSGD update (18). From this perspective, we see that BSGD does not have a mechanism to account for the noise in the stochastic estimators, which is the major reason why BSGD does not ensure convergence if the batch size of $\mathcal{C}_t$ is small.

**SCGD corresponds to $\alpha_t = \gamma_t' e^{-\nu_t}$.** If we set $\alpha_t = \gamma_t' e^{-\nu_t}$, then the SPMD update in (7) becomes:

$$e^{\nu_t} = \frac{1}{1 + \gamma_t'} e^{\nu_{t-1}} + \frac{\gamma_t'}{1 + \gamma_t'} e^{s(\mathbf{w}_t;\zeta_t)}.$$

Using a variable change $u_t = e^{\nu_t}$, the above update is equivalent to

$$u_t = \frac{1}{1 + \gamma_t'} u_{t-1} + \frac{\gamma_t'}{1 + \gamma_t'} e^{s(\mathbf{w}_t;\zeta_t)},$$

which is exactly the SCGD update (20) with $\gamma_t = \frac{\gamma_t'}{1 + \gamma_t'}$. From this perspective, our analysis of SCENT can yield a faster convergence rate of SCGD for minimizing the Log-E-Exp function, as discussed in the next section.

**SOX for solving the CERM problem.** The benefit of SCENT is better understood by considering the extension of SCGD for solving the CERM problem, which was proposed and analyzed by Wang & Yang (2022). The algorithm is known as

---

**Algorithm 2** The SCENT Algorithm for Solving Log-E-Exp (22)

---

1: Initialize $\mathbf{w}_1, \nu_0$, step sizes $\eta_t$ and $\alpha_t$, $\varphi(\nu) = e^{-\nu}$.
2: **for** $t = 1 \ldots, T-1$ **do**
3:     Sample $\zeta_t, \zeta_t'$
4:     Update $\nu_t = \arg\min_\nu e^{s(\mathbf{w}_t;\zeta_t)-\nu} + \nu + \frac{1}{\alpha_t}D_\varphi(\nu, \nu_{t-1})$
5:     Compute $\mathbf{v}_t = e^{s(\mathbf{w}_t;\zeta_t')-\nu_t}\nabla s(\mathbf{w}_t;\zeta_t')$
6:     Update $\mathbf{w}_{t+1} = \Pi_\mathcal{W}[\mathbf{w}_t - \eta_t\mathbf{v}_t]$
7: **end for**

---

SOX and the update is given by:

$$u_{i,t} = \begin{cases} (1-\gamma_t)u_{i,t-1} + \gamma_t e^{s(\mathbf{w}_t;\zeta_{i,t})} & i \in \mathcal{B}_t \\ u_{i,t-1} & i \notin \mathcal{B}_t \end{cases}$$

$$\mathbf{z}_t = \frac{1}{|\mathcal{B}_t|}\sum_{i\in\mathcal{B}_t}\frac{e^{s_i(\mathbf{w}_t;\zeta_{i,t}')}}{u_{i,t}}\nabla s_i(\mathbf{w}_t;\zeta_{i,t}'),$$

$$\mathbf{w}_{t+1} = \Pi_\mathcal{W}[\mathbf{w}_t - \eta_t\mathbf{z}_t].$$

A similar connection with our framework can be established. In particular, if we change the global step size $\alpha_t$ in (9) to coordinate-dependent step sizes $\alpha_{t,i} = \gamma_t' e^{-\nu_{i,t-1}}$, then the update of $\nu_{i,t}$ in (9) becomes

$$e^{\nu_{i,t}} = \frac{1}{1+\gamma_t'}e^{\nu_{i,t-1}} + \frac{\gamma_t'}{1+\gamma_t'}e^{s_i(\mathbf{w}_t;\zeta_{i,t})},$$

which is equivalent to the $u_{i,t}$ update above with change of variable.

## B. Convergence Analysis of SCENT for Solving the Log-E-Exp Problem (CERM with $n=1$)

In this section, we present the results of solving a special case of CERM (1) when $n=1$:

$$\min_\mathbf{w} F_{\text{CERM}}(\mathbf{w}) = \log\left(\mathbb{E}_\zeta[e^{s(\mathbf{w};\zeta)}]\right). \tag{21}$$

The problem is also known as Log-E-Exp, a more general form of the Log-Sum-Exp function, where the middle "E" denotes an expectation and highlights the associated computational challenges. The min-min reformulation of Log-E-Exp is

$$\min_\mathbf{w}\min_\nu F(\mathbf{w},\nu) = \mathbb{E}_\zeta[e^{s(\mathbf{w};\zeta)-\nu}] + \nu, \tag{22}$$

where we ignored the constant $-1$ in the objective; see (19). The SCENT algorithm for this case is presented in Algorithm 2.

### B.1. Properties of Log-E-Exp and SCENT

In this section, we will introduce some basic properties of the Log-E-Exp problem and the SCENT algorithm. One useful property of the problem is its joint convexity in $\mathbf{w}$ and $\boldsymbol{\nu}$ when $s_i(\cdot;\zeta)$ is convex.

**Lemma B.1.** $F(\mathbf{w},\nu)$ is jointly convex in terms of $(\mathbf{w}^\top,\nu)^\top$ if $s(\cdot;\zeta)$ is convex $\forall\,\zeta$.

*Proof.* Let $\Phi(\mathbf{w},\nu;\zeta) = e^{s(\mathbf{w};\zeta)-\nu} + \nu$. We prove that $\Phi(\mathbf{w},\nu;\zeta)$ is jointly convex in terms of $(\mathbf{w}^\top,\nu)^\top$. Then the convexity of $F(\mathbf{w},\nu)$ follows. Let $\mathbf{u} = (\mathbf{w}^\top,\nu)^\top$. Consider $\mathbf{u}_1, \mathbf{u}_2, \alpha \in [0,1]$, and $\bar{\mathbf{u}} = (\bar{\mathbf{w}}^\top,\bar{\nu})^\top = \alpha\mathbf{u}_1 + (1-\alpha)\mathbf{u}_2$. If $s(\cdot;\zeta)$ is convex, we have $s(\bar{\mathbf{w}};\zeta) \le \alpha s(\mathbf{w}_1;\zeta) + (1-\alpha)s(\mathbf{w}_2;\zeta)$. Since the exponential function is non-decreasing, we have

$$\exp(s(\bar{\mathbf{w}};\zeta) - \bar{\nu}) \le \exp(\alpha(s(\mathbf{w}_1;\zeta) - \nu_1) + (1-\alpha)(s(\mathbf{w}_2;\zeta) - \nu_2)).$$

Since the exponential function is convex, we further have

$$\exp(\alpha(s(\mathbf{w}_1;\zeta) - \nu_1) + (1-\alpha)(s(\mathbf{w}_2;\zeta) - \nu_2))$$
$$\le \alpha\exp(s(\mathbf{w}_1;\zeta) - \nu_1) + (1-\alpha)\exp(s(\mathbf{w}_2;\zeta) - \nu_2).$$

Thus, $\Phi(\mathbf{u}; \zeta)$ is convex in terms of $\mathbf{u}$ because

$$\Phi(\bar{\mathbf{u}}; \zeta) \leq \alpha \Phi(\mathbf{u}_1; \zeta) + (1 - \alpha)\Phi(\mathbf{u}_2; \zeta).$$

Then we complete the proof. $\qquad\square$

An advantage of the proximal mirror descent update of $\nu$ in SCENT, as shown in Lemma 3.1, is that it admits a closed-form solution. Here we present its proof.

*Proof of Lemma 3.1.* From (6) we have

$$\frac{\partial}{\partial \nu} D_\varphi(\nu, \nu_{t-1}) = -\varphi(\nu) - \varphi'(\nu_{t-1}).$$

With $\varphi(\nu) = e^{-\nu}$, we compute the gradient of the problem (5) and set it to zero for computing the optimal solution $\nu_t$, i.e.,

$$-e^{s(\mathbf{w}_t; \zeta_t) - \nu_t} + 1 + \frac{1}{\alpha_t}(-e^{-\nu_t} + e^{-\nu_{t-1}}) = 0,$$

which is

$$-\left(e^{s(\mathbf{w}_t; \zeta_t)} + \frac{1}{\alpha_t}\right)e^{-\nu_t} + 1 + \frac{1}{\alpha_t}e^{-\nu_{t-1}} = 0. \tag{23}$$

Rearranging the terms, we get

$$\begin{aligned}
e^{\nu_t} &= \frac{e^{s(\mathbf{w}_t; \zeta_t)} + 1/\alpha_t}{1 + e^{-\nu_{t-1}}/\alpha_t} \\
&= \frac{e^{\nu_{t-1}} + \alpha_t e^{\nu_{t-1}} e^{s(\mathbf{w}_t; \zeta_t)}}{1 + \alpha_t e^{\nu_{t-1}}},
\end{aligned}$$

which leads to (7). This completes the proof. $\qquad\square$

Moreover, we also have the following update of $e^{-\nu_t}$.

**Lemma B.2.** *Let $\pi_t = e^{-\nu_t}$. If $\nu_t$ follows the update of (5) with a Bregman divergence defined in (6), we have*

$$\pi_t = \frac{\pi_{t-1} + \alpha_t}{1 + \alpha_t e^{s(\mathbf{w}_t; \zeta_t)}}.$$

*Proof.* From (23) and rearranging the terms, we can immediately get the desired result. $\qquad\square$

We can show that the following terms are bounded with the update of $\nu$ in SCENT for the Log-E-Exp problem.

**Lemma B.3.** *Under Assumption 3.2 (ii), if $\nu_0 \in [c_0, c_1]$, then $\nu_t \in [c_0, c_1], \forall t$. If in addition Assumption 3.2 (iii) holds, and let*

$$\begin{aligned}
\sigma_t^2 &:= \mathbb{E}_{\zeta_t'}[\|e^{s(\mathbf{w}_t; \zeta_t') - \nu_t} \nabla s(\mathbf{w}_t; \zeta_t')\|_2^2], \\
\delta_t^2 &:= \mathbb{E}_{\zeta_t}[e^{-\nu_{t-1}}|e^{s(\mathbf{w}_t; \zeta_t)} - \mathbb{E}_\zeta[e^{s(\mathbf{w}_t; \zeta)}]|^2],
\end{aligned}$$

*then $\sigma_t, \delta_t$ are finite $\forall t$.*

*Proof.* The proof of this lemma is by induction. It is trivial that $\nu_0 \in [c_0, c_1]$. If the result holds for $v_{t-1}$, then $e^{\nu_{t-1}} \in [e^{c_0}, e^{c_1}]$. Assumption 3.2 (ii) implies that $e^{s(\mathbf{w}_t; \zeta_t)} \in [e^{c_0}, e^{c_1}]$ as well. As $e^{\nu_t}$ in (7) is a convex combination of $e^{\nu_{t-1}}$ and $e^{s(\mathbf{w}_t; \zeta_t)}$, we have $e^{\nu_t} \in [e^{c_0}, e^{c_1}]$. Thus, $\nu_t \in [c_0, c_1]$. Then we know $\sigma_t, \delta_t$ are finite because $e^{\nu_t}, e^{\nu_{t-1}}$ and $\exp(s(\mathbf{w}_t; \zeta_t))$ are upper and lower bounded, and $\mathbb{E}_{\zeta_t'}[\|\nabla s(\mathbf{w}_t; \zeta_t')\|_2^2]$ is upper bounded from Assumption 3.2 (iii). This completes the proof. $\qquad\square$

## B.2. Convergence Analysis of SCENT

In order to prove the convergence of SCENT for solving the Log-E-Exp problem, we need the following three lemmas.

**Lemma B.4.** *Under Assumption 3.2, if $\alpha_t \leq \rho e^{-\nu_{t-1}}$, then we have*

$$|\mathbb{E}[(\nabla_\nu F(\mathbf{w}_t, \nu_t)^\top (\nu_t - \nu_*) - \nabla_\nu \Phi(\mathbf{w}_t, \nu_t; \zeta_t)^\top (\nu_t - \nu_*)]| \leq \alpha_t \delta_t^2 (1 + \rho)(1 + c_1 - c_0).$$

*Proof.* In the following proof, $\mathcal{F}_{t-1}$ denotes the filtration (i.e., the "information available") up to iteration $t-1$. Define $z_t = e^{s(\mathbf{w}_t; \zeta_t)}$, $m_t = \mathbb{E}_\zeta[e^{s(\mathbf{w}_t; \zeta)} | \mathcal{F}_{t-1}]$, and $\pi_t = e^{-\nu_t}$. Since $\nu_t$ depends on $z_t$, we define the following random functions:

$$\pi_t(z) = \frac{\pi_{t-1} + \alpha_t}{\alpha_t z + 1}, \quad \nu_t(z) = -\log \pi_t(z)$$

$$h_t(z) = e^{-\nu_t(z)} \big(\nu_t(z) - \nu_*\big).$$

According to Lemma B.2, we have $\pi_t = \pi_t(z_t), \nu_t = \nu_t(z_t)$, and thus $h_t(z_t) = \pi_t(z_t)(\nu_t(z_t) - \nu_*)$. For the target, we have

$$\mathbb{E}[(\nabla_\nu \Phi(\mathbf{w}_t, \nu_t; \zeta_t) - \nabla_\nu F(\mathbf{w}_t, \nu_t))^\top (\nu_t - \nu_*) \mid \mathcal{F}_{t-1}] = \mathbb{E}[(\mathbb{E}_\zeta[e^{s(\mathbf{w}_t; \zeta)}] - e^{s(\mathbf{w}_t; \zeta_t)})e^{-\nu_t}(\nu_t - \nu_*) \mid \mathcal{F}_{t-1}]$$
$$= \mathbb{E}[(m_t - z_t)h_t(z_t) \mid \mathcal{F}_{t-1}] = \mathbb{E}_z[(m_t - z)h_t(z)|\mathcal{F}_{t-1}]. \quad (24)$$

Let $z$ and $z'$ two independent variables so that $\mathbb{E}[z|\mathcal{F}_{t-1}] = \mathbb{E}[z'|\mathcal{F}_{t-1}] = m_t$. Using the conditional independence,

$$\mathbb{E}[(m_t - z)h_t(z) \mid \mathcal{F}_{t-1}] = \mathbb{E}[(z' - z)h_t(z) \mid \mathcal{F}_{t-1}].$$

By the exchangeability of $(z, z')$ conditioned on $\mathcal{F}_{t-1}$,

$$\mathbb{E}[(z' - z)h_t(z') \mid \mathcal{F}_{t-1}] = -\mathbb{E}[(z' - z)h_t(z) \mid \mathcal{F}_{t-1}].$$

Combining the above two equations, we get

$$\mathbb{E}[(m_t - z)h_t(z) \mid \mathcal{F}_{t-1}] = \frac{1}{2}\mathbb{E}[(z' - z)\big(h_t(z) - h_t(z')\big) \mid \mathcal{F}_{t-1}]. \quad (25)$$

Next, we show that $h(z)$ is Lipschitz continuous. By definition,

$$\pi_t(z) = \frac{\pi_{t-1} + \alpha_t}{\alpha_t z + 1}, \quad h_t(z) = \pi_t(z)\big(\nu_t(z) - \nu_*\big).$$

Differentiating $\pi_t(z)$ with respect to $z$, we get

$$\frac{d\pi_t(z)}{dz} = (\pi_{t-1} + \alpha_t)\frac{d}{dz}\big((\alpha_t z + 1)^{-1}\big) = -\frac{\alpha_t(\pi_{t-1} + \alpha_t)}{(\alpha_t z + 1)^2}.$$

Using the equation $\pi_t(z)(\alpha_t z + 1) = \pi_{t-1} + \alpha_t$, we can rewrite the above equality as

$$\frac{d\pi_t(z)}{dz} = -\frac{\alpha_t \pi_t(z)}{\alpha_t z + 1}.$$

Since $\nu_t(z) = -\log \pi_t(z)$, we have

$$\frac{d\nu_t(z)}{dz} = -\frac{1}{\pi_t(z)}\frac{d\pi_t(z)}{dz} = \frac{\alpha_t}{\alpha_t z + 1}.$$

As a result,

$$\frac{dh_t(z)}{dz} = \frac{d\pi_t(z)}{dz}\big(\nu_t(z) - \nu_*\big) + \pi_t(z)\frac{d\nu_t(z)}{dz} = \frac{\alpha_t \pi_t(z)}{\alpha_t z + 1}\big(1 - (\nu_t(z) - \nu_*)\big).$$

From Assumption 3.2 (ii), we have

$$\mathbb{E}_\zeta[e^{s(\mathbf{w}_*; \zeta)}] \in [e^{c_0}, e^{c_1}], \quad \nu_* = \log \mathbb{E}_\zeta[e^{s(\mathbf{w}_*; \zeta)}] \in [c_0, c_1].$$

Since $\nu_t(z) \in [c_0, c_1]$ as well, we get

$$\big|1 - (\nu_t(z) - \nu_*)\big| \leq 1 + c_1 - c_0.$$

Since $\pi_t(z) = \frac{\pi_{t-1} + \alpha_t}{\alpha_t z + 1} \leq \pi_{t-1} + \alpha_t \leq (1 + \rho)\pi_{t-1}$, we have

$$\left|\frac{dh_t}{dz}\right| \leq \alpha_t \pi_{t-1}(1 + \rho)(1 + c_1 - c_0),$$

which means $h_t$ is $L_t$-Lipschitz with $L_t \leq \alpha_t \pi_{t-1}(1 + \rho)(1 + c_1 - c_0)$. Then we have

$$\big|(z' - z)\big(h_t(z) - h_t(z')\big)\big| \leq L_t (z' - z)^2 \leq C\alpha_t \pi_{t-1}(z' - z)^2,$$

where $C = (1 + \rho)(1 + c_1 - c_0)$. Thus,

$$\mathbb{E}\left[\left|(z' - z)\big(h_t(z) - h_t(z')\big)\right| \mid \mathcal{F}_{t-1}\right] \leq C\alpha_t \mathbb{E}[\pi_{t-1}(z' - z)^2 \mid \mathcal{F}_{t-1}]$$

$$\leq C\alpha_t \cdot 2\mathbb{E}[\pi_{t-1}(z - \mathbb{E}[z])^2 \mid \mathcal{F}_{t-1}] = 2C\alpha_t \delta_t^2,$$

where the last step uses the second equality in Lemma B.3. Applying the above result to (25), we have

$$\left|\mathbb{E}\big[(m_t - z)h_t(z) \mid \mathcal{F}_{t-1}\big]\right| = \frac{1}{2}\mathbb{E}\left[\left|(z' - z)\big(h_t(z) - h_t(z')\big)\right| \mid \mathcal{F}_{t-1}\right] \leq C\alpha_t \delta_t^2.$$

By noting (24) and the definition of $C = (1 + \rho)(1 + c_1 - c_0)$, we finish the proof. $\qquad\square$

The following lemma characterizes the change when we update $\nu_{t+1}$ from $\nu_t$.

**Lemma B.5.** *Let* $\Phi(\mathbf{w}, \nu; \zeta) = e^{s(\mathbf{w};\zeta) - \nu} + \nu$, *and consider the update of* $\nu_t$:

$$\nu_t = \arg\min_{\nu} \alpha_t \Phi(\mathbf{w}_t, \nu; \zeta_t) + D_\varphi(\nu, \nu_{t-1}).$$

*Then we have*

$$\alpha_t \nabla_\nu \Phi(\mathbf{w}_t, \nu_t; \zeta_t)^\top (\nu_t - \nu_*) \leq D_\varphi(\nu_*, \nu_{t-1}) - D_\varphi(\nu_*, \nu_t) - D_\varphi(\nu_t, \nu_{t-1}).$$

*Proof.* Recall the definition

$$\varphi(\nu) = e^{-\nu}, \quad D_\varphi(a, b) = \varphi(a) - \varphi(b) - \langle \nabla\varphi(b), a - b \rangle.$$

The first-order optimality of $\nu_t$ gives

$$\alpha_t \nabla_\nu \Phi(\mathbf{w}_t, \nu_t; \zeta_t) + \nabla\varphi(\nu_t) - \nabla\varphi(\nu_{t-1}) = 0.$$

Taking inner product with $(\nu_t - \nu_*)$ and rearranging the terms, we get

$$\alpha_t \nabla_\nu \Phi(\mathbf{w}_t, \nu_t; \zeta_t)^\top (\nu_t - \nu_*) = (\nabla\varphi(\nu_{t-1}) - \nabla\varphi(\nu_t))^\top (\nu_t - \nu_*). \tag{26}$$

We have

$$D_\varphi(\nu_*, \nu_t) - D_\varphi(\nu_*, \nu_{t-1})$$
$$= -\varphi(\nu_t) - \nabla\varphi(\nu_t)^\top (\nu_* - \nu_t) + \varphi(\nu_{t-1}) + \nabla\varphi(\nu_{t-1})^\top (\nu_* - \nu_{t-1})$$
$$= (\nabla\varphi(\nu_t) - \nabla\varphi(\nu_{t-1}))^\top (\nu_t - \nu_*) - \varphi(\nu_t) + \varphi(\nu_{t-1}) + \nabla\varphi(\nu_{t-1})^\top (\nu_t - \nu_{t-1})$$
$$= (\nabla\varphi(\nu_t) - \nabla\varphi(\nu_{t-1}))^\top (\nu_t - \nu_*) - D_\varphi(\nu_t, \nu_{t-1}).$$

Rearranging the terms, we get

$$(\nabla\varphi(\nu_{t-1}) - \nabla\varphi(\nu_t))^\top (\nu_t - \nu_*) = D_\varphi(\nu_*, \nu_{t-1}) - D_\varphi(\nu_*, \nu_t) - D_\varphi(\nu_t, \nu_{t-1}). \tag{27}$$

Combining (26) and (27) completes the proof. $\qquad\square$

The following lemma characterizes the change when we update $\mathbf{w}_{t+1}$ from $\mathbf{w}_t$.

**Lemma B.6.** *Under Assumption 3.2 (ii), let* $\Phi(\mathbf{w}, \nu; \zeta) = e^{s(\mathbf{w};\zeta) - \nu} + \nu$ *and* $\sigma_t^2 := \mathbb{E}_{\zeta_t'}[\|\nabla_\mathbf{w}\Phi(\mathbf{w}, \nu; \zeta)\|_2^2]$, *and consider the update of* $\mathbf{w}_{t+1}$:

$$\mathbf{w}_{t+1} = \Pi_\mathcal{W}[\mathbf{w}_t - \eta_t \nabla_\mathbf{w}\Phi(\mathbf{w}_t, \nu_t; \zeta_t')].$$

*Then we have*

$$\mathbb{E}[\nabla_\mathbf{w} F(\mathbf{w}_t, \nu_t)^\top (\mathbf{w}_t - \mathbf{w}_*)] \leq \mathbb{E}\left[\frac{1}{2\eta_t}\|\mathbf{w}_* - \mathbf{w}_t\|_2^2 - \frac{1}{2\eta_t}\|\mathbf{w}_* - \mathbf{w}_{t+1}\|_2^2\right] + \frac{\eta_t}{2}\sigma_t^2.$$

*Proof.* First we note that a finite $\sigma_t^2$ is well-defined from Assumption 3.2 (ii) and Lemma B.3. Note that the update of $\mathbf{w}_{t+1}$ is equivalent to

$$\mathbf{w}_{t+1} = \arg\min_{\mathbf{w}} \nabla_\mathbf{w}\Phi(\mathbf{w}_t, \nu_t; \zeta_t')^\top (\mathbf{w} - \mathbf{w}_t) + \frac{1}{2\eta_t}\|\mathbf{w} - \mathbf{w}_t\|_2^2 + r(\mathbf{w}),$$

where

$$r(\mathbf{w}) = 1_{\mathcal{W}}(\mathbf{w}) = \begin{cases} 0, & \text{if } \mathbf{w} \in \mathcal{W}, \\ +\infty, & \text{otherwise.} \end{cases}$$

By the first-order optimality condition, for any $\mathbf{w}$ we have

$$(\nabla_{\mathbf{w}}\Phi(\mathbf{w}_t, \nu_t; \zeta_t') + \partial r(\mathbf{w}_{t+1}) + \frac{1}{\eta_t}(\mathbf{w}_{t+1} - \mathbf{w}_t))^{\top}(\mathbf{w} - \mathbf{w}_{t+1}) \geq 0.$$

By the convexity of $r$, we have

$$r(\mathbf{w}_{t+1}) \leq r(\mathbf{w}) + \partial r(\mathbf{w}_{t+1})^{\top}(\mathbf{w}_{t+1} - \mathbf{w}).$$

Combining the above two inequalities, we have

$$\nabla_{\mathbf{w}}\Phi(\mathbf{w}_t, \nu_t; \zeta_t')^{\top}(\mathbf{w}_{t+1} - \mathbf{w}) + r(\mathbf{w}_{t+1}) - r(\mathbf{w}) \leq \frac{1}{\eta_t}(\mathbf{w}_t - \mathbf{w}_{t+1})^{\top}(\mathbf{w}_{t+1} - \mathbf{w})$$

$$= \frac{1}{2\eta_t}(\|\mathbf{w}_t - \mathbf{w}\|_2^2 - \|\mathbf{w}_{t+1} - \mathbf{w}\|_2^2 - \|\mathbf{w}_t - \mathbf{w}_{t+1}\|_2^2),$$

where the last equality uses the fact that $2(a-b)^{\top}(b-c) = \|a-c\|_2^2 - \|a-b\|_2^2 - \|b-c\|_2^2$. When $\mathbf{w} = \mathbf{w}_*$, we have $\mathbf{w}_{t+1}, \mathbf{w}_* \in \mathcal{W}$, and thus $r(\mathbf{w}_{t+1}) = r(\mathbf{w}_*) = 0$. Rearranging the terms, we get

$$\nabla_{\mathbf{w}}\Phi(\mathbf{w}_t, \nu_t; \zeta_t')^{\top}(\mathbf{w}_t - \mathbf{w}_*)$$

$$\leq \frac{1}{2\eta_t}\|\mathbf{w}_* - \mathbf{w}_t\|_2^2 - \frac{1}{2\eta_t}\|\mathbf{w}_* - \mathbf{w}_{t+1}\|_2^2 - \frac{1}{2\eta_t}\|\mathbf{w}_{t+1} - \mathbf{w}_t\|_2^2 + \nabla_{\mathbf{w}}\Phi(\mathbf{w}_t, \nu_t; \zeta_t')^{\top}(\mathbf{w}_{t+1} - \mathbf{w}_t)$$

$$\leq \frac{1}{2\eta_t}\|\mathbf{w}_* - \mathbf{w}_t\|_2^2 - \frac{1}{2\eta_t}\|\mathbf{w}_* - \mathbf{w}_{t+1}\|_2^2 - \frac{1}{2\eta_t}\|\mathbf{w}_{t+1} - \mathbf{w}_t\|_2^2 + \frac{\eta_t}{2}\|\nabla_{\mathbf{w}}\Phi(\mathbf{w}_t, \nu_t; \zeta_t')\|_2^2 + \frac{1}{2\eta_t}\|\mathbf{w}_{t+1} - \mathbf{w}_t\|_2^2,$$

where the last inequality uses the Young's inequality. Taking expectation on both sides, and recalling the definition of $\sigma_t^2$, we have

$$\mathbb{E}[\nabla_{\mathbf{w}}\Phi(\mathbf{w}_t, \nu_t; \zeta_t')^{\top}(\mathbf{w}_t - \mathbf{w}_*)] \leq \mathbb{E}\left[\frac{1}{2\eta_t}\|\mathbf{w}_* - \mathbf{w}_t\|_2^2 - \frac{1}{2\eta_t}\|\mathbf{w} - \mathbf{w}_{t+1}\|_2^2\right] + \frac{\eta_t}{2}\sigma_t^2.$$

Since $\mathbf{w}_t$ is independent of $\zeta_t'$, we have $\mathbb{E}[\nabla_{\mathbf{w}}\Phi(\mathbf{w}_t, \nu_t; \zeta_t')^{\top}(\mathbf{w}_t - \mathbf{w}_*)] = \mathbb{E}[\nabla_{\mathbf{w}}F(\mathbf{w}_t, \nu_t)^{\top}(\mathbf{w}_t - \mathbf{w}_*)]$. Thus we get

$$\mathbb{E}[\nabla_{\mathbf{w}}F(\mathbf{w}_t, \nu_t)^{\top}(\mathbf{w}_t - \mathbf{w}_*)] = \mathbb{E}[\nabla_{\mathbf{w}}\Phi(\mathbf{w}_t, \nu_t; \zeta_t')^{\top}(\mathbf{w}_t - \mathbf{w}_*)]$$

$$\leq \mathbb{E}\left[\frac{1}{2\eta_t}\|\mathbf{w}_* - \mathbf{w}_t\|_2^2 - \frac{1}{2\eta_t}\|\mathbf{w}_* - \mathbf{w}_{t+1}\|_2^2\right] + \frac{\eta_t}{2}\sigma_t^2.$$

Then we complete the proof. $\qquad\square$

Now we are ready to prove the convergence of SCENT.

**Theorem B.7.** *Under Assumption 3.2, let $\eta_t = \eta\alpha_t$, $\alpha_t < \rho e^{-\nu_{t-1}}$. Then SCENT guarantees that*

$$\mathbb{E}\left[\sum_{t=1}^{T}\alpha_t(F(\mathbf{w}_t, \nu_t) - F(\mathbf{w}_*, \nu_*))\right] \leq \frac{1}{2\eta}\|\mathbf{w}_1 - \mathbf{w}_*\|_2^2 + D_{\varphi}(\nu_*, \nu_0) + \mathbb{E}\left[\sum_{t=1}^{T}\frac{\eta\alpha_t^2\sigma_t^2}{2} + \sum_{t=1}^{T}C\alpha_t^2\delta_t^2\right].$$

*Proof.* Since $\eta_t = \eta\alpha_t$, from Lemma B.6, we obtain

$$\mathbb{E}[\alpha_t\nabla_{\mathbf{w}}F(\mathbf{w}_t, \nu_t)^{\top}(\mathbf{w}_t - \mathbf{w}_*)] \leq \mathbb{E}\left[\frac{1}{2\eta}\|\mathbf{w}_t - \mathbf{w}_*\|_2^2 - \frac{1}{2\eta}\|\mathbf{w}_{t+1} - \mathbf{w}_*\|_2^2 + \frac{\eta\alpha_t^2\sigma_t^2}{2}\right].$$

Combining the above inequality with Lemmas B.4 and B.5, we get

$$\mathbb{E}[\alpha_t(\nabla_{\mathbf{w}}F(\mathbf{w}_t, \nu_t)^{\top}(\mathbf{w}_t - \mathbf{w}_*) + \nabla_{\nu}F(\mathbf{w}_t, \nu_t)^{\top}(\nu_t - \nu_*))]$$

$$\leq \mathbb{E}\left[\frac{1}{2\eta}\|\mathbf{w}_t - \mathbf{w}_*\|_2^2 - \frac{1}{2\eta}\|\mathbf{w}_{t+1} - \mathbf{w}_*\|_2^2 + D_{\varphi}(\nu_*, \nu_{t-1}) - D_{\varphi}(\nu_*, \nu_t)\right] + \mathbb{E}\left[\frac{\eta\alpha_t^2\sigma_t^2}{2} + C\alpha_t^2\delta_t^2\right]. \tag{28}$$

By the joint convexity of $F(\mathbf{w}, \nu)$ from Lemma B.1, we have

$$\alpha_t(F(\mathbf{w}_t, \nu_t) - F(\mathbf{w}_*, \nu_*)) \leq \alpha_t(\nabla_{\mathbf{w}}F(\mathbf{w}_t, \nu_t)^{\top}(\mathbf{w}_t - \mathbf{w}_*) + \nabla_{\nu}F(\mathbf{w}_t, \nu_t)^{\top}(\nu_t - \nu_*)). \tag{29}$$

Combining (28) and (29) and summing over $t = 1, \ldots, T$, we have

$$\mathbb{E}\left[\sum_{t=1}^{T} \alpha_t (F(\mathbf{w}_t, \nu_t) - F(\mathbf{w}_*, \nu_*))\right] \leq \frac{1}{2\eta}\|\mathbf{w}_1 - \mathbf{w}_*\|_2^2 + D_\varphi(\nu_*, \nu_0) + \mathbb{E}\left[\sum_{t=1}^{T} \frac{\eta \alpha_t^2 \sigma_t^2}{2} + \sum_{t=1}^{T} C\alpha_t^2 \delta_t^2\right].$$

Then we complete the proof. $\qquad \square$

Next we present a corollary of Theorem B.7 with a specific choice of learning rate for $\nu$, which leads to the SCGD algorithm.

**Corollary B.8.** *Under Assumption 3.2, let* $\eta_t = \eta \alpha_t$, $\alpha_t = \frac{\alpha e^{-\nu_{t-1}}}{\sqrt{T}}$. *If* $\frac{1}{T}\sum_{t=1}^{T} e^{-\nu_{t-1}} \geq S$ *almost surely, then SCENT guarantees that*

$$\mathbb{E}\left[F_{\text{CERM}}(\hat{\mathbf{w}}_T) - F_{\text{CERM}}(\mathbf{w}_*)\right] \leq \frac{\frac{1}{2\eta}\|\mathbf{w}_1 - \mathbf{w}_*\|_2^2 + D_\varphi(\nu_*, \nu_0)}{\alpha\sqrt{T}S} + \frac{\alpha\bar{V}}{\sqrt{T}S}.$$

*where* $\hat{\mathbf{w}}_T = \frac{\sum_t \alpha_t \mathbf{w}_t}{\sum_{t=1}^{T} \alpha_t}$ *and*

$$\bar{V} = \mathbb{E}\left[\frac{\eta \sum_{t=1}^{T} e^{-2\nu_{t-1}}\sigma_t^2}{2T} + \frac{\sum_{t=1}^{T} Ce^{-2\nu_{t-1}}\delta_t^2}{T}\right].$$

*Proof.* Let $\hat{\alpha}_t = \frac{\alpha_t}{\sum_{t'=1}^{T} \alpha_{t'}}$. From Theorem B.7, we have

$$\mathbb{E}\left[\sum_{t'=1}^{T} \alpha_{t'}\left(\sum_{t=1}^{T} \hat{\alpha}_t (F(\mathbf{w}_t, \nu_t) - F(\mathbf{w}_*, \nu_*))\right)\right] \leq \frac{1}{2\eta}\|\mathbf{w}_1 - \mathbf{w}_*\|_2^2 + D_\varphi(\nu_*, \nu_0) + \mathbb{E}\left[\sum_{t=1}^{T} \frac{\eta \alpha_t^2 \sigma_t^2}{2} + \sum_{t=1}^{T} C\alpha_t^2 \delta_t^2\right].$$

Since $\sum_{t'=1}^{T} \alpha_{t'} = \sum_{t'=1}^{T} \frac{\alpha e^{-\nu_{t'-1}}}{\sqrt{T}} \geq \alpha\sqrt{T}S$, then from Jensen's inequality

$$\mathbb{E}\left[\sum_{t=1}^{T} \hat{\alpha}_t (F(\mathbf{w}_t, \nu_t) - F(\mathbf{w}_*, \nu_*))\right] \leq \frac{\frac{1}{2\eta}\|\mathbf{w}_1 - \mathbf{w}_*\|_2^2 + D_\varphi(\nu_*, \nu_0)}{\alpha\sqrt{T}S} + \frac{\alpha\bar{V}}{\sqrt{T}S}.$$

Applying the joint convexity of $F(\mathbf{w}, \nu)$ and $F_{\text{CERM}} = \min_\nu F(\mathbf{w}, \nu)$, we get

$$\mathbb{E}\left[F_{\text{CERM}}(\sum_{t=1}^{T} \hat{\alpha}_t \mathbf{w}_t) - F_{\text{CERM}}(\mathbf{w}_*)\right] \leq \mathbb{E}\left[F(\sum_{t=1}^{T} \hat{\alpha}_t \mathbf{w}_t, \sum_{t=1}^{T} \hat{\alpha}_t \nu_t) - F(\mathbf{w}_*, \nu_*)\right]$$

$$\leq \mathbb{E}\left[\sum_{t=1}^{T} \hat{\alpha}_t F(\mathbf{w}_t, \nu_t) - F(\mathbf{w}_*, \nu_*)\right]$$

Combining the above two equations, we complete the proof. $\qquad \square$

## C. Proofs of Results in Section 3.1

In this section, we present the convergence analysis of SCENT for solving CERM.

*Proof of Lemma 3.3.* This lemma is directly implied from applying Lemma B.3 to each $i$. $\qquad \square$

Then we are ready to analyze the update of $\mathbf{w}_t$.

*Proof of Lemma 3.4.* Let $\mathbf{z}_t = \frac{1}{B} \sum_{i \in \mathcal{B}_t} e^{s_i(\mathbf{w}_t; \zeta'_{i,t}) - \nu_{i,t}} \nabla s_i(\mathbf{w}_t; \zeta'_{i,t})$. We first bound $\mathbb{E}[\|\mathbf{z}_t\|_2^2 \mid \mathcal{F}_{t-1}]$.

$$\mathbb{E}[\|\mathbf{z}_t\|_2^2 \mid \mathcal{F}_{t-1}] = \mathbb{E}\left[\left\|\frac{1}{B} \sum_{i \in \mathcal{B}_t} e^{s_i(\mathbf{w}_t; \zeta'_{i,t}) - \nu_{i,t}} \nabla s_i(\mathbf{w}_t; \zeta'_{i,t})\right\|_2^2 \mid \mathcal{F}_{t-1}\right]$$

$$= \mathbb{E}_{\mathcal{B}_t, \zeta_t} \mathbb{E}_{\zeta'_t}\left[\left\|\frac{1}{B} \sum_{i \in \mathcal{B}_t} e^{s_i(\mathbf{w}_t; \zeta'_{i,t}) - \nu_{i,t}} \nabla s_i(\mathbf{w}_t; \zeta'_{i,t})\right\|_2^2 \mid \mathcal{F}_{t-1}, \mathcal{B}_t, \zeta_t\right]$$

$$\leq \mathbb{E}_{\mathcal{B}_t, \zeta_t}\left[\frac{1}{B} \sum_{i \in \mathcal{B}_t} \sigma_{i,t}^2\right] = \frac{1}{n} \sum_{i=1}^{n} \sigma_{i,t}^2.$$

Since $\bar{\nu}_{i,t} = \nu_{i,t}, \forall i \in \mathcal{B}_t$, we have

$$\mathbb{E}[\mathbf{z}_t \mid \mathcal{F}_{t-1}] = \mathbb{E}_{\zeta'_t, \zeta_t, \mathcal{B}_t}\left[\frac{1}{B} \sum_{i \in \mathcal{B}_t} \nabla_{\mathbf{w}} \Phi_i(\mathbf{w}_t, \bar{\nu}_{i,t}; \zeta'_{i,t})\right] = \nabla_{\mathbf{w}} F(\mathbf{w}_t, \bar{\nu}_t).$$

Replacing $\nabla_{\mathbf{w}} \Phi(\mathbf{w}_t, \nu_t; \zeta'_t)$ with $\mathbf{z}_t$ in Lemma B.6, we finish the proof. $\square$

Next, we analyze the update of $\bar{\nu}_t$.

*Proof of Lemma 3.5.* By applying Lemma B.4 and Lemma B.5 for each coordinate of $\bar{\nu}_{i,t}$, we have

$$\mathbb{E}[\alpha_t \nabla_\nu F_i(\mathbf{w}_t, \bar{\nu}_{i,t})^\top (\bar{\nu}_{i,t} - \nu_{i,*})] \leq D_\varphi(\nu_{i,*}, \nu_{i,t-1}) - D_\varphi(\nu_{i,*}, \bar{\nu}_{i,t}) + C\alpha_t^2 \delta_{i,t}^2, \forall i.$$

Averaging the above inequality over $i = 1, \ldots, n$, we have

$$\mathbb{E}[\alpha_t \nabla_{\boldsymbol{\nu}} F(\mathbf{w}_t, \bar{\boldsymbol{\nu}}_t)^\top (\bar{\boldsymbol{\nu}}_t - \boldsymbol{\nu}_*)] \leq \frac{1}{n} \sum_{i=1}^{n} (D_\varphi(\nu_{i,*}, \nu_{i,t-1}) - D_\varphi(\nu_{i,*}, \bar{\nu}_{i,t})) + C\alpha_t^2 \delta_t^2. \tag{30}$$

Due to the randomness of $\mathcal{B}_t$, we have

$$\mathbb{E}[D_\varphi(\nu_{i,*}, \nu_{i,t})] = \mathbb{E}\left[(1 - \frac{B}{n})D_\varphi(\nu_{i,*}, \nu_{i,t-1}) + \frac{B}{n} D_\varphi(\nu_{i,*}, \bar{\nu}_{i,t})\right], \forall i.$$

Hence

$$\mathbb{E}\left[\frac{1}{n} \sum_{i=1}^{n} (D_\varphi(\nu_{i,*}, \nu_{i,t-1}) - D_\varphi(\nu_{i,*}, \bar{\nu}_{i,t}))\right]$$

$$= \mathbb{E}\left[\frac{1}{n} \sum_{i=1}^{n} \left(D_\varphi(\nu_{i,*}, \nu_{i,t-1}) - \frac{n}{B} D_\varphi(\nu_{i,*}, \nu_{i,t}) + (\frac{n}{B} - 1)D_\varphi(\nu_{i,*}, \nu_{i,t-1})\right)\right]$$

$$= \frac{1}{B} \cdot \mathbb{E}\left[\sum_{i=1}^{n}(D_\varphi(\nu_{i,*}, \nu_{i,t-1}) - D_\varphi(\nu_{i,*}, \nu_{i,t}))\right].$$

Combining the above equality with (30), we finish the proof. $\square$

Finally, we prove the convergence result of SCENT.

*Proof of Theorem 3.6.* Since $\eta_t = \eta \alpha_t$, from Lemma 3.4, we obtain

$$\mathbb{E}[\alpha_t \nabla_{\mathbf{w}} F(\mathbf{w}_t, \bar{\nu}_t)^\top (\mathbf{w}_t - \mathbf{w}_*)] \leq \mathbb{E}\left[\frac{1}{2\eta}\|\mathbf{w}_t - \mathbf{w}_*\|_2^2 - \frac{1}{2\eta}\|\mathbf{w}_{t+1} - \mathbf{w}_*\|_2^2\right] + \frac{\eta \alpha_t^2 \sigma_t^2}{2}.$$

Combining the above inequality with Lemma 3.5 and using the definition of $D_\varphi$ in Equation (11), we have

$$\mathbb{E}[\alpha_t (\nabla_{\mathbf{w}} F(\mathbf{w}_t, \bar{\nu}_t)^\top (\mathbf{w}_t - \mathbf{w}_*) + \nabla_{\boldsymbol{\nu}} F(\mathbf{w}_t, \bar{\nu}_t)^\top (\bar{\boldsymbol{\nu}}_t - \boldsymbol{\nu}_*))]$$

$$\leq \mathbb{E}\left[\frac{1}{2\eta}\|\mathbf{w}_t - \mathbf{w}_*\|_2^2 - \frac{1}{2\eta}\|\mathbf{w}_{t+1} - \mathbf{w}_*\|_2^2 + \frac{1}{B} D_\varphi(\boldsymbol{\nu}_*, \boldsymbol{\nu}_{t-1}) - \frac{1}{B} D_\varphi(\boldsymbol{\nu}_*, \boldsymbol{\nu}_t)\right] + \frac{\eta \alpha_t^2 \sigma_t^2}{2} + C\alpha_t^2 \delta_t^2. \tag{31}$$

By the joint convexity of $F(\mathbf{w}, \boldsymbol{\nu})$ from Lemma B.1, we have

$$\alpha_t(F(\mathbf{w}_t, \bar{\boldsymbol{\nu}}_t) - F(\mathbf{w}_*, \boldsymbol{\nu}_*)) \leq \alpha_t(\nabla_{\mathbf{w}} F(\mathbf{w}_t, \bar{\boldsymbol{\nu}}_t)^\top (\mathbf{w}_t - \mathbf{w}_*) + \nabla_{\boldsymbol{\nu}} F(\mathbf{w}_t, \bar{\boldsymbol{\nu}}_t)^\top (\bar{\boldsymbol{\nu}}_t - \boldsymbol{\nu}_*)). \tag{32}$$

Combining (31) and (32) and summing over $t = 1, \ldots, T$, we have

$$\mathbb{E}\left[\sum_{t=1}^{T} \alpha_t(F(\mathbf{w}_t, \bar{\boldsymbol{\nu}}_t) - F(\mathbf{w}_*, \boldsymbol{\nu}_*))\right] \leq \frac{1}{2\eta}\|\mathbf{w}_1 - \mathbf{w}_*\|_2^2 + \frac{1}{B}D_\varphi(\boldsymbol{\nu}_*, \boldsymbol{\nu}_0) + \sum_{t=1}^{T}\frac{\eta\alpha_t^2\sigma_t^2}{2} + \sum_{t=1}^{T}C\alpha_t^2\delta_t^2.$$

Since $F_{\mathrm{CERM}}(\mathbf{w}_*) = F(\mathbf{w}_*, \boldsymbol{\nu}_*)$, and $F_{\mathrm{CERM}}(\mathbf{w}_t) \leq F(\mathbf{w}_t, \bar{\boldsymbol{\nu}}_t)$, we have

$$\mathbb{E}\left[\sum_{t=1}^{T} \alpha_t(F_{\mathrm{CERM}}(\mathbf{w}_t) - F_{\mathrm{CERM}}(\mathbf{w}_*))\right] \leq \frac{1}{2\eta}\|\mathbf{w}_1 - \mathbf{w}_*\|_2^2 + \frac{1}{B}D_\varphi(\boldsymbol{\nu}_*, \boldsymbol{\nu}_0) + \sum_{t=1}^{T}\frac{\eta\alpha_t^2\sigma_t^2}{2} + \sum_{t=1}^{T}C\alpha_t^2\delta_t^2.$$

Plugging in the choice of $\alpha_t = \frac{\alpha}{\sqrt{T}}$, we obtain

$$\frac{\alpha}{\sqrt{T}}\mathbb{E}\left[\sum_{t=1}^{T}(F_{\mathrm{CERM}}(\mathbf{w}_t) - F_{\mathrm{CERM}}(\mathbf{w}_*))\right] \leq \frac{1}{2\eta}\|\mathbf{w}_1 - \mathbf{w}_*\|_2^2 + \frac{1}{B}D_\varphi(\boldsymbol{\nu}_*, \boldsymbol{\nu}_0) + \frac{\alpha^2}{T}\cdot\left(\sum_{t=1}^{T}\frac{\eta\sigma_t^2}{2} + \sum_{t=1}^{T}C\delta_t^2\right).$$

Multiplying $1/(\sqrt{T}\alpha)$ on both sides completes the proof. $\qquad\square$

## D. Proof of Results in Section 4

### D.1. Bounds on the Variance Terms

In this section, we present the proof of Lemma 4.2. First, we prove that $e^{\nu_* - \nu}$ is always bounded by the optimality gap $F(\mathbf{w}, \nu) - F(\mathbf{w}, \nu_*)$.

**Lemma D.1** (Self-bounding inequality). *For any $r > 0$, we have $r \leq 2(r - \log r)$. Equivalently, for $r(\nu) := e^{\nu_* - \nu}$ and any $\mathbf{w}, \nu$,*

$$r(\nu) \leq 2\big(F(\mathbf{w}, \nu) - F(\mathbf{w}, \nu_*) + 1\big),$$

*where $\nu_* = \arg\min_\nu F(\mathbf{w}, \nu)$. In the case where $\mathbf{w}$ is fixed as in (13), the notation becomes*

$$r(\nu) \leq 2\big(F(\nu) - F(\nu_*) + 1\big).$$

*Proof.* If $0 < r \leq 2$, then $r \leq 2 \leq 2(r - \log r)$ since $r - \log r \geq 1$ for all $r > 0$. If $r \geq 2$, then $\log r \leq r/2$, hence $r - \log r \geq r/2$, i.e. $r \leq 2(r - \log r)$. This completes the proof for the first part. For the second part, note that the optimality gap can be written as

$$\begin{aligned} F(\mathbf{w}, \nu) - F(\mathbf{w}, \nu_*) &= \mathbb{E}_\zeta[e^{s(\mathbf{w}, \zeta) - \nu}] - \mathbb{E}_\zeta[e^{s(\mathbf{w}, \zeta) - \nu_*}] + \nu - \nu_* \\ &= r(\nu) - 1 - \log r(\nu), \end{aligned}$$

where the last equality comes from the definition of $\nu_*$. Applying the first part with $r = r(\nu)$ completes the proof. $\quad\square$

Then we are ready to prove Lemma 4.2.

*Proof of Lemma 4.2.* We first prove the bound on $\delta_t^2$. Recalling the definitions of $z(\mathbf{w}_t, \zeta_t)$ and $m_t$ in Section 4.1, we get

$$\delta_t^2 = \mathbb{E}_{\zeta_t}\left[e^{-\nu_{t-1}}\big(z(\mathbf{w}_t; \zeta_t) - m_t\big)^2\right] = e^{-\nu_{t-1}}\mathrm{Var}(z(\mathbf{w}_t; \zeta)).$$

By Assumption 4.1 (i), we have $\mathrm{Var}(z(\mathbf{w}_t; \zeta)) \leq (\kappa - 1)m_t^2$. Hence

$$\delta_t^2 \leq (\kappa - 1)e^{-\nu_{t-1}}m_t^2 = (\kappa - 1)m_t \cdot (m_t e^{-\nu_{t-1}}). \tag{33}$$

Let $\tilde{r}_{t-1} = m_t e^{-\nu_{t-1}}$. Then we have

$$F(\mathbf{w}_t, \nu_{t-1}) = \mathbb{E}_{\zeta_t}[e^{s(\mathbf{w}_t; \zeta_t) - \nu_{t-1}}] + \nu_{t-1} = \tilde{r}_{t-1} + \nu_{t-1}.$$

Since $\tilde{r}_{t-1} = \exp(\log m_t - \nu_{t-1})$, with the definition of $\mu$ in Section 4.1, we get

$$F(\mathbf{w}_t, \nu_{t-1}) - (1 + \mu_t) = \tilde{r}_{t-1} + \nu_{t-1} - (1 + \mu_t) = \tilde{r}_{t-1} - \log\tilde{r}_{t-1} - 1.$$

Using Lemma D.1, we have

$$\tilde{r}_{t-1} \leq 2\big(F(\mathbf{w}_t, \nu_{t-1}) - (1 + \mu_t) + 1\big).$$

Since $\mathbf{w}_*$ minimizes $\mu(\mathbf{w})$, we have $\mu_t = \mu(\mathbf{w}_t) \geq \mu(\mathbf{w}_*)$ and thus $(1 + \mu_t) \geq (1 + \mu(\mathbf{w}_*)) = F(\mathbf{w}_*, \nu_*)$, implying

$$F(\mathbf{w}_t, \nu_{t-1}) - (1 + \mu_t) \leq F(\mathbf{w}_t, \nu_{t-1}) - F(\mathbf{w}_*, \nu_*).$$

As a result, we have

$$\tilde{r}_{t-1} \leq 2\big(F(\mathbf{w}_t, \nu_{t-1}) - F(\mathbf{w}_*, \nu_*) + 1\big). \tag{34}$$

Combining (33) with (34), we obtain the desired result on $\delta_t^2$. Next we prove the bound on $\sigma_t^2$. We have

$$\sigma_t^2 = \mathbb{E}_{\zeta_t'}[\|\exp(s(\mathbf{w}_t; \zeta_t') - \nu_t)\nabla s(\mathbf{w}_t; \zeta_t')\|_2^2],$$
$$= \mathbb{E}_{\zeta_t'}[e^{2(\mu_t - \nu_t)}\|\exp(s(\mathbf{w}_t; \zeta_t') - \mu_t)\nabla s(\mathbf{w}_t; \zeta_t')\|_2^2] \leq r_t^2 \sigma'^2,$$

where $r_t = e^{\mu_t - \nu_t}$. Similar to (34), we can show that

$$r_t \leq 2\big(F(\mathbf{w}_t, \nu_t) - F(\mathbf{w}_*, \nu_*) + 1\big).$$

Hence,

$$\sigma_t^2 \leq 4\sigma'^2 \big(F(\mathbf{w}_t, \nu_t) - F(\mathbf{w}_*, \nu_*) + 1\big)^2.$$

Then we complete the proof. $\qquad\square$

### D.2. Convergence Analysis of SPMD for Fixed $\mathbf{w}$

In this section, we present the proof of SPMD when $\mathbf{w}$ is fixed.

*Proof of Theorem 4.3.* By applying Lemma B.4 and Lemma B.5, we obtain the SPMD averaged bound

$$\bar{G}_T := \frac{1}{T}\sum_{t=1}^{T}\mathbb{E}[F(\nu_t) - F(\nu_*)] \leq \frac{D_\varphi(\nu_*, \nu_0)}{\alpha T} + C\,\alpha\,V, \tag{35}$$

where

$$V := \frac{1}{T}\sum_{t=1}^{T}\mathbb{E}[\delta_t^2], \qquad \delta_t^2 = \mathbb{E}\big[e^{-\nu_{t-1}}(z-m)^2\big] = e^{-\nu_{t-1}}\mathrm{Var}(z).$$

Since $e^{-\nu_{t-1}} = r(\nu_{t-1})/m$, we can rewrite $V$ as

$$V = \frac{\mathrm{Var}(z)}{m} \cdot \frac{1}{T}\sum_{t=1}^{T}\mathbb{E}[r(\nu_{t-1})]. \tag{36}$$

From Lemma D.1, we have

$$\frac{1}{T}\sum_{t=1}^{T}\mathbb{E}[r(\nu_{t-1})] \leq \frac{2}{T}\sum_{t=1}^{T}\mathbb{E}[F(\nu_{t-1}) - F(\nu_*) + 1] = 2\left(1 + \frac{1}{T}\sum_{t=1}^{T}\mathbb{E}[F(\nu_{t-1}) - F(\nu_*)]\right).$$

Observing the index shift, we get

$$\sum_{t=1}^{T}\mathbb{E}[F(\nu_{t-1}) - F(\nu_*)] = \mathbb{E}[F(\nu_0) - F(\nu_*)] + \sum_{t=1}^{T-1}\mathbb{E}[F(\nu_t) - F(\nu_*)]$$
$$\leq \mathbb{E}[F(\nu_0) - F(\nu_*)] + \sum_{t=1}^{T}\mathbb{E}[F(\nu_t) - F(\nu_*)].$$

Dividing both sides by $T$ yields

$$\frac{1}{T}\sum_{t=1}^{T}\mathbb{E}[F(\nu_{t-1}) - F(\nu_*)] \leq \frac{\mathbb{E}[F(\nu_0) - F(\nu_*)]}{T} + \bar{G}_T.$$

Combining the above inequality with (36), we have

$$V \leq \frac{2\operatorname{Var}(z)}{m}\left(1 + \bar{G}_T + \frac{\mathbb{E}[F(\nu_0) - F(\nu_*)]}{T}\right). \tag{37}$$

Plugging (37) into (35) yields

$$\bar{G}_T \leq \frac{D_\varphi(\nu_*, \nu_0)}{\alpha T} + \frac{2C\alpha\operatorname{Var}(z)}{m}\left(1 + \bar{G}_T + \frac{\mathbb{E}[F(\nu_0) - F(\nu_*)]}{T}\right).$$

Since $\alpha \leq \frac{m}{4C\operatorname{Var}(z)}$, we have $\frac{2C\alpha\operatorname{Var}(z)}{m} \leq \frac{1}{2}$, and therefore

$$\bar{G}_T \leq \frac{2D_\varphi(\nu_*, \nu_0)}{\alpha T} + \frac{4C\alpha\operatorname{Var}(z)}{m}\left(1 + \frac{\mathbb{E}[F(\nu_0) - F(\nu_*)]}{T}\right)$$

$$\leq \frac{2D_\varphi(\nu_*, \nu_0)}{\alpha T} + \frac{4C\alpha\operatorname{Var}(z)}{m} + \frac{F(\nu_0) - F(\nu_*)}{T}.$$

Optimizing the right-hand side over $\alpha$ (assuming $T$ is large enough) gives the final bound. $\qquad \square$

### D.3. Convergence Analysis of SGD for Fixed w

For completeness of the paper, in this section we present the convergence results of SGD when $\mathbf{w}$ is fixed. Since we consider the projected SGD update (16) in Section 4.3, we consider the following problem and update, which is equivalent to projected SGD:

$$\min_\nu F(\nu) + 1_{[c_0, c_1]}(\nu),$$

where

$$1_{[c_0, c_1]}(\nu) = \begin{cases} 0, & \text{if } \nu \in [c_0, c_1], \\ +\infty, & \text{otherwise.} \end{cases}$$

And the projected SGD update is equivalent to

$$\nu_{t+1} = \arg\min_\nu F'(\nu_t; \zeta_t) \cdot (\nu - \nu_t) + 1_{[c_0, c_1]}(\nu) + \frac{1}{2\alpha_t}(\nu - \nu_t)^2$$
$$= \arg\min_\nu 1_{[c_0, c_1]}(\nu) + \frac{1}{2\alpha_t}(\nu - (\nu_t - \alpha_t F'(\nu_t; \zeta_t)))^2. \tag{38}$$

To see the equivalence between the above update and the projected SGD update, we note that the projected SGD update for minimizing a function $g$ on a set $[c_0, c_1]$ can be written as

$$\nu_{t+1} = \Pi_{[c_0, c_1]}(\nu_t - \alpha_t F'(\nu_t; \zeta_t)) = \arg\min_\nu 1_{[c_0, c_1]}(\nu) + \frac{1}{2\alpha_t}(\nu - (\nu_t - \alpha_t F'(\nu_t; \zeta_t)))^2.$$

Thus the update (38) is equivalent to the projected SGD update. In this section, we will then focus on the convergence analysis of (38). First we present the non-expansiveness property of the update.

**Lemma D.2.** *If $1_{[c_0, c_1]}(\cdot)$ is convex and let*

$$\operatorname{prox}_\alpha(\nu_1) := \arg\min_\nu 1_{[c_0, c_1]}(\nu) + \frac{1}{2\alpha}(\nu - \nu_1)^2,$$

*then we have*

$$|\operatorname{prox}_\alpha(\nu_1) - \operatorname{prox}_\alpha(\nu_2)| \leq |\nu_1 - \nu_2|.$$

*Proof.* By the optimality of $\operatorname{prox}_\alpha(\nu_1)$ and $\operatorname{prox}_\alpha(\nu_1)$ we have

$$u := \frac{\nu_1 - \operatorname{prox}_\alpha(\nu_1)}{\alpha} \in \partial 1_{[c_0, c_1]}(\operatorname{prox}_\alpha(\nu_1))$$

$$v := \frac{\nu_2 - \operatorname{prox}_\alpha(\nu_2)}{\alpha} \in \partial 1_{[c_0, c_1]}(\operatorname{prox}_\alpha(\nu_2)).$$

Since $1_{[c_0,c_1]}(\mathbf{x})$ is convex, we have

$$1_{[c_0,c_1]}(\text{prox}_\alpha(\nu_1)) \geq 1_{[c_0,c_1]}(\text{prox}_\alpha(\nu_2)) + v \cdot (\text{prox}_\alpha(\nu_1) - \text{prox}_\alpha(\nu_2))$$
$$1_{[c_0,c_1]}(\text{prox}_\alpha(\nu_2)) \geq 1_{[c_0,c_1]}(\text{prox}_\alpha(\nu_1)) + u \cdot (\text{prox}_\alpha(\nu_2) - \text{prox}_\alpha(\nu_1)).$$

Adding them together, we have

$$0 \geq (v - u) \cdot (\text{prox}_\alpha(\nu_1) - \text{prox}_\alpha(\nu_2))$$
$$= -\frac{1}{\alpha}(\nu_1 - \nu_2 + \text{prox}_\alpha(\nu_2) - \text{prox}_\alpha(\nu_1)) \cdot (\text{prox}_\alpha(\nu_1) - \text{prox}_\alpha(\nu_2)).$$

which implies

$$\frac{1}{\alpha}(\text{prox}_\alpha(\nu_1) - \text{prox}_\alpha(\nu_2))^2 \leq \frac{1}{\alpha}(\nu_1 - \nu_2) \cdot (\text{prox}_\alpha(\nu_1) - \text{prox}_\alpha(\nu_2))$$
$$\leq \frac{1}{\alpha}|\nu_1 - \nu_2| \cdot |\text{prox}_\alpha(\nu_1) - \text{prox}_\alpha(\nu_2)|$$

Thus $|\text{prox}_\alpha(\nu_1) - \text{prox}_\alpha(\nu_2)| \leq |\nu_1 - \nu_2|$. □

Before presenting the proof for the convergence of the projected SGD update, we first present the proof of Lemma 4.4.

*Proof of Lemma 4.4.* We have $F''(\nu) = me^{-\nu}$, which is decreasing in $\nu$, so the maximum over $[c_0, c_1]$ is attained at $c_0$. □

Then we are ready to prove the convergence of the projected SGD update (16), which is equivalent to the update (38).

*Proof of Theorem 4.5.* By the first-order optimality condition of (38), for any $\nu$ we have

$$(F'(\nu_t; \zeta_t) + \partial 1_{[c_0,c_1]}(\nu_{t+1}) + \frac{1}{\alpha_t}(\nu_{t+1} - \nu_t)) \cdot (\nu - \nu_{t+1}) \geq 0.$$

By the convexity of $1_{[c_0,c_1]}$, we have

$$1_{[c_0,c_1]}(\nu_{t+1}) \leq 1_{[c_0,c_1]}(\nu) + \partial 1_{[c_0,c_1]}(\nu_{t+1}) \cdot (\nu_{t+1} - \nu).$$

Adding the above two inequalities, we have

$$F'(\nu_t) \cdot (\nu_{t+1} - \nu) + 1_{[c_0,c_1]}(\nu_{t+1}) - 1_{[c_0,c_1]}(\nu) \leq \frac{1}{\alpha_t}(\nu_t - \nu_{t+1}) \cdot (\nu_{t+1} - \nu)$$
$$= \frac{1}{2\alpha_t}((\nu_t - \nu)^2 - (\nu_{t+1} - \nu)^2 - (\nu_t - \nu_{t+1})^2). \tag{39}$$

where the equality uses the fact that $2(a - b) \cdot (b - c) = (a - c)^2 - (b - c)^2 - (a - b)^2$. By Lemma 4.4, we have

$$F(\nu_{t+1}) \leq F(\nu_t) + F'(\nu_t) \cdot (\nu_{t+1} - \nu_t) + \frac{L}{2}(\nu_{t+1} - \nu_t)^2.$$

By the convexity of $F$, we have

$$F(\nu_t) \leq F(\nu) + F'(\nu_t) \cdot (\nu_t - \nu).$$

Adding the above two inequalities, we have

$$F(\nu_{t+1}) \leq F(\nu) + F'(\nu_t) \cdot (\nu_{t+1} - \nu) + \frac{L}{2}(\nu_{t+1} - \nu_t)^2.$$

Note that $1_{[c_0,c_1]}(\nu_*) = 0$ and $1_{[c_0,c_1]}(\nu_t) = 0, \forall t$. Combining the above inequality with (39), and setting $\nu = \nu_*$, we have

$$F(\nu_{t+1}) - F(\nu_*) \leq \frac{1}{2\alpha_t}((\nu_t - \nu_*)^2 - (\nu_{t+1} - \nu_*)^2 - (\nu_t - \nu_{t+1})^2) + \frac{L}{2}(\nu_{t+1} - \nu_t)^2$$
$$+ (F'(\nu_t) - F'(\nu_t; \zeta_t)) \cdot (\nu_{t+1} - \nu_*). \tag{40}$$

Define

$$\hat{\nu}_{t+1} = \arg\min_\nu \frac{1}{2\alpha_t}(\nu - (\nu_t - \alpha_t F'(\nu_t)))^2 + 1_{[c_0,c_1]}(\nu).$$

Then we can bound the expectation of last term on the RHS of (40):

$$
\begin{aligned}
\mathbb{E}[(F'(\nu_t) - F'(\nu_t; \zeta_t)) \cdot (\nu_{t+1} - \nu_*)] &= \mathbb{E}[(F'(\nu_t) - F'(\nu_t; \zeta_t)) \cdot (\nu_{t+1} - \hat{\nu}_{t+1} + \hat{\nu}_{t+1} - \nu_*)] \\
&= \mathbb{E}[(F'(\nu_t) - F'(\nu_t; \zeta_t)) \cdot (\nu_{t+1} - \hat{\nu}_{t+1})] \\
&\le \alpha_t \mathbb{E}[(F'(\nu_t) - F'(\nu_t, \zeta_t))^2] = \alpha_t \sigma_t^2,
\end{aligned} \tag{41}
$$

where the inequality is due to Lemma D.2. Taking expectation of (40) and plugging in (41), we get

$$
\mathbb{E}[F(\nu_{t+1}) - F(\nu_*)] \le \frac{1}{2\alpha_t}(\nu_t - \nu_*)^2 - \frac{1}{2\alpha_t}(\nu_{t+1} - \nu_*)^2 - \left(\frac{1}{2\alpha_t} - \frac{L}{2}\right)(\nu_t - \nu_{t+1})^2 + \alpha_t \sigma_t^2.
$$

Telescoping the sum for $t = 0, \ldots, T-1$, and noting that $\alpha_t = \alpha' \le 1/L$, we get

$$
\sum_{t=0}^{T-1} \mathbb{E}[F(\nu_{t+1}) - F(\nu_*)] \le \frac{(\nu_0 - \nu_*)^2}{2\alpha'} + \alpha' \sum_{t=0}^{T-1} \sigma_t^2.
$$

Dividing both sides by $T$, and from the definition of $\bar{\nu}_T$ and the convexity of $F$, we have

$$
\frac{1}{T} \sum_{t=1}^{T} \mathbb{E}[F(\nu_t) - F(\nu_*)] \le \frac{(\nu_0 - \nu_*)^2}{2\alpha' T} + \alpha' V',
$$

where

$$
V' = \frac{\alpha'}{T} \sum_{t=0}^{T-1} \sigma_t^2 = \frac{\mathrm{Var}(z)}{T} \sum_{t=0}^{T-1} \mathbb{E}[e^{-2\nu_t}] \le \mathrm{Var}(z) e^{-2c_0}.
$$

We finish the proof by noting that $\mathrm{Var}(z)e^{-2c_0} = m^2(\kappa - 1)e^{-2c_0} = e^{2(\nu_* - c_0)}(\kappa - 1)$ and optimizing the upper bound over $\alpha'$. $\qquad \square$

## E. A Distribution-free Lower Bound and Matching Upper Bound of SPMD

In this section, we present a lower bound on the complexity of algorithms solving (13). Then we show that with a specific choice of the learning rate, the convergence of SPMD matches the lower bound.

### E.1. A Distribution-free Lower Bound

We consider an optimal bound for a black-box oracle model where the underlying distribution of $z$ is unknown and for any query $\nu$ the oracle returns

$$
\Phi(\nu; \zeta) = ze^{-\nu} + \nu, \qquad g(\nu; \zeta) = \nabla_\nu \Phi(\nu; \zeta) = 1 - ze^{-\nu}.
$$

Since

$$
z = e^\nu(\Phi(\nu; \zeta) - \nu) = e^\nu(1 - g(\nu; \zeta)),
$$

any $T$-query algorithm can reconstruct $T$ i.i.d. samples $z_1, \ldots, z_T$ from $P$. Thus, it suffices to prove the lower bound in the standard i.i.d. sampling model for $z$. We first present three lemmas that are useful for our proof.

**Lemma E.1.** *Let* $\phi(u) := e^{-u} + u - 1$. *Then* $\phi(0) = \phi'(0) = 0$ *and* $\phi''(u) = e^{-u}$. *In particular, for all* $|u| \le 1$,

$$
\phi(u) \ge \frac{e^{-1}}{2} u^2.
$$

*Proof.* On the interval $[-1, 1]$, $\phi''(u) = e^{-u} \ge e^{-1}$, so $\phi$ is $e^{-1}$-strongly convex on $[-1, 1]$. Since $\phi(0) = \phi'(0) = 0$, strong convexity implies $\phi(u) \ge \frac{e^{-1}}{2}u^2$ for all $|u| \le 1$. $\qquad \square$

**Lemma E.2.** *Let* $\phi(u) = e^{-u} + u - 1$. *Fix* $\nu_0 < \nu_1$ *and let* $\Delta := \nu_1 - \nu_0$. *Define*

$$
H(\nu) := \phi(\nu - \nu_0) + \phi(\nu - \nu_1).
$$

*Then $H$ is strictly convex and its unique minimizer $\nu^\dagger$ lies in $(\nu_0, \nu_1)$. Moreover, if $\Delta \le 1$, then*

$$
\inf_{\nu \in \mathbb{R}} H(\nu) \ge \frac{e^{-1}}{4} \Delta^2.
$$

*Proof.* From Lemma E.1 we know $H$ is strictly convex with

$$H'(\nu) = \phi'(\nu - \nu_0) + \phi'(\nu - \nu_1) = 2 - e^{-(\nu - \nu_0)} - e^{-(\nu - \nu_1)}.$$

At the endpoints,

$$H'(\nu_0) = 2 - 1 - e^{-(\nu_0 - \nu_1)} = 1 - e^{\Delta} < 0, \qquad H'(\nu_1) = 2 - e^{-(\nu_1 - \nu_0)} - 1 = 1 - e^{-\Delta} > 0.$$

Since $H'$ is strictly increasing (because $H'' > 0$), there is a unique root $\nu^\dagger \in (\nu_0, \nu_1)$ and thus $\inf_{\nu \in \mathbb{R}} H(\nu) = \inf_{\nu \in [\nu_0, \nu_1]} H(\nu)$. Assume $\Delta \leq 1$. Then for all $\nu \in [\nu_0, \nu_1]$ we have $|\nu - \nu_0| \leq \Delta \leq 1$ and $|\nu - \nu_1| \leq \Delta \leq 1$. Applying Lemma E.1, we know that for all $\nu \in [\nu_0, \nu_1]$,

$$H(\nu) \geq \frac{e^{-1}}{2}\big((\nu - \nu_0)^2 + (\nu - \nu_1)^2\big).$$

Minimizing the right-hand-side over $\nu$ yields $\inf_\nu \big((\nu - \nu_0)^2 + (\nu - \nu_1)^2\big) = \Delta^2/2$, this completes the proof. $\square$

**Lemma E.3** (Le Cam's Two-point Method). *Let $P_0, P_1$ be two distributions and let $L_0(\cdot), L_1(\cdot)$ be nonnegative loss functions. For any estimator $\widehat{a}$ measurable w.r.t. the data,*

$$\max\{\mathbb{E}_{P_0}[L_0(\widehat{a})], \, \mathbb{E}_{P_1}[L_1(\widehat{a})]\} \geq \frac{1 - \mathrm{TV}(P_0, P_1)}{2} \inf_a \big(L_0(a) + L_1(a)\big),$$

*where* TV *is the total variation distance.*

*Proof.* Let $M := (P_0 + P_1)/2$ and write $dP_0 = (1 + f)\, dM$, $dP_1 = (1 - f)\, dM$ where $|f| \leq 1$ and $\int |f|\, dM = \mathrm{TV}(P_0, P_1)$. Then for any (possibly random) decision $A$,

$$\begin{aligned}
\mathbb{E}_{P_0}[L_0(A)] + \mathbb{E}_{P_1}[L_1(A)] &= \int \Big(L_0(A)(1 + f) + L_1(A)(1 - f)\Big)\, dM \\
&= \int \Big((L_0(A) + L_1(A)) + f(L_0(A) - L_1(A))\Big)\, dM \\
&\geq \int \Big((L_0(A) + L_1(A)) - |f|\,(L_0(A) + L_1(A))\Big)\, dM \\
&= \int (L_0(A) + L_1(A))(1 - |f|)\, dM \\
&\geq \inf_a (L_0(a) + L_1(a)) \int (1 - |f|)\, dM \\
&= (1 - \mathrm{TV}(P_0, P_1)) \inf_a (L_0(a) + L_1(a)).
\end{aligned}$$

Taking half and using $\max\{x, y\} \geq (x + y)/2$ completes the proof. $\square$

The final distribution-free suboptimality lower bound is stated in the following theorem.

**Theorem E.4.** *Let $z = e^{s(\zeta)} \geq 0$ with $m(P) = \mathbb{E}_P[z]$ and $\nu_*(P) = \log m(P)$. For $\kappa \geq 2$, define*

$$\mathcal{P}_\kappa := \left\{ P : \; z \geq 0, \; 0 < \mathbb{E}_P[z] < \infty, \; \frac{\mathbb{E}_P[z^2]}{\mathbb{E}_P[z]^2} \leq \kappa \right\}.$$

*Let $F_P(\nu) := m(P)e^{-\nu} + \nu$ and $\nu_*(P) = \arg\min_\nu F_P(\nu)$. Then there exists an absolute constant $c > 0$ such that for all $T \geq \kappa$, any (possibly adaptive) algorithm using $T$ value/gradient oracle calls and outputting $\widehat{\nu}$ satisfies*

$$\sup_{P \in \mathcal{P}_\kappa} \mathbb{E}_P[F_P(\widehat{\nu}) - F_P(\nu_*(P))] \geq c\, \frac{\kappa - 1}{T}. \tag{42}$$

*Proof.* We construct two strictly positive hard instances in $\mathcal{P}_\kappa$. Fix $\varepsilon \in (0, 1]$ and define two distributions supported on $\{\varepsilon, \kappa\}$:

$$P_i^\varepsilon : \quad \mathbb{P}(z = \kappa) = p_i, \qquad \mathbb{P}(z = \varepsilon) = 1 - p_i, \qquad i \in \{0, 1\},$$

where

$$p_0 := \frac{1}{\kappa}, \qquad p_1 := p_0 + h, \qquad h := \frac{1}{8\sqrt{\kappa T}}.$$

Since $T \geq \kappa$, we have $h \leq \frac{1}{8\kappa}$ so $p_1 \in (0,1)$. Next we show that $P_0^\varepsilon, P_1^\varepsilon \in \mathcal{P}_\kappa$. For a generic $p \in (0,1)$ and support $\{\varepsilon, \kappa\}$, define

$$R_\varepsilon(p) := \frac{\mathbb{E}[z^2]}{\mathbb{E}[z]^2} = \frac{p\kappa^2 + (1-p)\varepsilon^2}{\big(p\kappa + (1-p)\varepsilon\big)^2}.$$

Let $u := \varepsilon/\kappa \in (0, 1/\kappa] \subset (0,1]$. Then

$$R_\varepsilon(p) = \frac{p + (1-p)u^2}{\big(p + (1-p)u\big)^2}.$$

We claim $R_\varepsilon(p) \leq \frac{1}{p}$ for all $u \in [0,1]$. Indeed,

$$\big(p + (1-p)u\big)^2 - p\big(p + (1-p)u^2\big) = p^2 + 2p(1-p)u + (1-p)^2u^2 - p^2 - p(1-p)u^2$$

$$= (1-p)u\Big(2p + (1-2p)u\Big) \geq 0,$$

since $u \in [0,1]$ and $2p + (1-2p)u \geq \min\{2p, 1\} \geq 0$. Thus $R_\varepsilon(p) \leq 1/p$. Since $p_0 = 1/\kappa$ and $p_1 \geq p_0$, we have $1/p_i \leq \kappa$, hence $R_\varepsilon(p_i) \leq \kappa$ and therefore $P_0^\varepsilon, P_1^\varepsilon \in \mathcal{P}_\kappa$. Next, we compute the separation $\Delta$ between $\nu_*$'s. Let $m_i^\varepsilon = \mathbb{E}_{P_i^\varepsilon}[z] = \varepsilon + p_i(\kappa - \varepsilon)$ and $\nu_i^\varepsilon = \log m_i^\varepsilon$. Then

$$m_1^\varepsilon - m_0^\varepsilon = h(\kappa - \varepsilon) \geq h(\kappa - 1), \qquad m_0^\varepsilon = \varepsilon + p_0(\kappa - \varepsilon) = 1 + \Big(1 - \frac{1}{\kappa}\Big)\varepsilon \in [1, 2].$$

Hence

$$\Delta := |\nu_1^\varepsilon - \nu_0^\varepsilon| = \log\Big(1 + \frac{m_1^\varepsilon - m_0^\varepsilon}{m_0^\varepsilon}\Big) \geq \frac{1}{2} \cdot \frac{h(\kappa - 1)}{2} = \frac{\kappa - 1}{32\sqrt{\kappa T}},$$

where we used $\log(1 + x) \geq x/2$ for $x \in [0, 1/2]$ and the fact that $\frac{h(\kappa - \varepsilon)}{m_0^\varepsilon} \leq h\kappa \leq 1/8$. In particular, $\Delta \leq h\kappa \leq 1/8 < 1$. Next, we show the lower bound of $\inf_\nu \Big((F_0(\nu) - F_0(\nu_0^\varepsilon)) + (F_1(\nu) - F_1(\nu_1^\varepsilon))\Big)$. Under $P_i^\varepsilon$ the objective is $F_i(\nu) = m_i^\varepsilon e^{-\nu} + \nu$ and the optimal value is $F_i(\nu_i^\varepsilon) = 1 + \nu_i^\varepsilon$. Thus the suboptimality can be written as

$$F_i(\nu) - F_i(\nu_i^\varepsilon) = e^{\nu_i^\varepsilon - \nu} + (\nu - \nu_i^\varepsilon) - 1 = \phi(\nu - \nu_i^\varepsilon), \qquad \phi(u) = e^{-u} + u - 1.$$

Let $\nu_0^\varepsilon < \nu_1^\varepsilon$ and set $u = \nu - \nu_0^\varepsilon$. Then

$$\phi(\nu - \nu_0^\varepsilon) + \phi(\nu - \nu_1^\varepsilon) = \phi(u) + \phi(u - \Delta).$$

The function $u \mapsto \phi(u) + \phi(u - \Delta)$ is convex and its minimizer lies in $[0, \Delta]$. Since $\Delta \leq 1$, applying Lemma E.2 gives

$$\phi(u) + \phi(u - \Delta) \geq \frac{e^{-1}}{4}\Delta^2.$$

Therefore,

$$\inf_\nu \Big((F_0(\nu) - F_0(\nu_0^\varepsilon)) + (F_1(\nu) - F_1(\nu_1^\varepsilon))\Big) \geq \frac{e^{-1}}{4}\Delta^2. \tag{43}$$

Next, we show the total variation between $P_0^\varepsilon$, and $P_1^\varepsilon$ is bounded. Because the two distributions differ only in the Bernoulli parameter,

$$\mathrm{KL}(P_0^\varepsilon, P_1^\varepsilon) = p_0 \log \frac{p_0}{p_1} + (1 - p_0) \log \frac{1 - p_0}{1 - p_1}.$$

Using the bound $\mathrm{KL}(P, Q) \leq \chi^2(P, Q)$ and the fact that for Bernoulli measures $\chi^2(P_0^\varepsilon, P_1^\varepsilon) = \frac{h^2}{p_1(1 - p_1)}$, we get

$$\mathrm{KL}(P_0^\varepsilon, P_1^\varepsilon) \leq \frac{h^2}{p_1(1 - p_1)}.$$

Since $h \leq \frac{1}{2\kappa}$, we have $p_1 \leq p_0 + h \leq \frac{3}{2\kappa} \leq \frac{3}{4}$, hence $1 - p_1 \geq 1/4$, and also $p_1 \geq p_0 = 1/\kappa$. Therefore $p_1(1 - p_1) \geq \frac{1}{4\kappa}$ and

$$\mathrm{KL}(P_0^\varepsilon, P_1^\varepsilon) \leq 4\kappa h^2.$$

For $T$ i.i.d. samples, this gives

$$\mathrm{KL}\big((P_0^\varepsilon)^{\otimes T}, (P_1^\varepsilon)^{\otimes T}\big) = T\,\mathrm{KL}(P_0^\varepsilon, P_1^\varepsilon) \leq 4\kappa T h^2 = \frac{1}{16}.$$

By Pinsker's inequality,

$$\text{TV}\big((P_0^\varepsilon)^{\otimes T}, (P_1^\varepsilon)^{\otimes T}\big) \le \sqrt{\frac{1}{2}\text{KL}\big((P_0^\varepsilon)^{\otimes T}, (P_1^\varepsilon)^{\otimes T}\big)} \le \sqrt{\frac{1}{32}} \le \frac{1}{4}.$$

Finally, we apply Lemma E.3 to $P_0 = (P_0^\varepsilon)^{\otimes T}$, $P_1 = (P_1^\varepsilon)^{\otimes T}$ and losses

$$L_i(\nu) := F_i(\nu) - F_i(\nu_i^\varepsilon) \ge 0.$$

Using (43) and $\text{TV} \le 1/4$ yields for any estimator $\widehat{\nu}$,

$$\max_{i\in\{0,1\}} \mathbb{E}_{P_i^\varepsilon}[F_i(\widehat{\nu}) - F_i(\nu_i^\varepsilon)] \ge \frac{1-\text{TV}}{2} \cdot \frac{e^{-1}}{4}\Delta^2 \ge \frac{3}{8} \cdot \frac{e^{-1}}{4}\Delta^2 = \frac{3e^{-1}}{32}\Delta^2.$$

Substituting $\Delta^2 \ge \frac{(\kappa-1)^2}{1024\,\kappa\,T} \ge \frac{\kappa-1}{2048\,T}$ (since $\kappa \ge 2$) gives

$$\max_{i\in\{0,1\}} \mathbb{E}_{P_i^\varepsilon}[F_i(\widehat{\nu}) - F_i(\nu_i^\varepsilon)] \ge \frac{3}{65536\,e} \cdot \frac{\kappa-1}{T}.$$

Since $P_0^\varepsilon, P_1^\varepsilon \in \mathcal{P}_\kappa$, this implies (42) with $c = \frac{3}{65536\,e}$. Then we complete the proof. $\square$

### E.2. An Optimal Bound for SPMD

In fact, we can improve the convergence rate of SPMD to $O\left(\frac{\kappa-1}{T}\right)$, which matches the lower bound established above. The key is to use a specially designed learning rate scheme $\alpha_t$. Recall the SPMD update in Lemma B.2:

$$\pi_t = \frac{\pi_{t-1} + \alpha_t}{1 + \alpha_t z_t}, \tag{44}$$

where $\pi_{t-1} = e^{-\nu_{t-1}}$, $z_t = e^{s(\zeta_t)}$. We focus on the case where $s(\zeta)$ follows a subgaussian distribution.

**Assumption E.5.** $s(\zeta)$ is $\sigma^2$-subgaussian, i.e.,

$$\mathbb{E}\big[e^{\lambda(s(\zeta)-\mathbb{E}[s(\zeta)])}\big] \le e^{\lambda^2\sigma^2/2} \quad \forall\lambda \in \mathbb{R}.$$

The following lemma indicates that with our specific choice of the learning rate, $\nu_t$ is the exact minimizer of an empirical objective.

**Lemma E.6.** Let $S_t := \sum_{i=1}^{t} z_i$ and $\bar{z}_t := S_t/t$. Initialize $\pi_1 = 1/z_1$ (or equivalently $\alpha_1 = \infty$) and for $t \ge 2$ choose

$$\alpha_t := \frac{\pi_{t-1}}{t-1} = \frac{1}{S_{t-1}}. \tag{45}$$

Then for all $t \ge 1$,

$$\pi_t = \frac{t}{S_t}, \qquad \nu_t = -\log \pi_t = \log\left(\frac{S_t}{t}\right) = \log \bar{z}_t. \tag{46}$$

In particular, $\nu_t$ is the exact minimizer of the empirical objective

$$\widehat{F}_t(\nu) := \bar{z}_t e^{-\nu} + \nu \quad \text{since} \quad \arg\min_\nu \widehat{F}_t(\nu) = \log \bar{z}_t.$$

*Proof.* We prove (46) by induction. For $t = 1$, $\pi_1 = 1/z_1 = 1/S_1$ holds by initialization. Assume $\pi_{t-1} = (t-1)/S_{t-1}$. Then (45) gives $\alpha_t = 1/S_{t-1}$, and the recursion (44) yields

$$\pi_t = \frac{\frac{t-1}{S_{t-1}} + \frac{1}{S_{t-1}}}{1 + \frac{z_t}{S_{t-1}}} = \frac{\frac{t}{S_{t-1}}}{\frac{S_{t-1}+z_t}{S_{t-1}}} = \frac{t}{S_{t-1} + z_t} = \frac{t}{S_t}.$$

Thus $\pi_t = t/S_t$ and $\nu_t = -\log \pi_t = \log(S_t/t) = \log \bar{z}_t$. This completes the proof. $\square$

Since $\frac{\text{Var}(z)}{(\mathbb{E}[z])^2} = \kappa - 1$, we have

$$\text{Var}(\bar{z}_T) = \frac{\text{Var}(z)}{T} = \frac{(\kappa-1)m^2}{T}.$$

Since Lemma E.6 gives $\nu_T = \log \bar{z}_T$, in light of Lemma D.1 we can write

$$F(\nu_T) - F(\nu_*) = \frac{m}{\bar{z}_T} - 1 + \log\left(\frac{\bar{z}_T}{m}\right) = \frac{1}{Q_T} + \log Q_T - 1, \qquad Q_T := \frac{\bar{z}_T}{m}. \tag{47}$$

Note that $\mathbb{E}[Q_T] = 1$ and $\mathrm{Var}(Q_T) = (\kappa - 1)/T$. Let $U_T := Q_T - 1 = (\bar{z}_T - m)/m$. Then $\mathbb{E}[U_T] = 0$ and $\mathbb{E}[U_T^2] = (\kappa - 1)/T$. Define

$$g(u) := \frac{1}{1+u} + \log(1+u) - 1, \forall u > -1$$

so that by (47) we have $F(\nu_T) - F(\nu_*) = g(U_T)$. Next we present three lemmas that help prove an upper bound on $g$.

**Lemma E.7.** *For all $u \geq -\frac{1}{2}$,*

$$g(u) \leq 2u^2.$$

*Proof.* Define $h(u) := 2u^2 - g(u)$ for $u > -1$. Since $g'(u) = \frac{u}{(1+u)^2}$, we have

$$h'(u) = 4u - \frac{u}{(1+u)^2} = u\left(4 - \frac{1}{(1+u)^2}\right).$$

For $u \geq -\frac{1}{2}$, $(1+u)^2 \geq \frac{1}{4}$, hence $\frac{1}{(1+u)^2} \leq 4$. Therefore $h'(u) \leq 0$ for $u \in [-\frac{1}{2}, 0]$ and $h'(u) \geq 0$ for $u \geq 0$. Thus $h$ attains its minimum over $[-\frac{1}{2}, \infty)$ at $u = 0$, where $h(0) = 0$. Hence $h(u) \geq 0$ on $[-\frac{1}{2}, \infty)$, i.e., $g(u) \leq 2u^2$. This completes the proof. $\square$

**Lemma E.8.** *Let $z_i \geq 0$ i.i.d. with finite $\kappa$. Then*

$$\mathbb{P}(Q_T \leq 1/2) = \mathbb{P}(\bar{z}_T \leq m/2) \leq e^{-T/(8\kappa)}.$$

*Proof.* For any $\lambda > 0$, by the Chernoff bound, we have

$$\mathbb{P}\left(\sum_{i=1}^{T} z_i \leq \frac{Tm}{2}\right) = \mathbb{P}\left(e^{-\lambda \sum_{i=1}^{T} z_i} \geq e^{-\lambda Tm/2}\right) \leq e^{\lambda Tm/2}\left(\mathbb{E}[e^{-\lambda z}]\right)^T.$$

Using $e^{-x} \leq 1 - x + x^2/2$ for $x \geq 0$,

$$\mathbb{E}[e^{-\lambda z}] \leq 1 - \lambda m + \frac{\lambda^2}{2}\mathbb{E}[z^2] \leq \exp\left(-\lambda m + \frac{\lambda^2}{2}\mathbb{E}[z^2]\right).$$

Therefore

$$\mathbb{P}(\bar{z}_T \leq m/2) \leq \exp\left(T\left(\lambda m/2 - \lambda m + \frac{\lambda^2}{2}\mathbb{E}[z^2]\right)\right) = \exp\left(-T\left(\frac{\lambda m}{2} - \frac{\lambda^2}{2}\mathbb{E}[z^2]\right)\right).$$

Choosing $\lambda = m/(2\mathbb{E}[z^2])$, we get $-Tm^2/(8\mathbb{E}[z^2]) = -T/(8\kappa)$. This completes the proof. $\square$

**Lemma E.9.** *If $s$ is $\sigma^2$-subgaussian, then*

$$m^2 \mathbb{E}[z^{-2}] = (\mathbb{E}[e^s])^2 \mathbb{E}[e^{-2s}] \leq e^{3\sigma^2}.$$

*Proof.* Let $\mu = \mathbb{E}[s]$ and $X = s - \mu$. Then $\mathbb{E}[X] = 0$ and $z = e^s = e^\mu e^X$. Thus

$$m^2\mathbb{E}[z^{-2}] = \left(e^\mu \mathbb{E}[e^X]\right)^2 \cdot \left(e^{-2\mu}\mathbb{E}[e^{-2X}]\right) = \left(\mathbb{E}[e^X]\right)^2 \mathbb{E}[e^{-2X}].$$

By subgaussianity,

$$\mathbb{E}[e^X] \leq e^{\sigma^2/2}, \qquad \mathbb{E}[e^{-2X}] \leq e^{(2^2)\sigma^2/2} = e^{2\sigma^2}.$$

Hence $m^2\mathbb{E}[z^{-2}] \leq e^{\sigma^2}e^{2\sigma^2} = e^{3\sigma^2}$. This completes the proof. $\square$

Then we are ready to prove the convergence of SPMD with our specific choice of learning rate.

**Theorem E.10.** *Under Assumption E.5, the SPMD iterate $\nu_T$ produced by $\alpha_t = \pi_{t-1}/(t-1)$ satisfies*

$$\mathbb{E}\left[F(\nu_T) - F(\nu_*)\right] \leq \frac{2(\kappa - 1)}{T} + \exp\left(\frac{3}{2}\sigma^2 - \frac{T}{16\kappa}\right).$$

*In particular, since the second term is exponentially small in $T/\kappa$, and we have*

$$\mathbb{E}\big[F(\nu_T) - F(\nu_*)\big] = O(\kappa/T),$$

*for every $\sigma^2$-subgaussian $s(\zeta)$.*

*Proof.* Since $F(\nu_T) - F(\nu_*) = g(U_T)$, we split the expectation on the events $\{U_T \geq -1/2\}$ and $\{U_T < -1/2\}$:

$$\mathbb{E}[g(U_T)] = \mathbb{E}[g(U_T)\mathbf{1}\{U_T \geq -1/2\}] + \mathbb{E}[g(U_T)\mathbf{1}\{U_T < -1/2\}].$$

On $\{U_T \geq -1/2\}$, Lemma E.7 yields

$$\mathbb{E}[g(U_T)\mathbf{1}\{U_T \geq -1/2\}] \leq 2\,\mathbb{E}[U_T^2] = 2\,\mathrm{Var}(Q_T) = 2\,\frac{\mathrm{Var}(z)}{m^2 T} = \frac{2(\kappa - 1)}{T}. \tag{48}$$

On $\{U_T < -1/2\}$ we have $Q_T \leq 1/2$, and since $\log Q_T - 1 \leq 0$,

$$g(U_T) = \frac{1}{Q_T} + \log Q_T - 1 \leq \frac{1}{Q_T}.$$

Hence, by Cauchy–Schwarz inequality, we have

$$\mathbb{E}[g(U_T)\mathbf{1}\{U_T < -1/2\}] \leq \mathbb{E}[Q_T^{-1}\mathbf{1}\{Q_T \leq 1/2\}] \leq \big(\mathbb{E}[Q_T^{-2}]\big)^{1/2}\,\mathbb{P}(Q_T \leq 1/2)^{1/2}.$$

By Jensen inequality and Lemma E.9,

$$\mathbb{E}[Q_T^{-2}] = m^2\,\mathbb{E}[\bar{z}_T^{-2}] \leq m^2\,\mathbb{E}[z^{-2}] \leq e^{3\sigma^2}.$$

By Lemma E.8, $\mathbb{P}(Q_T \leq 1/2) \leq \exp(-T/(8\kappa))$. Therefore,

$$\mathbb{E}[g(U_T)\mathbf{1}\{U_T < -1/2\}] \leq \exp\left(\frac{3}{2}\sigma^2 - \frac{T}{16\kappa}\right). \tag{49}$$

Combining (48) and (49), we complete the proof. $\square$

# F. Additional Experiment Results

In this section, we present additional experiment results. In Appendix F.1, we present more results on extreme classification, partial AUC maximization, and the comparison between SGD and SPMD. And in Appendices F.2 and F.3, we present experiment results on CLIP training and KL-regularized distributionally robust optimization, respectively. Finally, we present the implementation details and hyperparameter choices in Appendix F.4.

## F.1. Supplementary Results for Sections 4 and 5

**SGD with momentum optimizer**. We conduct additional experiments on extreme classification and partial AUC maximization using the SGD with momentum optimizer. We apply the same hyperparameter tuning process for all methods as the SGD optimizer. We present the results in Figures 4 and 5, and we observe similar trend as the SGD optimizer in Section 5.

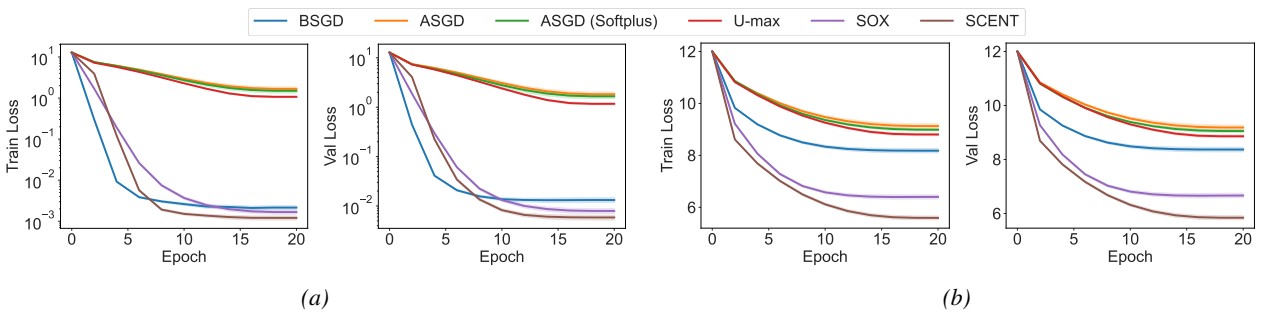

*Figure 4.* (a): cross-entropy loss of different methods on the training set (left) and validation dataset (right) of Glint360K, using SGD with momentum optimizer. (b): cross-entropy loss of different methods on the training set (left) and validation dataset (right) of TreeOfLife-10M, using SGD with momentum optimizer.

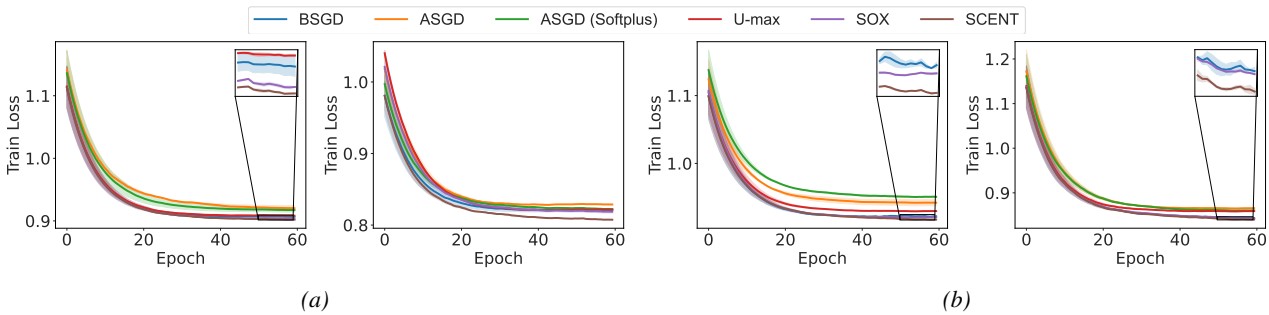

*Figure 5.* Training loss curves of different methods using SGD with momentum optimizer for partial AUC maximization. (a): on the dataset CIFAR-10 with $\tau = 0.05$ (left) and $\tau = 0.1$. (b): on the dataset CIFAR-100 with $\tau = 0.05$ (left) and $\tau = 0.1$.

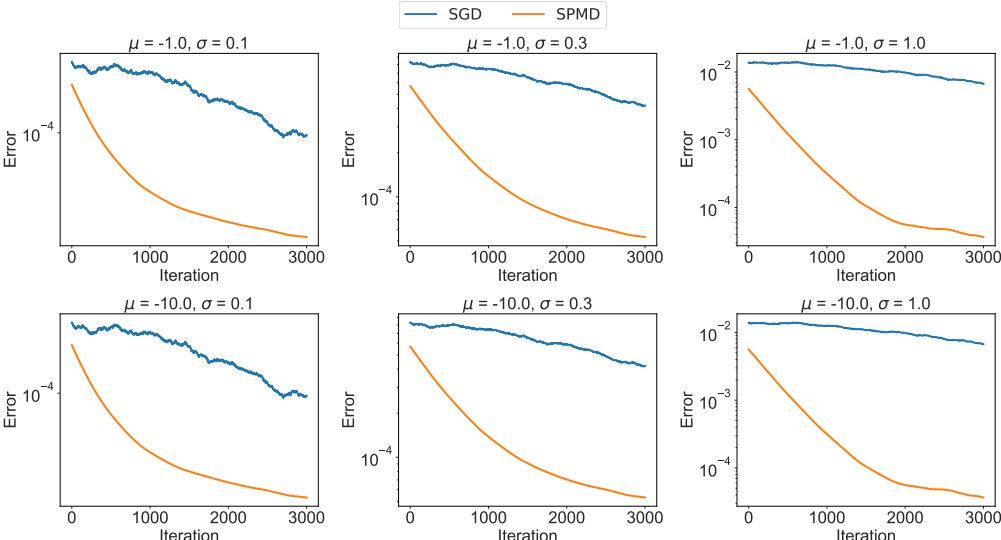

*Figure 6.* Error between $\nu_t$ and $\nu_*$ when trained using different methods on Gaussian noise with different mean (top to bottom: $\mu = -1.0, -10.0$) and standard deviation (left to right: $\sigma = 0.1, 0.3, 1.0$)

**Comparison between SGD and SPMD with fixed w**. In Figure 1 we present the ratio between the error of SPMD and that of SGD when they are run on Gaussian noise with different means and variances. Here in Figure 6, we plot the value of the error of the two methods that are used to compute the ratio.

### F.2. CLIP Training

We apply our method to image-text representation learning tasks, namely CLIP (Radford et al., 2021). Given a dataset of image-text pairs $\mathcal{S} = \{(\mathbf{x}_1, \mathbf{y}_1), \ldots, (\mathbf{x}_n, \mathbf{y}_n)\}$, CLIP aims to train a model $h$ (parameterized by $\mathbf{w}$) that learns the representation of images and texts. In this paper, we consider the Robust Global Contrastive Loss (Wei et al., 2024):

$$\min_{\mathbf{w} \in \mathbb{R}^d, \tau \in \mathbb{R}} \tau \cdot \frac{1}{|\mathcal{S}|} \sum_{i \in \mathcal{S}} \log \left( \varepsilon + \frac{1}{|\mathcal{S}| - 1} \sum_{j \in \mathcal{S}, j \neq i} \exp \left( \frac{h(\mathbf{x}_i)^\top (h(\mathbf{y}_j) - h(\mathbf{y}_i))}{\tau} \right) \right)$$

$$+ \tau \cdot \frac{1}{|\mathcal{S}|} \sum_{i \in \mathcal{S}} \log \left( \varepsilon + \frac{1}{|\mathcal{S}| - 1} \sum_{j \in \mathcal{S}, j \neq i} \exp \left( \frac{h(\mathbf{y}_i)^\top (h(\mathbf{x}_j) - h(\mathbf{x}_i))}{\tau} \right) \right) + 2\tau\rho,$$

where $\tau$ is the temperature parameter, $\rho > 0$ is a hyperparameter, and $\varepsilon$ is a small constant. The equivalent min-min formulation then becomes

$$\min_{\mathbf{w}\in\mathbb{R}^d, \tau\in\mathbb{R}, \boldsymbol{\nu}_1\in\mathbb{R}^n, \boldsymbol{\nu}_2\in\mathbb{R}^n} \tau \cdot \frac{1}{|\mathcal{S}|} \sum_{i\in\mathcal{S}} \left\{ \left( \varepsilon + \frac{1}{|\mathcal{S}|-1} \sum_{j\in\mathcal{S}, j\neq i} \exp\left( \frac{h(\mathbf{x}_i)^\top (h(\mathbf{y}_j) - h(\mathbf{y}_i))}{\tau} \right) \right) \cdot e^{-\nu_{1,i}} + \nu_{1,i} \right\}$$

$$+ \tau \cdot \frac{1}{|\mathcal{S}|} \sum_{i\in\mathcal{S}} \left\{ \left( \varepsilon + \frac{1}{|\mathcal{S}|-1} \sum_{j\in\mathcal{S}, j\neq i} \exp\left( \frac{h(\mathbf{y}_i)^\top (h(\mathbf{x}_j) - h(\mathbf{x}_i))}{\tau} \right) \right) \cdot e^{-\nu_{2,i}} + \nu_{2,i} \right\} + 2\tau\rho.$$

In CLIP training, BSGD is named as OpenCLIP (Cherti et al., 2023) and SOX is named as FastCLIP (Wei et al., 2024). We use the DFN-14M dataset (Fang et al., 2024) for training. The trained models of different methods are evaluated on Datacomp (Gadre et al., 2023), a zero-shot evaluation benchmark, which consists of 35 zero-shot image-classification tasks and 3 zero-shot retrieval tasks. We present the average of top-1 accuracy on classification tasks and recall at 1 on retrieval tasks, and denote the metric as Datacomp Average. Moreover, we also present the average performance on two subsets of the benchmark: (1) ImageNet, which is the average top-1 accuracy on ImageNet-1K (Deng et al., 2009) and 6 distribution shift datasets (Wang et al., 2019; Recht et al., 2019; Hendrycks et al., 2021a;b; Barbu et al., 2019), and (2) Retrieval, which is the average of recall at 1 on MSCOCO (Chen et al., 2015) and Flickr30K (Young et al., 2014). We present the results in Figure 7, from which we can observe that SCENT has similar or slightly better performance, while ASGD-type methods perform poorly.

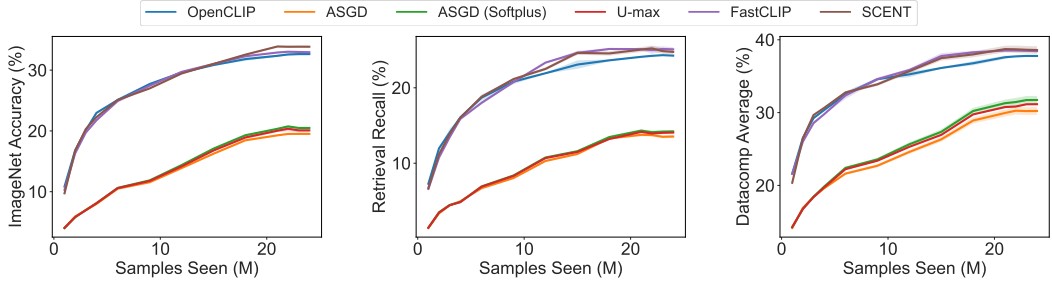

*Figure 7.* Zero-shot evaluation performance of different methods trained on DFN-14M. Left: ImageNet-1K top1 accuracy. Middle: Retrieval recall. Right: Datacomp average performance.

### F.3. KL-Regularized Distributionally Robust Optimization

We also consider KL-regularized distributionally robust optimization problem. Specifically, we consider linear regression task on a dataset $\mathcal{S} = \{(\mathbf{x_1}, y_1), \ldots, (\mathbf{x_n}, y_n)\}$:

$$\min_{\mathbf{a}\in\mathbb{R}^d, b\in\mathbb{R}} \max_{\mathbf{p}\in\Delta^n} \sum_{i\in\mathcal{S}} p_i (\mathbf{a}^\top \mathbf{x}_i + b - y_i)^2 - \tau D_{KL}(\mathbf{p}, \mathbf{1}/n), \tag{50}$$

where $\Delta^n$ is the unit simplex in $\mathbb{R}^n$ and $D_{KL}(\mathbf{p}, \mathbf{1}/n) \coloneqq \sum_{i=1}^n p_i \log(np_i)$ is the Kullback–Leibler divergence. For fixed parameters $\mathbf{a}$ and $b$, the optimal solution $\mathbf{p}$ of the maximization problem is given by $p_i^* = \frac{\exp((\mathbf{a}^\top \mathbf{x}_i + b - y_i)^2/\tau)}{\sum_j \exp((\mathbf{a}^\top \mathbf{x}_j + b - y_j)^2/\tau)}$. Then original problem (50) reduces to

$$\min_{\mathbf{a}\in\mathbb{R}^d, b\in\mathbb{R}} \tau \cdot \log\left( \frac{1}{|\mathcal{S}|} \sum_{i\in\mathcal{S}} \exp\left( \frac{(\mathbf{a}^\top \mathbf{x}_i + b - y_i)^2}{\tau} \right) \right). \tag{51}$$

The equivalent min-min formulation then becomes

$$\min_{\mathbf{a}\in\mathbb{R}^d, b\in\mathbb{R}, \nu\in\mathbb{R}} \tau \cdot \frac{1}{|\mathcal{S}|} \sum_{i\in\mathcal{S}} \left\{ \exp\left( \frac{(\mathbf{a}^\top \mathbf{x}_i + b - y_i)^2}{\tau} - \nu \right) + \nu - 1 \right\}.$$

We consider datasets California housing (Pace & Barry, 1997) and abalone (Nash et al., 1994). California housing consists of 20,640 objects represented by 8 features, while abalone dataset consists of 4,177 objects represented by 8 features. We compare all methods as previous experiments except ASGD, since it suffers from an overflow issue. Noticing SCGD is a special case of SOX when $n = 1$. We present the numerical result in Table 2, showing the objective value (51) (mean ±

---

**Algorithm 3** The SCENT Algorithm for Extreme Classification

---

**input** $\mathbf{w}_1 \in \mathbb{R}^{K \times d}, \boldsymbol{\nu}_0 \in \mathbb{R}^n$, step sizes $\eta_t, \alpha_t$, frozen backbone $h$, and a set of data with labels $\mathcal{S} = \{(\mathbf{x}_1, y_1), \ldots, (\mathbf{x}_n, y_n)\}$.

1: **for** $t = 1, \ldots, T-1$ **do**
2:     Sample $\mathcal{B}_t \subset \mathcal{S}$ with $|\mathcal{B}_t| = B$
3:     **for** each $(\mathbf{x}_i, y_i) \in \mathcal{B}_t$ **do**
4:         Update $\nu_{i,t}$:

$$\nu_{i,t} = \nu_{i,t-1} + \log\left(1 + \alpha_t \cdot \frac{1}{B-1} \sum_{j \in \mathcal{B}_t, j \neq i} \exp\left(h(\mathbf{x}_i)^\top (\mathbf{w}_{t,y_j} - \mathbf{w}_{t,y_i})\right)\right) - \log(1 + \alpha_t e^{\nu_{i,t-1}}).$$

5:     **end for**
6:     Compute the gradient estimator by $\mathbf{z}_t = \frac{1}{B} \sum_{i \in \mathcal{B}_t} \frac{1}{B-1} \sum_{j \in \mathcal{B}_t, j \neq i} \nabla_{\mathbf{w}} \exp\left(h(\mathbf{x}_i)^\top (\mathbf{w}_{t,y_j} - \mathbf{w}_{t,y_i}) - \nu_{i,t}\right)$.
7:     Update $\mathbf{w}_{t+1} = \mathbf{w}_t - \eta_t \mathbf{z}_t$
8: **end for**

---

standard deviation across 10 runs) after 300 epochs. The results show SCENT has better performance in most of cases.

*Table 2.* Objective value (51) across different $\tau$ value (mean ± std across 10 runs). Best results are shown in bold

| Methods | California housing | | | abalone | | |
|---|---|---|---|---|---|---|
| | $\tau = 0.2$ | $\tau = 1.0$ | $\tau = 5.0$ | $\tau = 0.2$ | $\tau = 1.0$ | $\tau = 5.0$ |
| BSGD | 7.943 (0.037) | 3.175 (0.014) | 0.743 (0.000) | 18.970 (0.033) | 11.313 (0.041) | 0.970 (0.000) |
| ASGD (Softplus) | 4.953 (0.006) | 2.030 (0.000) | 0.738 (0.002) | 16.094 (0.016) | 5.489 (0.002) | 0.965 (0.000) |
| U-max | 6.640 (0.173) | 2.066 (0.002) | 0.742 (0.000) | 10.951 (0.065) | 5.850 (0.027) | 0.966 (0.000) |
| SCGD | 5.182 (0.008) | 2.073 (0.002) | 0.738 (0.000) | **10.476** (0.043) | 5.625 (0.009) | **0.957** (0.000) |
| SCENT | **4.741** (0.071) | **2.001** (0.000) | **0.737** (0.001) | 13.664 (0.152) | **5.191** (0.001) | **0.957** (0.000) |

### F.4. Implementation Details and Hyperparameters

**Extreme classification**. For Glint360K, we use a ResNet-50 model released by the authors of the dataset to obtain the data used in this paper. Then we leverage the code released by the same authors to obtain the features. For TreeOfLife-10M, we use the CLIP ViT-B/16 model released by the authors of the dataset as well, and we use the code released by the same authors to obtain the features. We trained a linear model (a `torch.nn.Linear` model without bias) using both the SGD optimizer and the SGD with momentum optimizer. For the SGD optimizer, we train the model for 50 epochs. While for the SGD with momentum optimizer, we train the model for 20 epochs. For all methods, we tune the learning rate of the linear model from 1e-3 to 1e1. The learning rate follows a cosine schedule, where it starts from the tuned learning rate and gradually decreases to 0 in the end. For ASGD, ASGD (Softplus) and U-max, we tune the learning rate $\alpha$ of the dual variable from 1e-2 to 1e2, which also follows a cosine schedule. For ASGD (Softplus), we tune the approximation coefficient $\rho$ from 1e-5 to 1e-1, and we find that 1e-3 gives the best results across all settings. For U-max, we tune the threshold $\delta$ from 0.0 to 5.0, and we find that 1.0 gives the best results. For SOX, we tune the moving average coefficient $\gamma$ from 0 to 1, which also follows a cosine schedule. For SCENT, we tune the learning rate $\alpha$ of the dual variable by searching the value of $\log(\alpha)$ from 3 to 30. The algorithm we use is presented in Algorithm 3 and the hyperparameters are presented in Table 3.

**Partial AUC maximization**. For CIFAR-10 and CIFAR-100 (Krizhevsky, 2009), we construct imbalanced variants by randomly discarding a portion of positive samples following Zhu et al. (2022). Specifically, we group the first half of the classes as the negative class and the second half as the positive class, and then randomly remove 80% of the samples from the positive group to induce class imbalance. For both CIFAR-10 and CIFAR-100, we train convolutional neural networks using ResNet-18 (He et al., 2016) as the backbone. Our training pipeline consists of a pretraining stage followed by a classifier fine-tuning stage. In the pretraining stage, we optimize the full network using the cross-entropy (CE) loss with the SGD optimizer. We use a batch size of 64 and pretrain for 60 epochs with an initial learning rate of $10^{-3}$, which is decayed by a factor of 10 at epochs 20 and 40. After pretraining, we re-initialize the classifier layer, freeze the backbone, and fine-tune only the classifier using different methods. For all methods, we adopt the squared hinge loss as the surrogate

---

**Algorithm 4** The SCENT Algorithm for Partial AUC maximization

---

**input** $\mathbf{w}_1 \in \mathbb{R}^K, \boldsymbol{\nu}_0 \in \mathbb{R}^{|n_+|}$, step sizes $\eta_t, \alpha_t$, frozen backbone $h$, and a set of positive data $\mathcal{S}^+ = \{(\mathbf{x}_1, y_1), \ldots, (\mathbf{x}_{n_+}, y_{n_+})\}$ and a set of negative data $\mathcal{S}^- = \{(\mathbf{x}_1, y_1), \ldots, (\mathbf{x}_{n_-}, y_{n_-})\}$.

1: **for** $t = 1 \ldots, T - 1$ **do**
2:      Sample $\mathcal{S}_t^+ \subset \{1, \ldots, n_+\}$ with $|\mathcal{S}_t^+| = S^+$
3:      Sample $\mathcal{S}_t^- \subset \{1, \ldots, n_-\}$ with $|\mathcal{S}_t^-| = S^-$
4:      **for** each $i \in \mathcal{S}_t^+$ **do**
5:          Update $\nu_{i,t}$:

$$\nu_{i,t} = \nu_{i,t-1} + \log\left(1 + \alpha_t \cdot \frac{1}{S^-} \sum_{j \in \mathcal{S}_t^-} \exp\left(\frac{\ell(\mathbf{w}^\top(h(\mathbf{x}_j) - h(\mathbf{x}_i)))}{\tau}\right)\right) - \log(1 + \alpha_t e^{\nu_{i,t-1}}).$$

6:      **end for**
7:      Compute the gradient estimator by $\mathbf{z}_t = \frac{\tau}{S^+} \sum_{i \in \mathcal{S}_t^+} \frac{1}{S^-} \sum_{j \in \mathcal{S}_t^-} \nabla_{\mathbf{w}} \exp\left(\frac{\ell(\mathbf{w}^\top(h(\mathbf{x}_j) - h(\mathbf{x}_i)))}{\tau} - \nu_{i,t}\right)$.
8:      Update $\mathbf{w}_{t+1} = \mathbf{w}_t - \eta_t \mathbf{z}_t$
9: **end for**

---

*Table 3.* Hyperparameters of different methods on different datasets with different optimizers for extreme classification. Entries with "-" mean the corresponding hyperparameter is not used in the corresponding algorithm.

| Dataset | Optimizer | Hyper-parameter | BSGD | ASGD | ASGD (Softplus) | U-max | SOX | SCENT |
|---|---|---|---|---|---|---|---|---|
| Glint360K | SGD | lr | 1.0 | 0.5 | 0.5 | 0.5 | 5.0 | 5.0 |
| | | $\alpha$ | - | 1.0 | 1.0 | 1.0 | - | $e^{12}$ |
| | | $\gamma$ | - | - | - | - | 0.0 | - |
| | SGD w/ momentum | lr | 2e-3 | 1e-3 | 1e-3 | 1e-3 | 2e-3 | 1e-3 |
| | | $\alpha$ | - | 0.5 | 0.5 | 0.5 | - | $e^{30}$ |
| | | $\gamma$ | - | - | - | - | 0.2 | - |
| TreeOfLife-10M | SGD | lr | 2e-4 | 1e-3 | 1e-3 | 1e-3 | 5e-4 | 2e-2 |
| | | $\alpha$ | - | 2.0 | 2.0 | 2.0 | - | $e^3$ |
| | | $\gamma$ | - | - | - | - | 0.2 | - |
| | SGD w/ momentum | lr | 5e-4 | 2e-4 | 2e-4 | 2e-4 | 1e-3 | 2e-3 |
| | | $\alpha$ | - | 1.0 | 1.0 | 1.0 | - | $e^{10}$ |
| | | $\gamma$ | - | - | - | - | 0.6 | - |

loss $\ell(\cdot)$ with a fixed margin parameter of 0.5. We tune the learning rate for $\mathbf{w}$ from 1e-4 to 1e-2 for all methods and apply cosine learning-rate decay during training. For ASGD, the learning rate for updating $\nu$ is selected from 1e-3 to 1.0. For ASGD (Softplus), we additionally tune the approximation parameter $\rho$ from 1e-1 to 1e-5, which controls the approximation accuracy, and we use the same learning rate for the dual variable $\alpha$ as in Gladin et al. (2025). For U-max, we tune the learning rate of the dual variable from 1e-3 to 1e0 and select $\delta$ in 0 to 5. For SOX, we tune the moving-average parameter $\gamma$ from 0.3 to 0.9. For SCENT, we tune $\alpha_t$ for updating $\boldsymbol{\nu}$; in practice, we first train with SOX to inspect the convergence behavior of $\boldsymbol{\nu}$, and then choose $\alpha_t$ to be slightly smaller than the converged value of $\boldsymbol{\nu}$. We select $\tau = 0.05$ and 0.1 as the KL penalty coefficient, and when using momentum SGD, we fix the momentum parameter to 0.9. The algorithm we use is presented in Algorithm 4 and the hyperparameters are presented in Table 4.

**CLIP training**. We leverage the FastCLIP codebase for training, in which OpenCLIP and FastCLIP are already implemented. For all methods, we train a CLIP ViT-B/32 model (Dosovitskiy et al., 2021) using the AdamW optimizer (Loshchilov & Hutter, 2019). We train the model for 320M samples seen. For all methods, We tune the learning rate of the CLIP model from 1e-4 to 1e-3. The learning rate follows a cosine schedule. For ASGD, ASGD (Softplus) and U-max, we tune the learning rate $\alpha$ of the dual variable from 1e-2 to 1e2, which also follows a cosine schedule. For ASGD (Softplus), we tune the approximation coefficient $\rho$ from 1e-5 to 1e-1, and we find that 1e-3 gives the best evaluation performance. For U-max, we tune the threshold $\delta$ from 0.0 to 5.0, and we find that 1.0 gives the best results. For FastCLIP, we tune the moving average

*Table 4.* Hyperparameters of different methods on different datasets with different optimizers for partial AUC maximization with different $\tau$. Entries with "-" mean the corresponding hyperparameter is not used in the corresponding algorithm.

| Dataset | Optimizer | $\tau$ | Hyper-parameter | BSGD | ASGD | ASGD (Softplus) | U-max | SOX | SCENT |
|---|---|---|---|---|---|---|---|---|---|
| CIFAR-100 | SGD | 0.1 | lr | 1e-2 | 5e-3 | 1e-3 | 5e-3 | 5e-3 | 5e-3 |
| | | | $\alpha$ | - | 1.0 | 1e-3 | 1e-1 | - | $e^{-6}$ |
| | | | $\gamma$ | - | - | - | - | 0.9 | - |
| | | 0.05 | lr | 1e-3 | 1e-2 | 1e-3 | 1e-2 | 5e-3 | 5e-3 |
| | | | $\alpha$ | - | 1.0 | 1e-2 | 1e-1 | - | $e^{-15}$ |
| | | | $\gamma$ | - | - | - | - | 0.7 | - |
| | SGD w/ momentum | 0.1 | lr | 1e-3 | 1e-4 | 2e-4 | 1e-3 | 1e-3 | 1e-3 |
| | | | $\alpha$ | - | 1.0 | 1e-2 | 1.0 | - | $e^{-4}$ |
| | | | $\gamma$ | - | - | - | - | 0.7 | - |
| | | 0.05 | lr | 5e-4 | 1e-4 | 1e-4 | 1e-3 | 2e-4 | 1e-3 |
| | | | $\alpha$ | - | 1e-2 | 1e-2 | 1e-1 | - | $e^{-15}$ |
| | | | $\gamma$ | - | - | - | - | 0.7 | - |
| CIFAR-10 | SGD | 0.1 | lr | 1e-3 | 1e-2 | 1e-3 | 1e-2 | 5e-3 | 5e-3 |
| | | | $\alpha$ | - | 1.0 | 1e-2 | 1e-1 | - | $e^{-7}$ |
| | | | $\gamma$ | - | - | - | - | 0.7 | - |
| | | 0.05 | lr | 1e-2 | 1e-2 | 1e-3 | 1e-2 | 1e-2 | 1e-2 |
| | | | $\alpha$ | - | 1.0 | 1e-3 | 1e-1 | - | $e^{-16}$ |
| | | | $\gamma$ | - | - | - | - | 0.9 | - |
| | SGD w/ momentum | 0.1 | lr | 1e-4 | 1e-4 | 2e-4 | 2e-4 | 2e-4 | 5e-4 |
| | | | $\alpha$ | - | 1.0 | 1e-2 | 1.0 | - | $e^{-7}$ |
| | | | $\gamma$ | - | - | - | - | 0.3 | - |
| | | 0.05 | lr | 1e-3 | 1e-3 | 2e-4 | 1e-3 | 1e-3 | 1e-3 |
| | | | $\alpha$ | - | 1e-1 | 1e-2 | 1e-1 | - | $e^{-16}$ |
| | | | $\gamma$ | - | - | - | - | 0.9 | - |

---

**Algorithm 5** The SCENT Algorithm for CLIP Training

---

**input** CLIP model $h$ initialized with $\mathbf{w}_1 \in \mathbb{R}^d$, temperature parameter $\tau$, $\boldsymbol{\nu}_{1,0}, \boldsymbol{\nu}_{2,0} \in \mathbb{R}^n$, step sizes $\eta_t, \alpha_t$, and a set of image-text pairs $\mathcal{S} = \{(\mathbf{x}_1, \mathbf{y}_1), \ldots, (\mathbf{x}_n, \mathbf{y}_n)\}$.

1: **for** $t = 1, \ldots, T - 1$ **do**
2:     Sample $\mathcal{B}_t \subset \mathcal{S}$ with $|\mathcal{B}_t| = B$
3:     Obtain features of data in the batch: $\hat{\mathcal{B}}_t = \{(h(\mathbf{x}_i), h(\mathbf{y}_i)) : (\mathbf{x}_i, \mathbf{y}_i) \in \mathcal{B}_t\}$
4:     **for** each $(\mathbf{e}_{1,i}, \mathbf{e}_{2,i}) \in \hat{\mathcal{B}}_t$ **do**
5:         Update $\nu_{1,i,t}, \nu_{2,i,t}$:

$$\nu_{1,i,t} = \nu_{1,i,t-1} + \log\left(1 + \alpha_t \cdot \frac{1}{B-1} \sum_{j \in \mathcal{B}_t, j \neq i} \exp\left(\frac{\mathbf{e}_{1,i}^\top(\mathbf{e}_{2,j} - \mathbf{e}_{2,i})}{\tau}\right)\right) - \log(1 + \alpha_t e^{\nu_{1,i,t-1}}),$$

$$\nu_{2,i,t} = \nu_{2,i,t-1} + \log\left(1 + \alpha_t \cdot \frac{1}{B-1} \sum_{j \in \mathcal{B}_t, j \neq i} \exp\left(\frac{\mathbf{e}_{2,i}^\top(\mathbf{e}_{1,j} - \mathbf{e}_{1,i})}{\tau}\right)\right) - \log(1 + \alpha_t e^{\nu_{2,i,t-1}}).$$

6:     **end for**
7:     Compute the gradient estimator by

$$\mathbf{z}_t = \frac{1}{B} \sum_{i \in \mathcal{B}_t} \frac{1}{B-1} \sum_{j \in \mathcal{B}_t, j \neq i} \left(\nabla_\mathbf{w} \exp\left(\frac{\mathbf{e}_{1,i}^\top(\mathbf{e}_{2,j} - \mathbf{e}_{2,i})}{\tau} - \nu_{1,i,t}\right) + \nabla_\mathbf{w} \exp\left(\frac{\mathbf{e}_{2,i}^\top(\mathbf{e}_{1,j} - \mathbf{e}_{1,i})}{\tau} - \nu_{2,i,t}\right)\right)$$

8:     Update $\mathbf{w}_{t+1}$ using the AdamW optimizer with $\eta_t$ and $\mathbf{z}_t$
9: **end for**

---

coefficient $\gamma$ from 0 to 1, which also follows a cosine schedule. For SCENT, we tune the learning rate $\alpha$ of the dual variable by searching the value of $\log(\alpha)$ from 3 to 30. The algorithm we use is presented in Algorithm 5 and the hyperparameters are presented in Table 5.

**KL-regularized distributionally robust optimization** We consider linear regression tasks on the California Housing dataset (Pace & Barry, 1997) and the Abalone dataset (Nash et al., 1994). For Abalone, we normalize the target values to keep the loss on a numerically convenient scale, while leaving the feature space unchanged. We evaluate penalty coefficients $\tau$ in [0.2, 1, 5]. Across all methods, we use a batch size of 100 and train for 300 epochs using SGD with momentum 0.9. Following Gladin et al. (2025), we initialize optimization at the least-squares solution. For all methods, we tune the learning rate of $\mathbf{w}$ from 1e-7 to 1e-4 and apply cosine decay throughout training. For ASGD (Softplus), we tune the approximation parameter $\rho$ from 1e-5 to 1e-1, and set the learning rate for the dual variable $\alpha$ following Gladin et al. (2025). For U-max, we tune the dual learning rate from 1e-3 to 1e0 and $\delta$ from 0.1 to 5. For SCGD, we tune the moving-average parameter $\gamma$ from 0 to 1. For SCENT, we tune the step size $\alpha_t$ used to update $\nu$: specifically, we first run SCGD to inspect the convergence

---

**Algorithm 6** The SCENT Algorithm for KL DRO

---

**input** $\mathbf{a} \in \mathbb{R}^d, b \in \mathbb{R}, \nu_0 \in \mathbb{R}$, step sizes $\eta_t, \alpha_t$, and a set of data with labels $\mathcal{S} = \{(\mathbf{x}_1, y_1), \ldots, (\mathbf{x}_n, y_n)\}$.

1: **for** $t = 1 \ldots, T - 1$ **do**
2:     Sample $\mathcal{B}_t \subset \mathcal{S}$ with $|\mathcal{B}_t| = B$
3:     Update $\nu_t$:

$$\nu_t = \nu_{t-1} + \log\left(1 + \alpha_t \cdot \frac{1}{B} \sum_{i \in \mathcal{B}_t} \exp\left(\frac{(\mathbf{a}^\top\mathbf{x}_i + b - y_i)^2}{\tau}\right)\right) - \log(1 + \alpha_t e^{\nu_{t-1}}).$$

4:     Compute the gradient estimator for $\mathbf{a}$ by $\mathbf{z}_{t,1} = \frac{\tau}{B} \sum_{i \in \mathcal{B}_t} \nabla_\mathbf{a} \exp\left(\frac{(\mathbf{a}^\top\mathbf{x}_i + b - y_i)^2}{\tau} - \nu_t\right)$.
5:     Compute the gradient estimator for $b$ by $\mathbf{z}_{t,2} = \frac{\tau}{B} \sum_{i \in \mathcal{B}_t} \nabla_b \exp\left(\frac{(\mathbf{a}^\top\mathbf{x}_i + b - y_i)^2}{\tau} - \nu_t\right)$.
6:     Update $\mathbf{a}_{t+1} = \mathbf{a}_t - \eta_t \mathbf{z}_{t,1}, b_{t+1} = b_t - \eta_t \mathbf{z}_{t,2}$
7: **end for**

---

*Table 5.* Hyperparameters of different methods for CLIP training on DFN-14M

| Hyperparameter | BSGD | ASGD | ASGD (Softplus) | U-max | SOX | SCENT |
|---|---|---|---|---|---|---|
| lr | 5e-4 | 5e-4 | 5e-4 | 5e-4 | 5e-4 | 5e-4 |
| $\alpha$ | - | 0.1 | 0.1 | 0.1 | - | $e^{10}$ |
| $\gamma$ | - | - | - | - | 0.4 | - |

trajectory of $\nu$, and then choose $\alpha_t$ such that $\nu$ converges to a value slightly smaller than the SCGD limit. The algorithm we use is presented in Algorithm 6 and the hyperparameters are presented in Table 6.

*Table 6.* Hyperparameters of different methods on different datasets for KL-regularized distributionally robust optimization with different $\tau$. Entries with "-" mean the corresponding hyperparameter is not used in the corresponding algorithm.

| Dataset | $\tau$ | Hyperparameter | BSGD | ASGD (Softplus) | U-max | SCGD | SCENT |
|---|---|---|---|---|---|---|---|
| | | lr | 1e-5 | 1e-6 | 1e-5 | 5e-6 | 1e-5 |
| | 0.2 | $\alpha$ | - | 1e-6 | 1e-0 | - | $e^{-22}$ |
| | | $\gamma$ | - | - | - | 0.5 | - |
| | | lr | 5e-6 | 1e-6 | 5e-6 | 5e-6 | 5e-6 |
| California housing | 1.0 | $\alpha$ | - | 1e-6 | 1e-0 | - | $e^{-4}$ |
| | | $\gamma$ | - | - | - | 0.4 | - |
| | | lr | 5e-6 | 1e-5 | 1e-4 | 1e-5 | 1e-5 |
| | 5.0 | $\alpha$ | - | 1e-5 | 1e-0 | - | $e^{-1.1}$ |
| | | $\gamma$ | - | - | - | 0.8 | - |
| | | lr | 1e-5 | 5e-5 | 5e-5 | 5e-5 | 1e-4 |
| | 0.2 | $\alpha$ | - | 5e-5 | 1e-0 | - | $e^{-38}$ |
| | | $\gamma$ | - | - | - | 0.3 | - |
| | | lr | 1e-5 | 5e-5 | 1e-4 | 1e-5 | 5e-5 |
| abalone | 1.0 | $\alpha$ | - | 5e-5 | 1e-0 | - | $e^{-10}$ |
| | | $\gamma$ | - | - | - | 0.1 | - |
| | | lr | 1e-4 | 1e-4 | 1e-4 | 1e-4 | 1e-4 |
| | 5.0 | $\alpha$ | - | 1e-4 | 1e-1 | - | $e^{-4}$ |
| | | $\gamma$ | - | - | - | 0.9 | - |

**Comparison between SGD and SPMD on Gaussian noise**. For each combination of mean and variance, we sample 1 million points from the Gaussian distribution using `torch.normal`. Then we run SGD and SPMD on the training data, and record $\nu_t$ at each iteration. Finally, we plot the squared error between $\nu_t$ and $\nu_*$. We tune the learning rate $\alpha$ of the SGD update from 1e-2 to 1e2, and select 1.0 for all cases. We tune the learning rate $\alpha$ of the SPMD update from -8.0 to 5.0, and select -6.0 for all cases when the mean of the Gaussian distribution is -1.0, and select 3.0 for all cases when the mean of the Gaussian distribution is -10.0.

