# OpenReview forum: "A Geometry-Aware Efficient Algorithm for Compositional Entropic Risk Minimization"
_ICML.cc/2026/Conference — ICML 2026 regular_

### Official Review · Reviewer_HE8X · 2026-03-09

**Soundness:** 2
**Presentation:** 3
**Significance:** 2
**Originality:** 2
**Overall Recommendation:** 3
**Confidence:** 3

**Summary:**

The authors proposed to reformulate the original log-sum-exp loss function into a min-min form with an introduction of auxiliary variable, and apply stochastic mirror descent method to solve the related optimization problem. Both convergency analysis and empirical studies with ResNet model were presented to support the effectiveness of the proposed algorithm.

**Compliance With Llm Reviewing Policy:**

Affirmed.

**Key Questions For Authors:**

The authors should address the questions listed above.

**Limitations:**

no, but there is no negative social impact of this work.

**Strengths And Weaknesses:**

Strength:
1. The idea of applying stochastic mirror descent algorithm to the min-min reformulation of log-sum-exp loss is novel.
2. Both convergency analysis and empirical studies were presented to support the effectiveness of the proposed algorithm.

Weakness:
1. Since the number of auxiliary variables $\nu$ is $n$, the resulting convergency bound will depend on $n$, making it less useful in theory.
2. More analysis and experimental studies are needed to understand why using exponential function as basis for the Bregman distance is more effective. It appears that choice of Bregman distance function should not have impact on the overall convergency.
3. The author may want to compare the proposed algorithms to other compositional optimization algorithms (e.g. "Finite-Sum Coupled Compositional Stochastic Optimization: Theory and Applications") and extreme class classification algorithm (e.g. "Large-Scale Training System for 100-Million Classification at Alibaba").
4. For partial AUC optimization, the author should use the imagenet dataset, a significantly larger dataset than CIFAR 10 and 100, for experiments.

---

> ### Author Rebuttal · Authors · 2026-03-31
>
> We thank the reviewer for their valuable comments and suggestions.
>
> > **Q1**: Since the number of auxiliary variables $\nu$ is $n$, the resulting convergence bound will depend on $n$, making it less useful in theory.
>
> **A**: We respectfully disagree with the reviewer’s claim that the theory is less useful due to its dependence on $n$. First let us discuss the convergence of existing methods that do not leverage the dual variable. Consider biased SGD (BSGD) for conditional stochastic optimization (CSO) of the form $\mathbb E_{\xi}[f_{\xi}(\mathbb E_{\zeta} g_{\zeta}(\mathbf{w}; \xi))]$, which includes our problem as a special case. Hu et al. (2020) show that, for convex CSO, BSGD requires a batch size of $O(1/\epsilon)$ and an iteration complexity of $O(1/\epsilon^2)$, leading to an overall sample complexity of $O(1/\epsilon^3)$.
>
> In contrast, our method achieves a worst-case sample complexity of $O(\frac{n}{\epsilon^2})$, which is better than BSGD when $n < O(1/\epsilon)$. More broadly, the dependence on $n$ is standard in finite-sum stochastic optimization: methods such as SVRG, SPIDER, and stochastic dual coordinate descent for SVM all involve $n$ while improving the dependence on $\epsilon$ compared to standard SGD.
>
> > **Q2**: The choice of Bregman distance function should not have impact on the overall convergence.
>
> **A**: This is not correct, the choice of Bregman divergence **does have** impact on the overall convergence.
> - First, if we use the Euclidean distance, there is no closed-form update for $\nu_t$, making approximation necessary, which could slow down convergence.
> - Second, let us compare with the stochastic proximal method with Euclidean distance for solving Eq (12), i.e., $\min_{\nu}\mathbb E_{\zeta}[e^{s(\zeta)-\nu} + \nu]$. This is the problem on the dual variable with $\mathbf w$ fixed.  For the stochastic proximal method with Euclidean distance, its convergence rate was established in  Chadha et al. (2021) (see Eq (12) in their paper), which is similar to that of SGD. Due to the Euclidean update, its convergence depends on the variance bound of $\text{Var}(e^{s(\zeta)}e^{-\nu_{t-1}})\leq \text{Var}(z)e^{-2c_0}$, where $z= e^{s(\zeta)}$. Since $\text{Var}(z)=(\mathbb E z)^2(\kappa -1) = e^{2\nu_*}(\kappa -1)$, where $\nu_*=\log \mathbb E[z]$ is the optimal solution, the variance bound $\text{Var}(e^{s(\zeta)}e^{-\nu_{t-1}})\leq \text{Var}(z)e^{-c_0}$  will necessarily depend on the exponentially large constant $e^{2(\nu_*-c_0)}$.
> - The key benefit of using the proposed SPMD update is that the variance proxy of the dual update is given by $\delta_t = e^{-\nu_{t-1}}\text{Var}(z)$, which scales as $e^{-\nu_{t-1}}$ instead of $e^{-2\nu_{t-1}}$. As a result, it enjoys a self-bounding inequality (see Lemma 4.2). It is this key inequality that allows us to build a better bound of SPMD in Theorem 4.3 (see Appendix D.2).
>
> Chadha et al. "Accelerated, Optimal, and Parallel: Some Results on Model-Based Stochastic Optimization". 2021
>
> > **Q3**: The author should compare the proposed algorithms to other compositional optimization algorithms e.g. "Finite-Sum ...") and extreme class classification algorithm (e.g. "Large-Scale ...").
>
> **A**: Thanks for the suggestion. In fact, the reviewer **missed** that we have already compared with the first work, which is the SOX method in the paper. The second mentioned work is interesting. We will cite and discuss it in the revision. However, their work focused on improving algorithms for distributed setting with multiple GPUs (such as efficient communication strategy). In our paper, all experiments are conducted in a single GPU setting. In the future, we expect to extend the proposed algorithm in a distributed setting and compare with the second work.
>
> > **Q4**: Partial AUC optimization on larger datasets such as ImageNet.
>
> **A**: Thanks for suggestion. We would like to point out that even on CIFAR10 and CIFAR100 datasets, solving the partial AUC maximization problem is challenging as it involves a large number of 25K negative samples inside the log function, while the number of positive data is 6.25K.
>
> We have conducted additional experiments on Imagenet-100 and Chexpert dataset. Imagenet-100 includes 100 classes in total and each class has 1.4K images. We constructed an imbalanced version similar to the CIFAR100 experiment, resulting in a data with 16.25K positive samples and 65K negative samples. Chexpert is an imbalanced medical image dataset which includes 48K positive samples and 179K negative samples. The final training loss (averaged over three runs) and the standard deviation with $\lambda=0.05$ on these two datasets for different methods are reported below, and we can see that SCENT still outperforms the baselines.
> ||SCENT|SOX|ASGD|BSGD|
> |--|--|--|--|--|
> |Imagenet-100|**0.00620 (0.00)**|0.00626 (0.00)|0.00678 (0.00)|0.00627 (0.00)|
> |Chexpert|**0.9861 (0.00)**|0.9862 (0.00)|0.9895 (0.00)|0.9868 (0.00)|

---

### Official Review · Reviewer_z3qH · 2026-03-11

**Soundness:** 3
**Presentation:** 3
**Significance:** 2
**Originality:** 2
**Overall Recommendation:** 3
**Confidence:** 4

**Summary:**

This paper addresses the optimization challenges associated with Compositional Entropic Risk Minimization (CERM), a framework where the loss is formulated as a Log-Expectation-Exponential (Log-E-Exp) function. The authors propose SCENT (Stochastic optimization of Compositional ENTropic risk), a geometry-aware stochastic algorithm that tackles the min-min dual formulation of the entropic risk. To mitigate the exponentially large smoothness parameter associated with the dual variable, the authors introduce a stochastic proximal mirror descent (SPMD) update equipped with a Bregman divergence induced by a negative exponential function (φ(ν)=e^(-ν)). The paper provides an O(1/√T) convergence rate for convex problems and evaluates the algorithm empirically across various tasks, including extreme classification, partial AUC maximization, contrastive learning, and distributionally robust optimization.

**Compliance With Llm Reviewing Policy:**

Affirmed.

**Key Questions For Authors:**

The min–min formulation introduces a per-data dual variable vector ν ∈ ℝⁿ  The paper does not convincingly argue or quantify the memory and update cost for very large n (e.g., extreme classification / TreeOfLife-10M / million-example regimes). How are ν stored/updated in practice (persisted on disk/parameter server / per-shard)? If ν is maintained explicitly for all n, runtime/memory will be a bottleneck. It should provide a clear complexity (time + memory) per iteration in terms of n, the batch size B, and the number of sampled classes m, and  describe practical strategies (sparse updates, lazy updates, hashing, memory sharding) or an online variant that avoids storing full ν.

**Limitations:**

There is a gap between the theoretical claims and practice.

**Strengths And Weaknesses:**

Strengths：
1. The specific choice of φ(ν) for the Bregman divergence is effective, as it respects the geometry of the objective and yields a stable, closed-form update for the dual variable. This cleverly sidesteps the numerical overflow and instability issues that plague vanilla gradient methods dealing with exponential functions.
2. The paper establishes a connection with existing methods.
3. Experimental results validate the satisfactory performance.

Weaknesses:
1. There is a gap between the theory and practice, say, the $O(/\sqrt{T})$ convergence rate—is strictly predicated on Assumption 3.2, which requires the loss function to be convex with respect to the model parameters. But in empirical evaluations for partial AUC maximization and CLIP training, the authors deploy highly non-convex deep learning architectures, such as ResNet-18, ResNet-50, and CLIP ViT-B/16. This may not sufficiently support the theoretical claims.

2.  The overarching strategy of casting Log-E-Exp into a min-min formulation is a well-known technique dating back to 1986. Furthermore, utilizing compositional optimization for this problem has been explored extensively by prior works. The true novelty of this paper is confined to the specific choice of the Bregman divergence used in the optimizer for the dual variable. While this is a neat mathematical trick, it represents a precise algorithmic refinement rather than a paradigm-shifting innovation.

3. For CERM problems with n components, the dual formulation requires maintaining a variable vector ν∈R^n. For extreme classification on the TreeOfLife-10M dataset, this implies holding and updating millions of state variables. Although the authors utilize a stochastic block coordinate update to handle large n, the paper fails to systematically profile the actual memory consumption or discuss the I/O bottlenecks of maintaining this massive dual state in real-world distributed deep learning frameworks.

4. The theoretical bounds for the SPMD update strictly require the dual step size $\alpha_t$ to satisfy $\alpha_t \le \rho e^{-v_{i, t-1}}$ to control variance. Yet, the implementation details reveal that the authors relied on massive, arbitrary grid searches for this parameter (e.g., searching log⁡α from 3 to 30 for extreme classification and CLIP). This empirical tuning strategy seems to  contradict the careful  step-size bounds dictated by the theory, which implies that the algorithm may be fragile and expensive to tune across new datasets.

---

> ### Author Rebuttal · Authors · 2026-03-31
>
> We thank the reviewer for the suggestion on experiments.
>
> > **Q1**: In partial AUC maximization and CLIP training, the authors deploy highly non-convex deep learning architectures. This may not sufficiently support the theoretical claims, which assume the model to be convex.
>
> **A**: There is a **misunderstanding**.
> - In partial AUC maximization, we fix a pretrained encoder and only learn the linear head. Hence it is consistent with the theoretical setting.  It is true that the CLIP training is generally non-convex as we update the parameters of the whole network.
> - Thus all the experiments in our paper except CLIP training consider the convex setting, which fully support the theoretical claims.
>
> > **Q2**: The true novelty of this paper is confined to the specific choice of the Bregman divergence used in the optimizer for the dual variable. While this is a neat mathematical trick, it represents a precise algorithmic refinement rather than a paradigm-shifting innovation.
>
> **A**: We humbly disagree with the reviewer. Let us clarify the significance of this work.
> - While the min-min formulation is well-known, the existing algorithms based on it are not practical and are even worse than the naive mini-batch based SGD (BSGD). This can be seen from the experimental results on ASGD, ASGD (softplus) and U-max, which are based on the min-min formulation. Our work makes the stochastic algorithm based on the min-min formulation work in practice better than BSGD.
> - While compositional optimization algorithms have been developed, the existing convergence rate is worse than our results. This has been clarified in the related work section. For example, the convergence rate of SCGD in Wang et al. (2017) is $O(1/T^{1/4})$ for convex problems. The convergence analysis of a double-loop variant of SOX in Wang & Yang (2022) yields a rate of $O(1/T^{1/3})$. Other analyses of convex compositional optimization that enjoy the rate $O(1/\sqrt{T})$ require much stronger assumptions such as convexity of the outer function $f$ (Zhang & Lan 2020) or the mean smoothness conditions of inner functions and requires a double-loop design (Jiang et al. 2022). This is the **first time** that a practical algorithm is proposed with the best convergence rate.
> - The proposed algorithmic framework provides a unified venue to understand the limitations of existing methods including BSGD, ASGD and SOX (please see Appendix A for more details).
>
> > **Q3**: The paper should provide a clear complexity analysis of maintaining the dual variable.
>
> **A**: Thank you for the suggestion.
> - We store the dual variable in CPU memory and at each iteration only move those entries in the mini-batch $\\{\nu_{i}: i\in \mathcal{B}\\}$ to GPU to update them. Thus the cost for CPU memory is $O(n)$, and that for GPU memory is $O(B)$. For extreme classification on Glint360K (17M images and 128 batch size), the memory cost for CPU is only 64MB (using standard fp32 precision) and that for GPU is 0.5KB, which is negligible for machine learning systems. For other datasets, the cost for GPU memory is similar and that for CPU will be smaller due to fewer samples.
> - The time cost comes from two parts: (1) updating $\\{\nu_{i}: i\in \mathcal{B}\\}$, which are a few simple operations (log, sum, exp, etc.) that can be finished in minimal time; and (2) moving $\\{\nu_{i}: i\in \mathcal{B}\\}$ to and from GPU, which is moving a $|\mathcal{B}|\times 1$ tensor, and is also negligible compared to moving the training data (a $|\mathcal{B}|\times 512$ tensor in extreme classification) to GPU.
> - We use PyTorch profiler to record the running time and GPU memory of SCENT and BSGD (the only baseline that does not maintain the dual variable) on Glint360K on a NVIDIA A100 GPU. The results below show that the two methods have similar cost.
>
> ||Memory (GB)|Time (ms)|
> |--|--|--|
> |BSGD|2.18|6.46|
> |SCENT|2.18|6.68|
> |Additional Cost|0%|3%|
>
> We will add these discussions into the revision.
>
> > **Q4**: The implementation details reveal that the authors relied on massive, arbitrary grid searches for $\alpha$.
>
> **A**: We agree with the reviewer that the parameter tuning of $\alpha$ is required. Nevertheless, we provide some guidance on tuning it without massive and arbitrary grid searches. We use the following strategy to find a good starting point $\alpha_{0}$, which greatly reduces the search range. Since Theorem 3.6 indicates that $\alpha_t$ depends on $\nu_{t-1}$ (the dual variable), we first run SOX (with default hyperparameters) for a few epochs and compute the median of $\nu$ across all samples at the end of each epoch. Then we find $\alpha\_{0}:= e^{\alpha'\_{0}}$ such that the curve of $\frac{\alpha\_{0} e^{\nu}}{1+ \alpha\_{0} e^{\nu}}$ vs. the epoch is similar to the cosine decay schedule of $\gamma_t$ used in SOX. Finally we conduct search in the range $e^{\alpha'\_{0} - 3}$, $e^{\alpha'\_{0}- 2}$, $e^{\alpha'\_{0}- 1}$, ..., $e^{\alpha'\_{0}+ 3}$ for the best $\alpha$ for SCENT.

---

> > ### Author Rebuttal · Reviewer_z3qH · 2026-04-03
> >
> > Thank you for the authors' detailed responses.  Thank you for the authors' detailed responses. However, the explanation of "fix a pretrained encoder and only learn the linear head" may not fully bridge the gap between the assumption and practical implementation. I would like to keep my original score.

---

> > > ### Author Response · Authors · 2026-04-03
> > >
> > > Thanks for acknowledging our rebuttal.
> > >
> > > We appreciate the opportunity to further clarify this issue. In fact, the convexity of partial AUC maximization can be established when the encoder $h(\mathbf{x})$ is fixed and only the linear head $\mathbf{w}$ is learned. Recall the formulation of partial AUC maximization:
> > > \begin{align}
> > > \min_{\mathbf{w}\in\mathcal{W}} \; F_{\text{pAUC}}(\mathbf{w})
> > > := \frac{1}{n_+}\sum_{i=1}^{n_+} \tau \log\left[\frac{1}{n_-}\sum_{j=1}^{n_-}\exp\left(\frac{\ell\big(\mathbf{w}^{\top}(h(\mathbf{x}_j^-) - h(\mathbf{x}_i^+))\big)}{\tau}\right)\right].
> > > \end{align}
> > >
> > > We adopt the squared hinge loss $\ell(t)=\max(t+0.5, 0)^2$ (see lines 1805–1806), which is convex and monotonically non-decreasing. In the paper, we reformulate this objective into the following min–min problem (up to an additive constant):
> > > $$
> > > \min_{\mathbf{w}, \boldsymbol{\nu}} \; F(\mathbf{w}, \boldsymbol{\nu})
> > > := \frac{1}{n_+}\sum_{i=1}^{n_+}\left[\frac{1}{n_-}\sum_{j=1}^{n_-}\tau \exp\big(s_i(\mathbf{w}, \mathbf{x}_j^-)-\nu_i\big) + \nu_i \right],
> > > $$
> > > where
> > > $$
> > > s_i(\mathbf{w}, \mathbf{x}_j^-)
> > > = \frac{\ell\big(\mathbf{w}^{\top}(h(\mathbf{x}_j^-) - h(\mathbf{x}_i^+))\big)}{\tau}.
> > > $$
> > >
> > > By treating $\mathbf x_j^-, j\in\\{1,\ldots,n_-\\}$, as a random variable $\zeta$, the above formulation can be equivalently written as
> > > $$
> > > \min_{\mathbf{w}, \boldsymbol{\nu}} \; F(\mathbf{w}, \boldsymbol{\nu})
> > > := \frac{1}{n_+}\sum_{i=1}^{n_+}\left[\mathbb{E}_{\zeta} \tau \exp\big(s_i(\mathbf{w}, \zeta)-\nu_i\big) + \nu_i \right],
> > > $$
> > > where
> > > $$
> > > s_i(\mathbf{w}, \zeta)
> > > = \frac{\ell\big(\mathbf{w}^{\top}(h(\zeta) - h(\mathbf{x}_i^+))\big)}{\tau}.
> > > $$
> > >
> > > Since $\ell(t)$ is convex, $s_i(\mathbf{w}, \zeta)$ is also convex in $\mathbf{w}$ as it is the composition of a convex function with an affine mapping. Building on this, **Lemma B.1** in the paper establishes the joint convexity of $F(\mathbf{w}, \boldsymbol{\nu})$ with respect to $(\mathbf{w}, \boldsymbol{\nu})$.
> > >
> > > Finally, by a standard result in convex optimization (see Section 3.2.5 of *Convex Optimization* by Boyd and Vandenberghe), the convexity of $F_{\text{pAUC}}(\mathbf{w})$ follows directly.
> > >
> > > Indeed, the convexity of  $F_{\text{pAUC}}(\mathbf{w})$ has also been proved in *Convex Optimization* (Boyd and Vandenberghe, Example 3.14). Since the objective is a composition of the log-sum-exp function with convex functions, the vector composition rules guarantees that the objective is convex.
> > >
> > > This should address your last concern.

---

### Official Review · Reviewer_rgcF · 2026-03-12

**Soundness:** 3
**Presentation:** 3
**Significance:** 3
**Originality:** 3
**Overall Recommendation:** 5
**Confidence:** 3

**Summary:**

The authors propose a novel optimization algorithm for solving compositional entropic risk minimization (CERM). CERM can be cast as a min-min problem from the dual perspective. The authors employ a stochastic proximal mirror descent for updating the dual variable, where a Bregman distance induced by a negative exponential function is used to mitigate the effect of noise caused by the stochastic estimator of the objective. They establish convergence analysis for the proposed method in the convex setting and show that it achieves convergence rate of $O(1/\sqrt{T})$. The proposed method is validated on various applications to demonstrate its effectiveness and robustness.

**Compliance With Llm Reviewing Policy:**

Affirmed.

**Final Justification:**

The authors have addressed my concerns, and I maintain my positive recommendation.

**Key Questions For Authors:**

1. Could the authors clarify what criteria or validation metrics are used for hyperparameter selection?

2. Could the authors clarify the step sizes used for each hyperparameter within the search ranges?

3. Since proximal stochastic gradient methods also do not require full access to the objective, could the authors discuss whether these methods can be applied to optimize the dual variable, and if not, what the main obstacles are?

**Limitations:**

It would be helpful if the authors could discuss the limitations of the proposed method in the non-convex setting, as well as possible extensions.

**Strengths And Weaknesses:**

**Strengths:**

The paper is well-written and theoretically solid. Connections with related methods for solving CREM have been discussed and the unified formulation of these methods under the proposed approach might help better understand their limitations. The analysis of the convergence bound supports the theoretical guarantee. Experimental evaluations across different tasks have been conducted to show the proposed method’s advantages over existing optimizers.


**Weaknesses:**

Some details of the evaluations are missing.

---

> ### Author Rebuttal · Authors · 2026-03-31
>
> We thank the reviewer for their constructive and positive evaluation of our work.
>
> > **Q1**: Could the authors clarify what criteria or validation metrics are used for hyperparameter selection?
>
> **A**: We select the hyperparameters based on the lowest training loss.
>
> > **Q2**: Could the authors clarify the step sizes used for each hyperparameter within the search ranges?
>
> **A**:
> - Regarding the learning rate $\eta$ for updating $\mathbf w$ of all methods, we search the best hyperparameter following a log pattern. For extreme classification (where we set the search range to be 1e-3 to 1e1), we run each method with learning rate 1e-3, 2e-3, 5e-3, 1e-2, 2e-2, 5e-2, 1e-1, ..., 5e0, 1e1. For other tasks we use the same strategy except that the bounds are different. Specifically, for partial AUC maximization, the learning rate is searched in 1e-5, 2e-5, 5e-5, 1e-4, 2e-4, 5e-4, ..., 1e-3. For CLIP training, the learning rate is searched in 1e-4, 2e-4, 5e-4 and 1e-3. And for DRO, the learning rate is searched in 1e-7, 2e-7, 5e-7, 1e-6, 2e-6, 5e-6, ..., 1e-4.
> - For $\gamma$ of SOX, we search the best hyperparameter at a step size of 0.1 between 0 and 0.9, along with 0.95, 0.99 and 1.0.
> - For $\alpha$, the hyperparameter is searched following the log pattern as well. For ASGD type methods, the learning rate $\alpha$ for updating $\nu$ is searched in the same way as that for $\eta$. For SCENT, we first choose an initial $\alpha\_{0}:= e^{\alpha'\_{0}}$ (please refer to response to Q4 of reviewer z3qH) and search the best $\alpha$ in $e^{\alpha'\_{0}- 3}$, $e^{\alpha'\_{0}- 2}$, $e^{\alpha'\_{0}- 1}$, ..., $e^{\alpha'\_{0}+ 3}$.
>
> > **Q3**: Could the authors discuss whether proximal stochastic gradient methods can be applied to optimize the dual variable?
>
> **A**: We understand the reviewer's question as using the update $\nu_t=\arg\min_{\nu} \Phi(\mathbf w_t, \nu, \zeta_t) + \frac{1}{2\alpha_t}|\nu - \nu_{t-1}|^2$; if this is not the case please clarify. The caveat of this update is that it does not have a closed-form solution and can only be approximately solved, incurring additional overhead. A similar update was studied in  Fagan & Iyengar (2018) as discussed in lines 110-128 (left) and that is why they introduced a second algorithm U-max.
>
> > **Q4**: It would be helpful if the authors could discuss the limitations of the proposed method in the non-convex setting, as well as possible extensions.
>
> **A**: The proposed algorithm can be applied to non-convex setting as in the experiment for CLIP training where we learn deep neural networks. The performance of SCENT is competitive with FastCLIP and better than all other baselines. One limitation is how to conduct the convergence analysis for non-convex setting to exhibit the potential advantages over existing methods such as SOX (Wang & Yang 2022).

---

> > ### Author Rebuttal · Reviewer_rgcF · 2026-04-02
> >
> > Thank you for your clarification, which has addressed my concerns.

---

> > > ### Author Response · Authors · 2026-04-02
> > >
> > > Thank you for acknowledging that our rebuttal fully addressed your concerns!

---

### Official Review · Reviewer_beZ9 · 2026-03-13

**Soundness:** 3
**Presentation:** 3
**Significance:** 3
**Originality:** 3
**Overall Recommendation:** 5
**Confidence:** 3

**Summary:**

The paper considers the composition of Log-E-Exp (a generalization of log-sum-exp where the sum is replaced with expectation) with random risk functions. The authors term the problem of minimizing the average of $n$ such compositions as Compositional Entropic Risk Minimization (CERM). Examples of applications include multinomial logistic regression with extremely many classes, partial AUC maximization, CLIP and KL-regularized distributionally robust optimization (DRO). The authors turn to the dual formulation, which expresses Log-E-Exp as the infimum over a scalar variable of the exponent expectation. Thus, CERM is cast as minimization over the original and $n$-dimensional dual variables which can be now solved by stochastic first-order methods. The main contribution of the paper is a stochastic proximal mirror descent with an appropriate choice of the Bregman divergence that respects the geometry of the problem. The authors derive the convergence rate of order $1/\sqrt{T}$ in the convex setting, which improves upon existing results in the field. Numerical experiments illustrate that the algorithm performs well in certain scenarios.

**Compliance With Llm Reviewing Policy:**

Affirmed.

**Final Justification:**

The rebuttal addressed my two main concerns, namely, that joint SGD was missing as a baseline, and that the discussion of limitations was insufficient. The authors added joint SGD results showing similar performance to alternating SGD, and agreed to add the limitations discussion, especially regarding stochastic approximation. Other questions were also addressed. Consequently, I raised my score from 4 to 5 (accept). The paper remains theoretically sound, well-written, and the method performs well in experiments.

**Key Questions For Authors:**

1. Is my understanding correct that Theorem 3.6 assumes that the step-sizes are constant (determined by the total number of iterations)? Does SCENT perform well in practice with a constant step-size, as opposed to the cosine schedule that you used?
1. In the extreme classification experiment, why are using a linear model **without bias**?
1. Do you use single precision in the experiments?
1. Suggested reference: K. Kan et al. *LSEMINK: A Modified Newton-Krylov Method for Log-Sum-Exp Minimization*

If joint SGD (with/without Softplus) is added to empirical comparison, and limitations are discussed in the Conclusion section, I am ready to raise the score.

**Limitations:**

Partially discussed

**Strengths And Weaknesses:**

Strengths:
1. The paper looks theoretically sound. Assumptions are stated clearly, results are formulated rigorously, proofs are available (although I haven’t inspected them thoroughly).
1. The work is well-written and has a clear structure.
1. The proposed algorithm performs well empirically.

Weaknesses:
1. As far as the baselines for the dual formulation are concerned, the authors consider alternating SGD, but ignore SGD w.r.t. the joint variable. In fact, the objective is jointly convex (if the risk functions are), and the work of Fagan & Iyengar (2018) discusses joint SGD.
1. Limitations of the algorithm are not properly discussed. For example, each dual variable component is only updated once per epoch; thus, several passes over the data are typically required for convergence, and the use of the algorithm in the stochastic approximation (SA) framework is not possible. Another limitation is that the algorithm’s updates involve exponential quantities that may potentially lead to floating point exceptions if low or moderate precision is used.

---

> ### Author Rebuttal · Authors · 2026-03-31
>
> We thank the reviewer for their valuable comments.
>
> > **Q1**: The authors consider alternating SGD, but ignore SGD w.r.t. the joint variable.
>
> **A:** Thank you for the suggestion! We would like to clarify the following points:
> - This design is consistent with our SCENT algorithm, where $\nu_t$ is first updated and then used to update $\mathbf{w}_{t+1}$. Therefore, comparing with alternating SGD (ASGD) allows us to isolate and directly evaluate the impact of updating $\nu_t$, which is the key difference between the two methods.
> - In addition, we have conducted new experiments comparing against joint SGD. The results (figures in the [anonoymous folder](https://github.com/icml2026scent/icml2026scent/blob/main/sgd_vs_asgd/), commit e649931) indicate that ASGD and joint SGD exhibit similar performance across different settings.
>
> > **Q2**: Limitations of the algorithm are not properly discussed: (1) The algorithm requires several passes for the convergence of the dual variable. (2) The use of the algorithm in the stochastic approximation framework is not possible. (3) The algorithm’s updates involve exponential quantities that may potentially lead to floating point exceptions.
>
> **A**: Thanks for the suggestion.
> - We politely disagree that (1) is a limitation. Please note that the dual variable is maintained and updated to facilitate the convergence of model parameters. BSGD does not maintain the dual variable explicitly but it converges slower than SCENT (see response to Q1 of reviewer HE8X for more analysis). All other baselines maintain dual variables or equivalent variants and also converge slower.  In our experiments on large-scale XC data (Glint360K, TreeOfLife-10M), the baseline algorithms BSGD, ASGD, ASGD (softplus), U-max require over 50 epochs to converge and perform much worse than SCENT.
> - We agree with the reviewer on (2). In the case of stochastic approximation, it is impossible to maintain $\nu$ for each individual sample. We will discuss this limitation in the revision.
> - (3) is not a limitation for SCENT. It admits a stable implementation, as all exponential terms appear either in log-sum-exp or softmax form. The dual variable update (please see line 216, left) involves the log-sum-exp function, which has numerically-stable and efficient implementation in frameworks such as PyTorch (e.g., [torch.logsumexp](https://docs.pytorch.org/docs/stable/generated/torch.logsumexp.html)). For the primal update, the practical implementation is also stable. Let us consider the same random variable $\zeta_t=\zeta'_t$ for computing $\nu_t$ and $\mathbf{z}_t$ in Eqs. (7) and (8). The exponential term in $\mathbf{z}_t$ is $\frac{e^{s(\mathbf{w}_t;\zeta_t)}}{e^{\nu_t}}$. Plugging in the update of $\nu_t$ from (7), we obtain $\frac{e^{s(\mathbf{w}\_t;\zeta\_t)}}{e^{\nu\_t}} = \frac{e^{s(\mathbf{w}\_t;\zeta\_t)}}{(1-\gamma'\_t)e^{\nu\_{t-1}}+\gamma'\_te^{s(\mathbf{w}\_t;\zeta\_t)}}$, where $\gamma'\_t=\frac{\alpha\_t e^{\nu\_{t-1}}}{1+\alpha\_t e^{\nu\_{t-1}}}$. Note that both $\gamma'_t$ and $\frac{e^{s(\mathbf{w}_t;\zeta_t)}}{e^{\nu_t}}$ can be written as a softmax form and hence they admit stable implementations. For example, if $\nu\_{t-1} > s(\mathbf{w}\_t;\zeta\_t)$, $\frac{e^{s(\mathbf{w}\_t;\zeta\_t)}}{e^{\nu\_t}}$ can be computed as $\frac{e^{s(\mathbf{w}\_t;\zeta\_t)-\nu\_{t-1}}}{(1-\gamma'\_t)+\gamma'\_t e^{s(\mathbf{w}\_t;\zeta\_t)-\nu\_{t-1}}}$,  otherwise as $\frac{1}{(1-\gamma'\_t)e^{\nu\_{t-1}-s(\mathbf{w}\_t;\zeta\_t)}+\gamma'\_t}.$ Hence, all computations only involve terms of the form $e^c$ with $c<0$.
>
> > **Q3**: Does Theorem 3.6 assume the step size is constant? What is the comparison between constant and cosine step size in experiments?
>
> **A**: Yes, Theorem 3.6 assumes a fixed step size depending on the total number of iterations. This is a standard practice in the literature of convergence analysis for stochastic methods. In practice, a fixed step size almost always performs worse than a cosine step size.
>
> > **Q4**: In the extreme classification experiment, why are you using a linear model without bias?
>
> **A**: The main focus of our experiments is the optimization speed of different methods. Since adding a bias term is equivalent to adding another component in the primal variable, the optimization problem is only slightly modified and hence adding a bias term should not affect the conclusion from an optimization viewpoint.
>
> > **Q5**: Do you use single precision in the experiments?
>
> **A**: For CLIP model training, we used mixed precision (fp32 and fp16, natively supported by PyTorch) to train the models. For all other experiments, we used the default single precision to train the models.
>
> > **Q6**: Suggested reference.
>
> **A**: Thank you for the suggestion. The mentioned paper solves the log-sum-exp problem using Newton-Krylov methods and is related. The difference is that theirs is deterministic and hence not efficient for solving large-scale problems. We will add the discussion in the revision.

---

> > ### Author Rebuttal · Reviewer_beZ9 · 2026-04-01
> >
> > My questions have been addressed and I am raising the score by one point.

---

> > > ### Author Response · Authors · 2026-04-02
> > >
> > > Thank you for acknowledging that and raising your rating. We appreciate it.

---

### Decision · Program_Chairs · 2026-04-30

**Decision:**

Accept (regular)

**Comment:**

The paper introduces a new optimization algorithm for solving compositional entropic risk minimization. The work formulates the objective into a dual min-min optimization problem, and propose to use stochastic proximal mirror descent updates with a specific Bregman divergence (negative exponential function). The paper establishes a convergence rate of $O(1/\sqrt{T})$ for the convex setting under weaker assumptions than prior works. The contributions are new and should be interesting to audience in optimization.

After the rebuttal, the reviewers mentioned remaining concerns such as the convexity of only training the linear head. One reviewer did not fully engage with author response. It looks like those mentioned weaknesses (e.g., comparisons with more baselines and how Bregman divergence affects convergence) have mostly been addressed by authors. I encourage authors to fully incorporate the feedback to improve the paper in the camera-ready version.